# Labels Are Not All You Need: Evaluating Node Embedding Quality without Relying on Labels

## Abstract

Graph Neural Network (GNN) based node embedding methods are a promising approach to learning node representations for downstream tasks such as link prediction, node classification, and node clustering. GNN-based methods usually work in an unsupervised or semi-supervised manner, learning node representations without or with limited label information. We empirically show, however, that the performance of learned node embeddings on downstream tasks may be heavily impacted by the GNN-method's hyperparameter configuration. Unfortunately, existing hyperparameter optimisation methods typically rely on labeled data for evaluation, making them unsuitable for unsupervised scenarios. This raises the question: *how can we tune the hyperparameters of GNNs without using label information to obtain high quality node embeddings?* To answer this, we propose a framework for evaluating node embedding quality without relying on labels. Specifically, our framework consists of two steps: *building prior beliefs* that characterize high-quality node embeddings, and *quantifying the extent* to which those prior beliefs are satisfied. More importantly, we instantiate our framework from two different but complementary perspectives: spatial and spectral information. First, we introduce the Consensus-based Space Occupancy Rate (CSOR) method that evaluates node embedding quality from a spatial view. It conducts pairwise comparisons of the spatial distances between node embeddings obtained from various hyperparameter configurations. Next, we present the Spectral Space Occupancy Rate (SSOR) method, which takes a spectral perspective and evaluates the embedding quality by examining the singular values of the node embedding matrices. Extensive experiments on seven GNN models with four benchmark datasets demonstrate the effectiveness of both CSOR and SSOR. Specifically, both methods consistently prioritize hyperparameter configurations that yield high-quality node embeddings for downstream tasks.

## 1 Introduction

Graph-structured data is ubiquitous due to its strong expressive capability in representing relationships between objects. Node embedding methods, including traditional approaches (Perozzi et al., 2014; Grover & Leskovec, 2016; Cao et al., 2015; Wang et al., 2016) and Graph Neural Network (GNN)-based methods (Kipf & Welling, 2016b; Pan et al., 2018; Hamilton et al., 2017a; Xu et al., 2018; Velickovic et al., 2017), aim to learn node representations in an unsupervised manner, though they can sometimes also be semi-supervised. These embeddings are then utilized for various downstream tasks such as link prediction (Zhang & Chen, 2018), node classification (Maurya et al., 2021), and graph clustering (Tsitsulin et al., 2023b). Despite the impressive results achieved by these methods, some long-standing challenges remain underexplored. Particularly, we empirically show that the performance of node embeddings in downstream tasks may heavily depend on the hyperparameters (HPs) configurations (see Figures 4, 5,6, 7,8,9, and 10 in Appendix B for details). Therefore, given a node embedding model, to achieve a robust performance on downstream tasks across different datasets, the HPs should be carefully optimised.

However, we found that most existing HP optimisation methods are designed for supervised learning settings (Feurer & Hutter, 2019; He et al., 2021; Bischl et al., 2023), making them unsuitable for

unsupervised learning scenarios where labels are not available. This inevitably leads to a question: *how could we tune the hyperparameters of unsupervised representation learning methods?* Broadly speaking, this can be approached in two distinct ways: through a *meta-learning-based strategy* (Hospedales et al., 2021) or an *internal strategy* (Ma et al., 2023). On the one hand, *meta-learning strategies* compare the internal structure of the labeled dataset to the unlabeled dataset, transferring learned biases to optimize performance on the same learning task. On the other hand, *internal strategies* investigate the internal structure of the unlabeled dataset itself and the behavior of learning model, using some "prior beliefs" (which will be elaborated later) to optimize performance. We argue that the underlying logic of *meta-learning strategies* is very intuitive and thus well understood for humans: given one new dataset, we employ the solution to one previously seen dataset that is closest to the given new dataset. In contrast, the logics behind various *internal strategies* are not uniform, and they are too complex to be condensed into a single explanation. Due to inherent challenges, there is considerably less research available on internal strategies than on meta-learning approaches.

In this paper, we focus on the so-called *internal strategies* for tuning unsupervised models, showing how they can be unified using one single framework. Specifically, "internal" means that the evaluation of embedding quality is performed without external information such as labels. While most existing *internal strategies* are initially designed for unsupervised representation learning on images, we extend them to unsupervised representation learning on graphs and focus on evaluating the quality of node embeddings obtained from unsupervised GNN models. After revisiting existing internal strategies, including UDR (Duan et al., 2020), Incoherence (Tsitsulin et al., 2023a), Self Cluster (Tsitsulin et al., 2023a), $\alpha$-ReQ (Assran et al., 2022), RankMe (Garrido et al., 2023; Roy & Vetterli, 2007), NESum (He & Ozay, 2022), Condition Number (Ben-Israel, 1966; Tsitsulin et al., 2023a), and Stable Rank (Tsitsulin et al., 2023a), we found that the design and development of internal strategies can be distilled into two fundamental steps: 1) **build prior beliefs**, where involves building prior beliefs (namely imposing assumptions) on the characteristics that indicate high-quality embeddings; and 2) **quantify prior beliefs**, which involves assessing the extent to which those prior beliefs are satisfied. With this framework, all existing internal strategies can be analyzed from a unified perspective (see Appendix C for details): using UDR (Duan et al., 2020) as an example, they first build the prior belief that *well-performing HP configurations generate stable, disentangled representations across different random seeds*. This is inspired by (Rolinek et al., 2019), which shows that the reconstruction mechanism in VAEs (Kingma & Welling, 2014) induces local orthogonality that results in disentangled representations and only well-performing HP configurations enable VAEs to exhibit this property. On this basis, they quantify the extent to which this prior belief is satisfied *by measuring the similarity of embeddings generated by the same HP configurations across random seeds*. Higher similarity indicates a higher degree of disentanglement.

When confining our attention to evaluating the quality of node embeddings obtained from unsupervised GNN models, we further instantiate two novel internal strategies within this framework (namely building prior beliefs and quantifying prior beliefs). Before introducing them, we first present spatial and spectral GNNs. Specifically, GNNs can be divided into: 1) **spatial-based** approaches, which aggregate node information directly in the node (spatial) domain by passing messages within local neighborhoods, and 2) **spectral-based** approaches, which transform nodes into the spectral domain using the graph Laplacian for aggregation before mapping back to the node domain. Chen et al. (2023) reveal that spatial- and spectral-based GNNs, while analyzing from different perspectives and employing different techniques, ultimately achieve the same objective. Building on this, we instantiate our framework from a spatial perspective by proposing CSOR, while developing SSOR from a spectral perspective.

**Consensus-based spatial Space Occupancy Rate (CSOR).** We begin by conducting extensive experiments with various unsupervised GNN models on four benchmark graph datasets, aiming to empirically draw inspirations from observations to build prior beliefs. Specifically, given a GNN model on a specific graph, we empirically observe that: among all node embeddings generated with different HP values, those that exhibit greater spatial separability from other embeddings tend to perform better in downstream tasks. On this basis, we build the following prior belief: *a given set of HP configurations that makes node embeddings more distinct among all embeddings is preferable* (which will be explained later). Based on this prior belief, we propose CSOR, which quantifies this prior belief by *comparing the spatial distances between embeddings obtained with different HP*

*configurations through pairwise comparisons.* The process of quantifying the prior belief is actually the process of assessing node embedding quality (or performing HP optimisation).

**Spectral Space Occupancy Rate (SSOR).** Node embeddings can also be analyzed from a spectral perspective. By examining the singular values of the node embedding matrices, we can gain insights into embedding quality, offering a complementary view to spatial-based evaluation. Based on this, we propose SSOR, which shares the same prior belief as CSOR but quantifies it differently. SSOR quantifies this belief *by analyzing the singular values of the node embedding matrices, where a higher sum of singular values (which are simultaneously distributed uniformly across different dimensions as much as possible) indicates higher quality.*

We conducted extensive experiments using seven unsupervised GNN models on four benchmark datasets to demonstrate the effectiveness of CSOR and SSOR. Results show that both methods consistently perform well for all GNN models across datasets. In several cases, they can even select the optimal HP configurations. Specifically, for obtained node embeddings we calculate the Spearman correlation coefficient (Zar, 2005) between the ranking scores provided by CSOR (or SSOR) and the actual performance metrics on downstream tasks. The correlation coefficients are often around 0.9 and never fall below 0.6, showing that CSOR and SSOR can effectively distinguish between different HP configurations and identify those that produce high-quality node embeddings.

Overall, the contributions of this paper are summarized as follows: 1) We propose a framework for developing internal strategies by establishing general principles: building prior beliefs and quantifying prior beliefs; 2) More importantly, we instantiate our framework from two different but complementary perspectives: CSOR from a spatial perspective and SSOR from a spectral perspective. We conduct extensive experiments to validate the effectiveness of both methods using seven unsupervised GNN models on four benchmark datasets; 3) To facilitate future research, we establish a generic testbed that allows researchers and practitioners to evaluate the effectiveness of our automatic HPO methods (or their newly proposed methods) on various unsupervised node embedding algorithms. The testbed is publicly available on Anonymous GitHub.

## 2 PROBLEM STATEMENT

Due to space limitations, the preliminaries and related work are deferred to Appendix E, and we directly begin by problem statement as follows.

An *attributed graph* is defined as $\mathcal{G} = (\mathcal{V}, \mathcal{E}, \mathbf{X})$, where $\mathcal{V} = \{v_1, ..., v_N\}$ represents the set of nodes, $\mathcal{E} = \{e_1, ..., e_M\}$ denotes the set of edges, and $\mathbf{X} \in \mathbb{R}^{N \times Q}$ is the node attribute matrix, with $N = |\mathcal{V}|$ being the number of nodes and $Q$ the dimensionality of the node attributes. Alternatively, the graph can be represented as $\mathcal{G} = (\mathbf{A}, \mathbf{X})$, where $\mathbf{A}$ is the adjacency matrix, with $\mathbf{A}_{ij} = 1$ if there is an edge between node $v_i$ and $v_j$, and $\mathbf{A}_{ij} = 0$ otherwise.

A (node-level) *unsupervised graph representation learning* function $f(\cdot)$ takes a graph $\mathcal{G}$ as input and outputs a node embedding vector for each individual node. Formally, we define $f : \mathcal{G} \to \mathbf{Z} \in \mathbb{R}^{N \times D}$, where $N$ is the number of nodes and $D$ denotes the dimensionality of the learned node embeddings. In this paper, we aim to address the following problem:

**Problem** (Hyperparameter Optimization for Unsupervised Graph Representation Learning ). Given a *graph* $\mathcal{G}$, an *unsupervised graph representation learning algorithm* $f(\cdot)$, and a set of *HP configurations* $\mathcal{H}$ for $f(\cdot)$, we aim to develop a HP optimization (HPO) method that can select an optimal HP configuration $h^* \in \mathcal{H}$ without relying on labels, such that the node embeddings obtained using $f_{h^*}(\cdot)$ can achieve optimal performance (which will be defined later based on the specific downstream task).

The main challenge of solving this problem lies in the absence of ground truth labels in unsupervised settings, rendering the evaluation of the HPO method inherently difficult. Furthermore, the unique characteristics of graph-structured data, especially its non-i.i.d. nature in graph representation learning, make this problem even more challenging to tackle. To address this, we employ the so-called *internal strategy* to evaluate the quality of graph embeddings without relying on labels.

**Definition** (Internal Strategy for Evaluating Node Embeddings). *Given a graph $\mathcal{G}$, an unsupervised graph representation learning algorithm $f(\cdot)$, a set of HP configurations $\mathcal{H}$, and the resulting node embedding matrices $\{\mathbf{Z}(h) \mid h \in \mathcal{H}\}$, where $\mathbf{Z}(h) = f_h(\mathcal{G})$, an internal strategy $\mathcal{Q} : \mathbb{R}^{N \times D} \to \mathbb{R}$*

*is defined as a function that takes a node embedding matrix $\mathbf{Z}$ as input and outputs a ranking score $s \in \mathbb{R}$. Formally, we have $s = \mathcal{Q}(\mathbf{Z}(h))$.*

Specifically, the internal strategy evaluates the quality of the node embeddings by assigning a ranking score, which allows for the comparison of different HP configurations. This ranking can then be used to select the optimal HP configuration that leads to the best node embeddings for downstream tasks.

## 3    REVISITING UDR AND BEYOND

By revisiting Unsupervised Disentanglement Ranking (UDR) (Duan et al., 2020), we demonstrate how the two fundamental steps in our proposed framework are motivated and performed: 1) building prior beliefs about what constitutes a good embedding; and 2) quantifying these prior beliefs.

**Inspiration.** Rolinek et al. (2019) reveal that the disentangled properties of Variational Autoencoders (VAEs) arise from their inherent reconstruction mechanism. The decoder's task in VAE models shares similarities with Principal Component Analysis (PCA) (Maćkiewicz & Ratajczak, 1993; Jolliffe & Cadima, 2016), as both aim to capture and reconstruct the data's key patterns through independent components. This results in the decoder encouraging orthogonal and disentangled latent variables from the encoder. Additionally, the imposition of a diagonal prior on the latent space pushes the encoder to produce locally orthogonal representations, further enhancing the disentangling effect. This interplay between the reconstruction objective and the diagonal prior naturally leads to disentangled representations without the need for explicit design in the model.

**Building and Quantifying Prior Beliefs.** Based on the understanding of how VAEs achieve disentanglement, UDR (Duan et al., 2020) is proposed to optimize HPs in unsupervised representation learning on images. Specifically, they build the following prior belief: *the reconstruction objective in VAEs is unique, causing well-performing HP configurations to generate stable, disentangled hidden variables, as the decoder (due to its PCA-like behavior) pushes the encoder towards producing robust, disentangled representations.* UDR's prior belief is rooted in the concept that "happy families are all alike; every unhappy family is unhappy in its own way" (Tolstoy, 2016), meaning that high-quality HP configurations produce consistent, stable representations across different random seeds, while poor HP configurations lead to diverse, unstable representations. UDR quantifies this prior belief through a consensus-based method (see Appendix E for definition), *measuring the similarity of representations generated under different random seeds.* By evaluating the consistency of these representations, UDR identifies the most disentangled and effective HP configurations.

For completeness, UDR works as follows. They train $H \times S$ models, where $H$ is the number of HP configurations, and $S$ is the number of seeds for initial model weights. For each HP configuration $h \in \mathcal{H}$,

1. they sample $P$ random seeds $\{\text{seed}_1, ..., \text{seed}_P\}$ with $P \leq S$, and obtain embeddings $\{\mathbf{Z}_{(h_1)}, ..., \mathbf{Z}_{(h_P)}\}$ using the learning model configured with $h$ under random seeds $\{\text{seed}_1, ..., \text{seed}_P\}$, respectively;

2. they conduct $\binom{P}{2}$ pairwise comparisons of embeddings $\{\mathbf{Z}_{(h_1)}, ..., \mathbf{Z}_{(h_P)}\}$ to obtain a list of scores $\{\text{UDR}_h^j | j \in \{1, 2, ..., \binom{P}{2}\}\}$; Without loss of generality, suppose the $j$-th pair is $(h_1, h_2)$, then $\text{UDR}_h^j = \text{sim}(\mathbf{Z}_{(h_1)}, \mathbf{Z}_{(h_2)})$ with $\text{sim}(\cdot)$ a similarity metric;

3. on this basis, they compute the median of this list as the final UDR score, namely $\text{UDR}_h = \text{median}\left\{\text{UDR}_h^j \mid j = 1, \ldots, \binom{P}{2}\right\}$, where a higher UDR score indicates a better HP configuration.

**Discussion.** UDR's development involves building prior beliefs (or drawing inspiration) from the mechanism analysis of VAEs, which raises the question: *Can or should we rely solely on analyzing the mechanisms of representation learning models to build prior beliefs, or are there alternative approaches?* To address this, we must consider two key points: 1) Given the inherent uncertainty and complexity of neural networks (Lipton, 2016), it is not always feasible to analyze their mechanisms from a mathematical perspective; 2) Despite the opacity of learning models, we can partially under-

stand their internal workings by observing the relationships between inputs and outputs—a principle underlying many post-hoc explainable AI (XAI) techniques (Arrieta et al., 2020).

In other words, while UDR was developed through the mechanism analysis of VAEs, we note that the high level of disentanglement can be observed in the outputs of VAEs (e.g., image embeddings) without directly studying their mechanisms. This suggests that it is not always necessary to analyze the model's internal mechanisms when dealing with complex models. Instead, we can relax this requirement by using observation-driven methods to build prior beliefs.

## 4 CONSENSUS BASED SPATIAL SPACE OCCUPANCY RATE (CSOR): A SPATIAL PERSPECTIVE

We now demonstrate how to leverage this observation-driven approach to draw inspirations and then build prior beliefs, where we assume that the distribution of embeddings can encapsulate the characteristics of the underlying mechanisms of GNN models. Next, we quantify these beliefs to evaluate unsupervised node embedding quality, resulting in a novel internal strategy dubbed Consensus-based spatial Space Occupancy Rate (CSOR).

**Visualisation.** Node embeddings from GNN models are typically high-dimensional (8 in our experiments), making it difficult to directly observe their distribution. With the help of PCA, we can visualize the distribution of node embeddings in a 3-dimensional space. PCA is used as it can effectively reduce the dimensionality while preserving much of the original variance (Jolliffe, 2002).

**Observations and Inspirations.** To draw some inspirations from node embeddings distributions and understand which characteristics lead to better performance in downstream tasks, we experiment with 1280 sets of HP configurations (see Appendix F for setting details). We select four node embeddings which are evenly sampled from the worst to the best performance on downstream node classification task, and visualize them with PCA. From Figure 1 (and more figures in Appendix F.1), we observe that when node classification performance is poor, the embeddings of nodes from different classes are mixed together. As performance improves, the separability between embeddings of nodes from different classes increases. Drawing inspirations from these observations, we propose a hypothesis: *node embeddings associated with a specific HP configuration become increasingly dispersed as downstream performance improves, which we refer to as intra-embedding. Simultaneously, node embeddings from a specific well-performing HP configuration tend to diverge further from those of poorly-performing configurations as its downstream performance improves, which we refer to as inter-embedding.* To validate this hypothesis, we use the worst-performing node embedding matrix as a baseline and calculate the Manhattan distance (Krause & Golovin, 2014) from all other embedding matrices to this baseline, effectively capturing aggregate differences across dimensions. As shown in Figure 42 (a) in Appendix F.2, we observe that node embedding matrices (where a point corresponds to an embedding matrix) farther from the worst-performing one tend to show better performance. This supports our hypothesis that the quality of node embedding matrices improves as they become spatially more distant from the worst-performing embedding matrix.

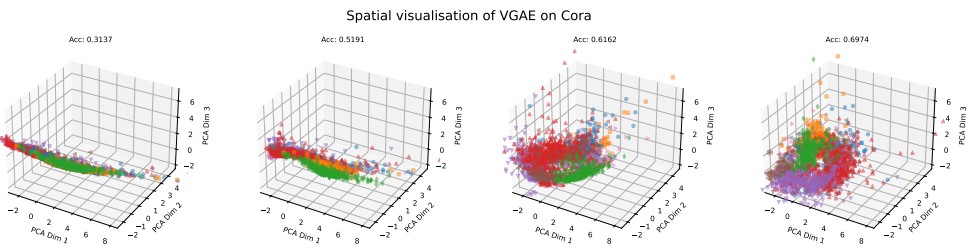

Figure 1: Relationship between node embedding distribution and node classification accuracy. Nodes are colored by labels to illustrate clustering. In this experiment on the Cora dataset using the VGAE model, higher Accuracy values (indicating better performance) correspond to more dispersed embedding distributions. (Experiment settings and more figures are given in Appendix F).

**Building and Quantifying Prior Beliefs.** Based on above observations and analyses, we empirically validated that node embeddings spatially farther from the worst-performing one exhibit higher qualities. We term this characteristic as *distinctness*, where a higher value means higher quality. On this basis, we build the following prior belief: *for node embeddings generated by message-passing-based GNNs, greater spatial distinctness correlates with higher performance in downstream tasks*. To quantify prior belief, we define *spatial distinctness* as the degree to which a node embedding matrix is spatially separated from *other node embedding matrices*. It is important to note that: due to the lack of label information, it is impossible to know the worst-performing embedding matrix on downstream task. Therefore, we compare each individual node embedding matrix to other node embedding matrices (which are actually used as baseline) rather than the worst-performing one. In this way, we are actually assuming that "poor-performing HP configurations are all alike" (e.g., the embeddings of nodes from different classes are mixed). In other words, we employ the consensus formed by all node embedding matrices as a baseline to quantify the prior belief. By comparing Figure 42 (a) and (b) in Appendix F.2, we can see that *spatial distinctness* calculated based on this consensus baseline (corresponding to Figure 42 (b)) shows a stronger correlation with performance in downstream tasks compared to using the worst-performing embedding matrix as the baseline (corresponding to Figure 42 (a)). This is because performing pairwise comparisons (corresponding to the consensus baseline) can capture more distributional information about all node embedding matrices.

**Formalisation of CSOR.** Given a graph representation learning model $f(\cdot)$ with a specific HP configuration $h \in \mathcal{H}$, denoted as $f_h(\cdot)$, the model maps nodes of a graph $\mathcal{G}$ to an embedding matrix $\mathbf{Z}(h)$, where $\mathbf{Z}(h) = f_h(\mathcal{G})$. Using this notation, we propose an internal strategy for HPO in unsupervised graph representation learning, called Consensus-based spatial Space Occupancy Rate (CSOR), which maps the node embedding matrix $\mathbf{Z}(h)$ to a ranking score $s \in \mathbb{R}$, quantifying the quality of the embeddings. Specifically, CSOR is designed as follows:

- For each pair of HP configurations $(h_i, h_j)$ with $i \neq j$, we calculate the difference $D_{i,j}$ between the resulting embedding matrices $\mathbf{Z}(h_i)$ and $\mathbf{Z}(h_j)$, with $D_{i,j} = \text{diff}(\mathbf{Z}(h_i), \mathbf{Z}(h_j))$ and $\text{diff}(\cdot, \cdot)$ is the Manhattan distance.

- Next, the CSOR score $s_i$ of HP configuration $h_i$ is computed as $\sum_{j=1, j \neq i}^{|\mathcal{H}|} D_{i,j}$, where $\mathcal{H}$ is the set of investigated HP configurations. Then the optimal HP configuration is determined as: $h^* = \arg \max_{h_i \in \mathcal{H}} \{s_i \mid i = 1, \ldots, |\mathcal{H}|\}$.

Specifically, the pseudocode for performing HPO with CSOR is given in Algorithm 1 and its complexity analysis is given in Appendix G.

## 5 SPECTRAL SPACE OCCUPANCY RATE (SSOR): A SPECTRAL PERSPECTIVE

In CSOR, we built the following prior belief: *for node embeddings generated by message-passing-based GNNs, greater spatial distinctness correlates with higher performance in downstream tasks*; To quantify this prior belief, we defined and measured the *spatial distinctness* from a spatial perspective. Now, we approach this problem from a spectral-based perspective, and present a novel internal strategy called Spectral Space Occupancy Rate (SSOR). To achieve this, we first build a similar prior belief: *for node embeddings generated by message-passing-based GNNs, greater spectral distinctness correlates with higher performance in downstream tasks*; To quantify this prior belief, we need to define and measure the *spectral distinctness*. Before giving its formal definition, we present the intuition and rational behind it.

**Intuition and Rational behind Spectral Distinctness**. Given a node embedding matrix, performing Singular Value Decomposition (SVD) on it can yield singular values that represent the extent to which the node embeddings are spreading across different dimensions in the latent space. We argue that a singular value can capture the spread extent of node embeddings in some (virtual) dimension, which does not necessarily correspond to a specific dimension in the latent space. Intuitively, by simply summing all the singular values, we can obtain the total spread extent of node embeddings across all dimensions in the latent space. Therefore, a higher value, indicating a larger total extent of spread, is associated with better separability. From this perspective, it is intuitive to use the sum of singular values to represent the distinctness of node embeddings in the spectral space, referred to as *spectral distinctness*. However, simply maximizing the sum of singular values can lead to

an undesirable effect known as *dimensional collapse* (Jing et al., 2021). This phenomenon occurs when embeddings fail to capture the full variability of the data and collapse into a lower-dimensional subspace, resulting in poor downstream performance. In this case, a high sum of singular values can be misleading if dominated by a few large singular values while the others remain very small. To mitigate this, we propose to maximise the sum of singular values while ensuring that the singular values are uniformly distributed as much as possible. For instance, if we have three groups of singular values $(4, 0, 0, 0), (2, 2, 0, 0), (1, 1, 1, 1)$, the group $(1, 1, 1, 1)$ is preferred. This leads to Spectral Space Occupancy Rate (SSOR), which can effectively quantify the *spectral distinctness* and thus the prior belief.

**High-level Idea of SSOR and Empirical Evidence.** Conceptually, give a node embedding matrix, our proposed Spectral Space Occupancy Rate (SSOR) approach attempts to quantify the prior belief by *computing the area occupied in spectral space spanned by the normalized singular values*. To visually demonstrate this, we distribute all the normalized singular values (which will be formally defined later) evenly across the 360 degrees in a 2-dimensional radar chart as shown in Figure 2. In this way, each normalised singular value represents one (virtual) dimension, and its value indicates the spread extent of the node embeddings in that dimension. By connecting these line segments, we form an irregular polygon. The area of this polygon is analogous to the *spatial distinctness* in CSOR, which is obtained by accumulating pairwise distances. From Figure 2 (and more figures in Appendix F.3), we can observe that as the downstream performance (i.e., Accuracy) of the node embedding matrix improves, the area of the irregular polygon formed by the singular values of the node embedding matrix becomes larger. This empirically shows that *quantifying the spectral distinctness of node embeddings using the area is effective*. Figure 77 and other figures in Appendix F.4 give more empirical evidences.

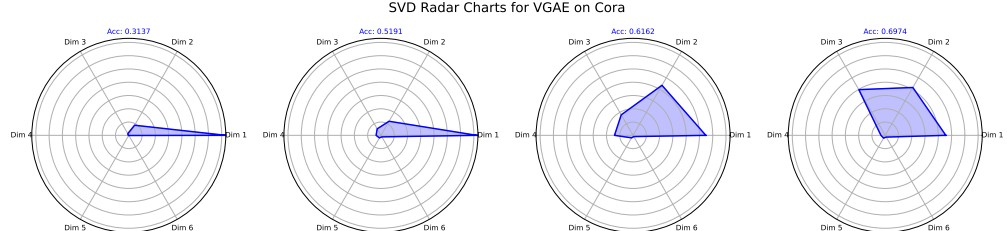

Figure 2: Relationship between spectral space occupancy rate and downstream node classification performance. We ran VGAE on the Cora dataset with 1280 HP configurations and selected 4 node embedding matrices, ranging from the worst to the best performance (Accuracy). Detailed settings and more figures are given in Appendix F.3.

**Formalisation of SSOR.** Given a node embedding matrix $\mathbf{Z}(h)$, we first perform the Singular Value Decomposition (SVD) on it, namely $\mathbf{Z}(h) = U\Sigma V^T$, where $\Sigma$ is a diagonal matrix containing the singular values $\sigma_i$. Next, we normalise the singular values as follows: $\tilde{\sigma}_i = \sigma_i/\sigma_{\max}$, which are used as vertices of a radar chart in Figure 2. Finally, the area of the radar chart, representing the spectral space occupancy rate, is computed as follows:

$$SSOR(\mathbf{Z}(h)) = 0.5 * \left| \sum_{i=1}^{r-1} \tilde{\sigma}_i \tilde{\sigma}_{i+1} \sin(\theta_{i+1} - \theta_i) \right|,$$

where $\theta_i$ indicates the angle between vertex $i$ (i.e., normalised singular value $\tilde{\sigma}_i$) and vertex 1 (namely $\tilde{\sigma}_1$) in polar coordinates. Given a node embedding matrix, the SSOR score considers both the magnitude of singular values and the evenness of their distribution. A larger area implies higher spectral space occupancy rate, suggesting more informative embeddings for downstream tasks. Due to space limitations, the complexity analysis of SSOR is given in Appendix G.

## 6 EXPERIMENTS

In this section, we conduct extensive experiments to answer the following research questions:

RQ1. Can CSOR and SSOR effectively rank HP configurations to identify those that can produce well-performing node embeddings for downstream tasks?

RQ2. How do CSOR and SSOR compare to state-of-the-art internal strategies in selecting optimal HP configurations for downstream tasks?

## 6.1 EXPERIMENTAL SETTINGS

Specifically, we consider four benchmark datasets, including Cora, Citeseer, Pubmed, and DBLP; Moreover, we consider the following GNN models due to their superior performance in learning node embeddings: GAE (Kipf & Welling, 2016b), VGAE (Kipf & Welling, 2016b), ARGA (Pan et al., 2018), ARGVA (Pan et al., 2018), GraphSAGE (Hamilton et al., 2017a), GIN (Xu et al., 2018), and GAT (Velickovic et al., 2017). For each GNN model, we generate a wide range of HP configurations by varying the number of layers, hidden dimensions per layer, and the number of maximal training epochs. In addition, we consider two typical downstream tasks, node classification and link prediction. For node classification, performance is evaluated using accuracy, while link prediction performance is measured by AUC-ROC.

Importantly, we consider a wide range of internal strategies as baselines, including Incoherence (Tsitsulin et al., 2023a), Self Cluster (Tsitsulin et al., 2023a), $\alpha$-ReQ (Assran et al., 2022), RankMe (Garrido et al., 2023; Roy & Vetterli, 2007), NESum (He & Ozay, 2022), Condition Number (Ben-Israel, 1966; Tsitsulin et al., 2023a), and Stable Rank (Tsitsulin et al., 2023a). To validate the effectiveness of CSOR and SSOR and compare them with other baselines, we consider the following evaluation metrics: 1) **Spearman coefficients** that assess the correlation between the ranking scores given by an internal strategy and the actual downstream task performance (either Accuracy for node classification or AUC-ROC for link prediction), 2) **Actual downstream performance** (in terms of Accuracy or AUC-ROC) of the node embedding matrix resulted by selected HP configuration, and the **relative rankings** of different internal strategies based on downstream performance. Due to space constraints, more details about datasets, GNN models, HP configurations, downstream tasks, and evaluation metrics are given in Appendix A.

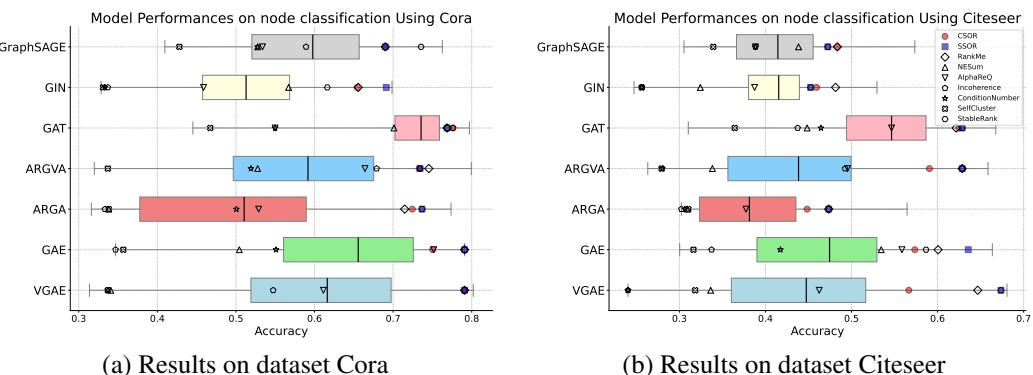

(a) Results on dataset Cora  (b) Results on dataset Citeseer

Figure 3: Actual downstream performance results across 7 GNN node embedding models on Cora and Citeseer (more results on Pubmed and DBLP are given in Figure 11 in Appendix D), with 1280 sets of different HP configurations for each combination of GNN model and dataset. These box plots show the node classification performance (in terms of accuracy values) of node embedding matrices resulted by selected HP configuration for different internal strategies. Particularly, we highlight the performances corresponding to three internal strategies with colors: blue squares for CSOR, red triangles for SSOR, and white diamonds for RankMe (which is the strongest baseline in most cases). We see that the performance of CSOR and SSOR are usually comparable to RankMe, and they can all effectively select well-performing HP configurations.

## 6.2 EXPERIMENT RESULTS AND ANALYSIS

Figures 3 and 11 (in Appendix D) illustrate the downstream performance of node embeddings produced by HP configurations selected through various internal strategies, with node classification

Table 1: Experimental results for node classification: accuracy values (relative rankings) of various internal strategies across 7 GNN models on the Cora and Citeseer datasets. Additional results for Pubmed and DBLP are deferred to Table 5 in Appendix H due to space limitations.

| Dataset | Method | VGAE | GAE | ARGA | ARGVA | GAT | GIN | GraphSAGE | Avg. Rank |
|---|---|---|---|---|---|---|---|---|---|
| Cora | CSOR | 0.79(1) | 0.75(5) | 0.72(3) | 0.73(4) | 0.78(1) | 0.65(3) | 0.69(2) | 2.71 |
| | SSOR | 0.79(1) | 0.79(1) | 0.74(1) | 0.73(2) | 0.77(3) | 0.69(2) | 0.69(2) | **1.57** |
| | RankMe | 0.79(1) | 0.79(1) | 0.71(4) | 0.75(1) | 0.77(3) | 0.66(2) | 0.69(2) | 2.00 |
| | NESum | 0.34(7) | 0.50(7) | 0.34(7) | 0.53(7) | 0.70(6) | 0.57(5) | 0.53(7) | 6.57 |
| | AlphaReQ | 0.61(5) | 0.75(4) | 0.53(5) | 0.66(6) | 0.55(7) | 0.46(6) | 0.53(6) | 5.57 |
| | Incoherence | 0.55(6) | 0.35(9) | 0.33(9) | 0.68(5) | 0.78(1) | 0.34(7) | 0.59(5) | 6.00 |
| | ConditionNumber | 0.34(8) | 0.55(6) | 0.50(6) | 0.52(8) | 0.55(7) | 0.33(8) | 0.53(7) | 7.14 |
| | SelfCluster | 0.34(8) | 0.36(8) | 0.34(7) | 0.34(9) | 0.47(9) | 0.33(9) | 0.43(9) | 8.43 |
| | StableRank | 0.79(1) | 0.79(1) | 0.74(1) | 0.73(2) | 0.77(3) | 0.62(4) | 0.74(1) | 1.86 |
| Citeseer | CSOR | 0.57(4) | 0.57(4) | 0.45(4) | 0.59(4) | 0.62(3) | 0.46(2) | 0.48(1) | 3.14 |
| | SSOR | 0.67(1) | 0.64(1) | 0.47(1) | 0.63(1) | 0.63(1) | 0.45(3) | 0.47(3) | **1.57** |
| | RankMe | 0.65(3) | 0.60(2) | 0.47(1) | 0.63(1) | 0.62(4) | 0.48(1) | 0.48(1) | 1.86 |
| | NESum | 0.34(6) | 0.53(6) | 0.31(7) | 0.34(7) | 0.45(7) | 0.32(6) | 0.44(5) | 6.29 |
| | AlphaReQ | 0.46(5) | 0.56(5) | 0.38(5) | 0.50(5) | 0.55(5) | 0.39(5) | 0.39(6) | 5.14 |
| | Incoherence | 0.24(8) | 0.34(8) | 0.30(9) | 0.49(6) | 0.44(8) | 0.26(7) | 0.39(6) | 7.43 |
| | ConditionNumber | 0.24(8) | 0.42(7) | 0.31(8) | 0.28(8) | 0.46(6) | 0.26(7) | 0.39(6) | 7.14 |
| | SelfCluster | 0.32(7) | 0.32(9) | 0.31(6) | 0.28(9) | 0.36(9) | 0.26(7) | 0.34(9) | 8.00 |
| | StableRank | 0.67(1) | 0.59(3) | 0.47(1) | 0.63(1) | 0.63(1) | 0.45(3) | 0.47(3) | 1.86 |

being the downstream task in this case. Additional results for link prediction are shown in Figure 12 and 13 in Appendix D. These figures display the performance distribution across different strategies. To provide further detail and insights, Tables 1 and 5 (in Appendix H) present accuracy values and rankings of each strategy for node classification, while Table 6 in Appendix H shows similar results for AUC-ROC values and rankings in link prediction. Moreover, Tables 7 and 8 in Appendix H present the Spearman correlation coefficients between the ranking scores produced by the internal strategies and the actual downstream task performance—AUC-ROC for link prediction (Table 7) and accuracy for node classification (Table 8). These results further validate the effectiveness of the proposed methods. From these figures and tables, we can answer the research questions as follows.

**Answer to RQ1 (Effectiveness of CSOR and SSOR in selecting HP configurations).** We answer this question from two aspects: 1) the actual downstream performance of the selected HP configurations, and 2) the Spearman correlation between ranking scores and actual downstream performance.

- First, as shown in Figures 3 and 11, CSOR and SSOR often select HP configurations that yield strong node classification performance, often outperforming 75% of configurations and, in some cases, approaching the best possible performance. For example, in models like ARGVA, our methods sometimes identify the optimal HP configurations, demonstrating their effectiveness in the HPO task. Similar results can be observed for link prediction in Figure 12 and 13.

- Second, to assess whether CSOR and SSOR consistently prioritize well-performing HP configurations (rather than selecting one by chance), we refer to the Spearman correlation results in Tables 7 (for link prediction) and 8 (for node classification). A high correlation between ranking scores and actual downstream performance indicates that the internal strategy reliably assigns higher scores to better-performing configurations, rather than relying on chance (as Figures 3, 11, 12and 13 only show the results on best-performing configuration). Concretely, the strong correlation coefficients in Tables 7 and 8 further validate the effectiveness of our methods. From Table 7, we can see that across seven GNN models and four datasets, with 1280 HP configurations, the average Spearman coefficient is 0.906 for CSOR, 0.969 for SSOR. These high values reflect a strong positive correlation between our ranking scores and actual performance, reinforcing that our methods consistently identify well-performing node embeddings across diverse experiments, rather than selecting them by chance.

**Answer to RQ2 (Comparison with SOTA internal strategies).** Similar to answering RQ1, we address this question from two perspectives:

- First, we consider the actual downstream performance and their relative rankings. From Figures 3, 11, 12 and 13, we observe that SSOR and CSOR often outperform weaker baselines such as NESum, AlphaReQ, Incoherence, Condition Number, and SelfCluster, and are comparable to the

strongest baselines, RankMe and StableRank. These results are further supported by the average ranking results shown in Tables 1, 5, and 6. For the node classification task across 28 HPO experiment settings (7 GNN models on 4 datasets), CSOR achieved the best performance 8 times (average rank 2.79), and SSOR 15 times (average rank 1.75). In the link prediction task, CSOR performed best 5 times (average rank 3.36), while SSOR excelled 18 times (average rank 1.39). This demonstrates that CSOR, and especially SSOR, are competitive with existing state-of-the-art internal strategies.

- Second, we consider the Spearman correlation between ranking scores and actual downstream performance by investigating the results from Tables 7 and 8. It can be seen that CSOR, especially SSOR, often outperform weaker baselines by a large margin. Meanwhile, they are comparable to the strongest baselines RankMe and StableRank in terms of consistently prioritising well-performing HP configurations.

## 7 DISCUSSIONS AND CONCLUSIONS

In this paper, we present a unified framework for developing *internal strategies* to evaluate the quality of node embeddings without the need for labels. Our approach is grounded in two fundamental principles: *building prior beliefs* and *quantifying these beliefs*. Firstly, we identified that prior beliefs about the quality of node embeddings can be built either through analyzing the mechanisms of representation learning models or through an observation-driven approach. We introduced spatial-based and spectral-based methods as two different but complementary ways of building these prior beliefs. The spatial-based method, CSOR, derives prior beliefs from the spatial distribution characteristics of the node embeddings. The spectral-based method, SSOR, observes the singular values of the embedding matrices to form similar prior beliefs. Secondly, we developed methods to quantify these prior beliefs. We demonstrated that quantification could be approached through consensus-based methods, which involve pairwise comparisons of embeddings generated with different hyperparameter values, as exemplified by CSOR. Alternatively, a stand-alone approach can be used, as in the case of SSOR, which leverages singular values directly for quantification without the need for comparative analysis. Through extensive experiments involving seven GNN models across four benchmark datasets, and 1280 sets of HP configurations for each combination of model and dataset, we validated the effectiveness of our proposed methods. The results consistently showed that both CSOR and SSOR could reliably evaluate the quality of node embeddings and identify well-performing HP configurations. Our methods exhibited strong correlations with actual performance metrics, indicating their high accuracy and stability.

**Limitations and Future Work.** In our experiments, we have only tested our internal strategies on GNN models based on the message passing mechanism. It remains to be seen whether our observations and conclusions hold for other types of node embedding models. Additionally, the four datasets we used are all homogeneous. If we were to use heterogeneous datasets, would our internal strategies still be effective? Regarding the downstream tasks to perform quantitative evaluations, we used link prediction and node classification, where link prediction has less bias compared to node classification but still cannot completely eliminate bias as unsupervised evaluation metrics do. Furthermore, if we attempt to use deeper graph neural networks, the issue of oversmoothing may arise. Can our internal strategies solve this problem? Therefore, future work should expand the scope of experiments to include a wider variety of GNN models and datasets. This will help to further validate the stability and generalizability of our methods.

Overall, our work not only introduces effective methods for unsupervised node embedding evaluation but also provides a clear direction for future research in developing internal strategies. By formalizing the building and quantifying of prior beliefs, we lay the groundwork for more sophisticated and reliable evaluation methods in the field of machine learning.

**Reproducibility Statement**. To ensure reproducibility, we include detailed explanations of our experimental setup in Appendix A, and complete experimental results in Appendix B, D, F, and H. Our code is available on Anonymous GitHub.

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

## APPENDIX FOR "LABELS ARE NOT ALL YOU NEED: EVALUATING NODE EMBEDDING QUALITY WITHOUT RELYING ON LABELS"

### TABLE OF CONTENTS

## A  EXPERIMENTAL SETTINGS

To answer the research questions, we perform the following steps, which ensure a thorough evaluation of our proposed approaches and their ability to select optimal HP configurations for GNN models.:

1. *Datasets:* We use four benchmark datasets: Cora, Citeseer, Pubmed, and DBLP (more details see Appendix A.2).

2. *GNN Models:* We evaluate seven different GNN models: VGAE, GAE, GAT, ARGA, ARGVA, GIN, and GraphSAGE (see Appendix A.3) .

3. *Hyperparameter Configurations:* For each GNN model, we generate multiple node embeddings using various hyperparameter configurations (see Appendix A.5).

4. *Evaluation Metrics:* We measure the performance of each hyperparameter configuration using the AUC value for a downstream task, link prediction. Additionally, we calculate the Spearman coefficient to assess the correlation between our ranking scores and the actual performance (see Appendix A.6).

5. *Comparison with Baselines:* We compare the performance of our proposed methods (CSOR and SSOR) with the best-performing existing methods RankMe, as well as with the median and top AUC values of all HP configurations, to determine their relative effectiveness (see Section 6.2).

In the reminder of this section, we will introduce the experimental settings and the motivations for choosing them, including the datasets, the GNN models used to generate graph embeddings, the downstream task for evaluation, and the candidate HPs.

All experiments are implemented in Python 3.10 (using PyTorch v2.0.1 and TensorFlow v2.13.0) and executed on a machine equipped with an AMD EPYC 9354 CPU (16 cores, 60.1 GB RAM), and an Nvidia RTX 4090 GPU (25.2 GB video memory). The system operates with Docker v20.10.10 and a 751.6 GB SSD.

## A.1 DOWNSTREAM TASK

In this work, we use two downstream tasks: node classification and link prediction. Node classification focuses on predicting the label of a node based on its own features as well as the local graph structure, placing emphasis on the node's neighborhood information. In contrast, link prediction is concerned with determining whether an edge exists between two nodes, focusing more on capturing the overall structural relationships within the graph.

By using both tasks, we gain a more comprehensive evaluation of the quality of node embeddings. Node classification assesses how well embeddings capture local neighborhood patterns, while link prediction evaluates the model's ability to represent global structural information. Together, these tasks allow us to better assess whether the Internal Strategy can select hyperparameters that holistically evaluate the quality of node embeddings, rather than being biased toward a specific downstream task.

## A.2 DATASETS

Table 2: Citation datasets

| Name | #nodes | #edges | #features | # classes |
|---|---|---|---|---|
| Cora | 2,708 | 10,556 | 1,433 | 7 |
| CiteSeer | 3,327 | 9,104 | 3,703 | 6 |
| PubMed | 19,717 | 88,648 | 500 | 3 |
| DBLP | 17,716 | 105,734 | 1,639 | 4 |

The datasets Cora, Citeseer, and Pubmed come from Yang et al. (2016), while DBLP comes from Bojchevski & Günnemann (2017). In all these datasets, nodes represent papers and edges represent citation links. All four datasets belong to the category of citation networks. We chose these datasets because they are well-established benchmarks in the field of graph neural networks and citation networks, providing a diverse set of characteristics and challenges for evaluating node embeddings. Details of the datasets are shown in Table 2.

## A.3 GNN MODELS

We consider the following unsupervised graph embedding algorithms, which are all GNN-based methods: GAE Kipf & Welling (2016b), VGAE Kipf & Welling (2016b), ARGA Pan et al. (2018), ARGVA Pan et al. (2018), GraphSAGE Hamilton et al. (2017a), GIN Xu et al. (2018), GAT Velickovic et al. (2017). **Note that, to maintain consistency in the training strategy and simplicity in the experimental framework, GAT, GIN, and GraphSAGE are all trained within the GAE training framework.**

These models were chosen because they have demonstrated strong performance in recent research and represent a diverse set of methodologies in graph representation learning. GAE and VGAE are foundational models in graph autoencoding, while ARGA and ARGVA introduce adversarial

training for enhanced robustness. GraphSAGE is known for its inductive learning capability, making it suitable for dynamic graphs. GIN provides a strong theoretical foundation for distinguishing graph structures, and GAT incorporates attention mechanisms to focus on relevant graph parts. This diversity allows us to test the generalizability of our Internal Strategy (IS) across different types of GNNs.

## A.4 BASELINE INTERNAL EVALUATION STRATEGY

Additionally, in other fields such as computer vision, singular values are also used to evaluate the quality of embeddings, including Incoherence (Tsitsulin et al., 2023a), Self Cluster (Tsitsulin et al., 2023a), $\alpha$-ReQ (Assran et al., 2022), RankMe (Garrido et al., 2023; Roy & Vetterli, 2007), NESum (He & Ozay, 2022), Condition Number (Tsitsulin et al., 2023a; Ben-Israel, 1966) and Stable Rank (Tsitsulin et al., 2023a) (For details on these algorithms, see Appendix C). Among these methods, RankMe has demonstrated the best performance when considering running time, robustness, and effectiveness, making it the baseline for our experiments.

## A.5 CANDIDATE HYPERPARAMETERS (SEARCH SPACE) AND SEARCH STRATEGY

Table 3: Candidate Hyperparameters for GNN Models

| Hyperparameter | Values |
|---|---|
| Num of neurons (Hidden Layer 1) | {8, 16, 24, 32, 40, 48, 56, 64, 72, 80, 88, 96, 104, 112, 120, 128} |
| Num of neurons (Hidden Layer 2) | {8, 16, 24, 32, 40, 48, 56, 64, 72, 80, 88, 96, 104, 112, 120, 128} |
| Num of epochs | {100, 150, 200, 250, 300} |

In our experiments, we use a consistent set of candidate HPs across all seven GNN models. This includes the network structure and the number of epochs, with details shown in Table 3. The search strategy we employ is **grid search**, which involves trying all possible HP settings within the search space.

The selection of these HP values is motivated by considerations of practicality and robustness:

- **Number of Layers**: In practical applications of GNNs, it is uncommon to use very deep networks. A two-layer GNN is often sufficient to capture necessary information, while deeper networks can suffer from the oversmoothing problem (Shi et al., 2022), where node features become indistinguishable. Thus, we chose 2 layers as it strikes a balance between performance and computational efficiency.

- **Hidden Units per Layer**: Pre-experimentation indicated that configurations with 32 or 64 hidden units per layer often yield the best performance. However, to ensure robustness and to verify if these configurations can handle extreme situations, we explored a wide range of values from 8 to 128 hidden units. This range allows us to validate the effectiveness of the IS in identifying appropriate hyperparameters by detecting performance degradation caused by extreme values.

- **Epochs**: The number of training epochs is a critical factor for model convergence. We included 100 epochs to represent insufficient training and 300 epochs to represent well-trained models. Intermediate values (150, 200, 250) were chosen to observe the progression of model performance with increasing training time and to identify the optimal number of epochs for each GNN model.

## A.6 EVALUATION METRICS

To evaluate the performance of algorithms, we consider two aspects:

- **Correlation Coefficient**: Firstly, the **correlation coefficient** measures the relationship between the rankings produced by the algorithm and the performance of the embeddings in downstream tasks. This metric is essential to determine if the IS method can reliably rank

the embeddings in a manner that reflects their true performance. A high correlation coefficient indicates that the IS method can effectively distinguish between high and low-quality embeddings, providing confidence in its use for hyperparameter tuning. Specifically, we use the Spearman correlation coefficient, which is calculated as follows:

$$\rho = 1 - \frac{6 \sum d_i^2}{n(n^2 - 1)}$$

where $d_i$ is the difference between the ranks of each pair of observations, and $n$ is the number of observations. Here, the Spearman correlation coefficient assesses the correlation between the ranking scores assigned by the IS method and the actual performance values obtained in downstream tasks.

- **Performance (AUC for link prediction, Accuracy for node classification) of the selected embedding (with the highest ranking score)**: This metric evaluates the IS's ability to identify the best-performing embeddings across two key downstream tasks, link prediction and node classification.

  For node classification, Accuracy is defined as:

  $$\text{Accuracy} = \frac{TP + TN}{TP + TN + FP + FN}$$

  where $TP$ is the number of true positives, $TN$ is the number of true negatives, $FP$ is the number of false positives, and $FN$ is the number of false negatives. Higher Accuracy values indicate better performance in correctly classifying nodes into their respective classes.

  For link prediction, the AUC (Area Under the Curve) is defined as:

  $$\text{AUC} = \frac{1}{N} \sum_{i=1}^{N} \left( \frac{\text{rank}(S_{i+}) - \text{rank}(S_{i-})}{|\text{rank}(S_{i+}) - \text{rank}(S_{i-})|} + 1 \right)$$

  where $S_{i+}$ represents the score of a correctly predicted link, $S_{i-}$ represents the score of an incorrectly predicted link, and $N$ is the total number of comparisons. Higher AUC values indicate better performance in predicting the existence of links between nodes.

The meanings of these two metrics are different. The **Correlation Coefficient** indicates if the IS is really correlated to the performance of the models, while the **Performance of the selected embedding** is more important for practical applications where models are deployed.

Due to the unsupervised nature of the GNN models we are using, it is crucial to choose evaluation metrics that do not bias towards any specific properties of the dataset. GNNs learn representations that are not tailored to any particular downstream task, and our IS is designed to select node embeddings that perform well generally, not just for specific tasks. To this end, using both node classification and link prediction allows us to better assess the overall quality of node embeddings and ensure the selected hyperparameters can achieve good performance across different downstream tasks.

# B  SENSITIVITY ANALYSIS

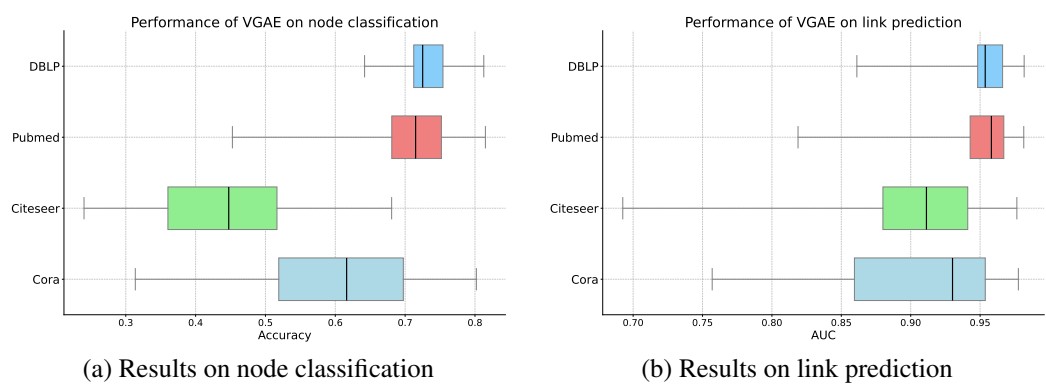

(a) Results on node classification                    (b) Results on link prediction

Figure 4: These figures show the performance differences of seven GNN models (VGAE, GAE, ARGA, etc.) across four datasets (Cora, Citeseer, Pubmed, and DBLP) under different hyperparameters (as listed in Table 3). Each model is evaluated on both node classification (Accuracy) and link prediction (AUC) tasks, with the y-axis representing datasets and the x-axis showing performance metrics. This figure specifically highlights the performance of the VGAE model. We can observe that the performance of the VGAE model on the node classification task varies significantly across different hyperparameters. On the Cora dataset, the performance gap between good and poor hyperparameters can be as large as 0.5. While the difference in performance is less extreme in the link prediction task, it is still substantial, with the AUC value on the Citeseer dataset differing by up to 0.3. This indicates that the VGAE model is highly sensitive to hyperparameter settings.

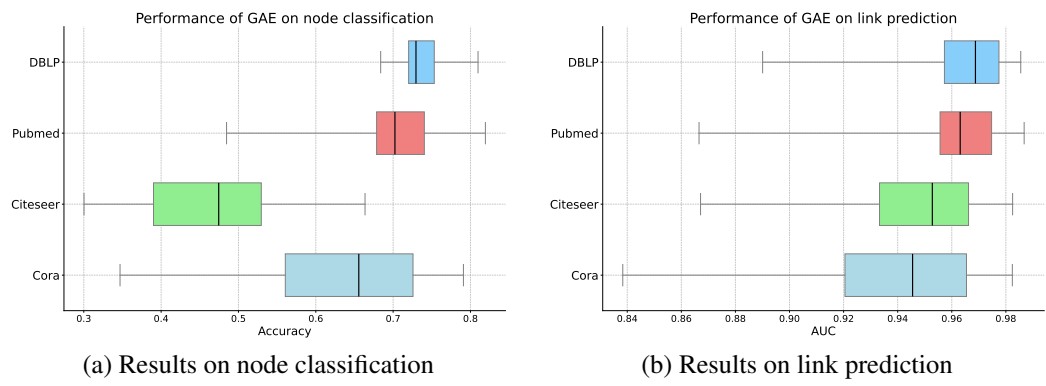

(a) Results on node classification                    (b) Results on link prediction

Figure 5: These figures show the performance differences of seven GNN models (VGAE, GAE, ARGA, etc.) across four datasets (Cora, Citeseer, Pubmed, and DBLP) under different hyperparameters (as listed in Table 3). Each model is evaluated on both node classification (Accuracy) and link prediction (AUC) tasks, with the y-axis representing datasets and the x-axis showing performance metrics. This figure specifically highlights the performance of the GAE model. We can observe that the performance of the GAE model on the node classification task shows significant variation across different hyperparameters. In comparison, the variation is less pronounced in the link prediction task, although GAE tends to achieve better results in link prediction. Overall, changes in hyperparameters have a substantial impact on the performance of the GAE model.

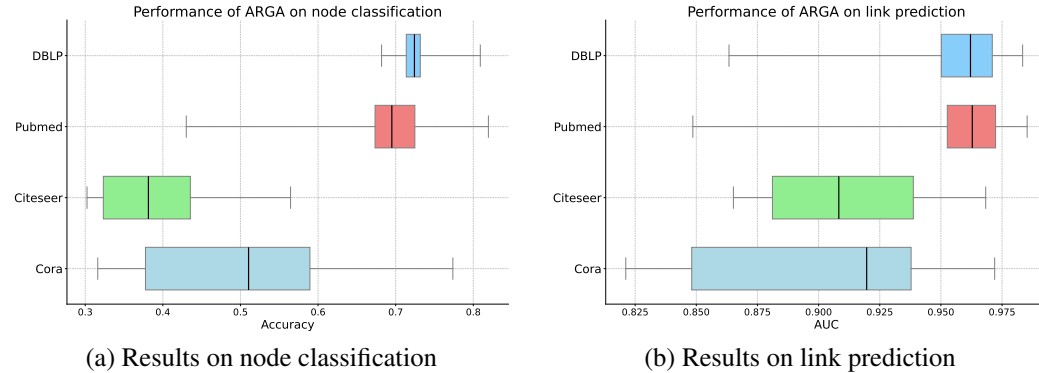

(a) Results on node classification        (b) Results on link prediction

Figure 6: These figures show the performance differences of seven GNN models (VGAE, GAE, ARGA, etc.) across four datasets (Cora, Citeseer, Pubmed, and DBLP) under different hyperparameters (as listed in Table 3). Each model is evaluated on both node classification (Accuracy) and link prediction (AUC) tasks, with the y-axis representing datasets and the x-axis showing performance metrics. This figure specifically highlights the performance of the ARGA model. We can observe that the performance of the ARGA model varies significantly across different hyperparameters on the node classification task. Its overall performance on the Citeseer dataset is quite poor, with even the best hyperparameter configuration failing to reach 0.6, which could pose challenges for hyperparameter optimization (HPO). In the link prediction task, the ARGA model performs well overall but still shows sensitivity to hyperparameters, with a performance range of nearly 0.2 between the best and worst configurations.

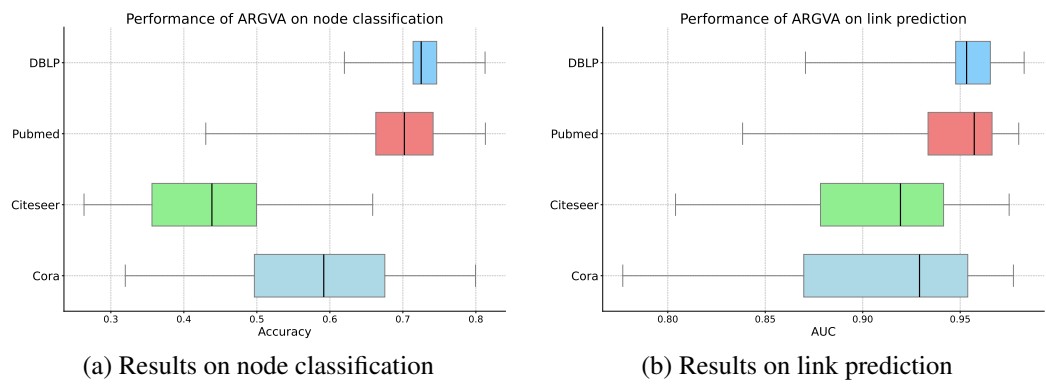

(a) Results on node classification        (b) Results on link prediction

Figure 7: These figures show the performance differences of seven GNN models (VGAE, GAE, ARGA, etc.) across four datasets (Cora, Citeseer, Pubmed, and DBLP) under different hyperparameters (as listed in Table 3). Each model is evaluated on both node classification (Accuracy) and link prediction (AUC) tasks, with the y-axis representing datasets and the x-axis showing performance metrics. This figure specifically highlights the performance of the ARGVA model. We can observe that the performance of the ARGVA model varies significantly across different hyperparameters on the node classification task. Similar to ARGA, its overall performance on the Citeseer dataset is quite poor, with the best hyperparameter configuration failing to reach 0.7, which could present challenges for hyperparameter optimization (HPO). In the link prediction task, the ARGVA model performs well overall but remains sensitive to hyperparameters, with a performance range of nearly 0.3 between the best and worst configurations.

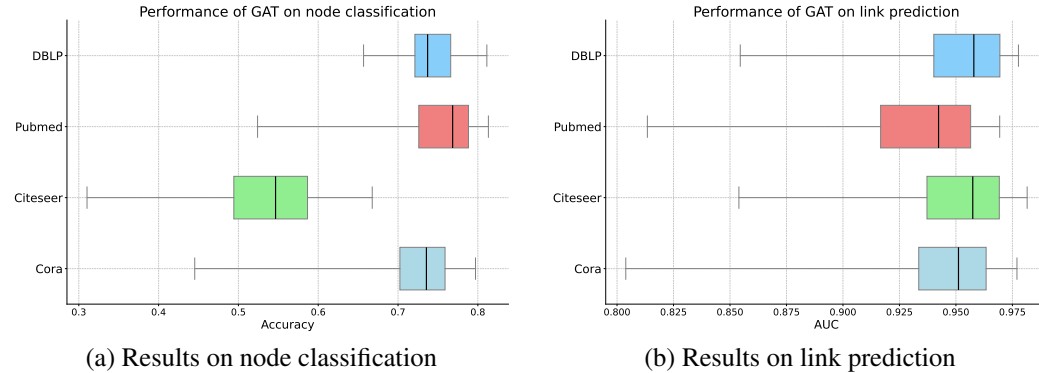

(a) Results on node classification   (b) Results on link prediction

Figure 8: These figures show the performance differences of seven GNN models (VGAE, GAE, ARGA, etc.) across four datasets (Cora, Citeseer, Pubmed, and DBLP) under different hyperparameters (as listed in Table 3). Each model is evaluated on both node classification (Accuracy) and link prediction (AUC) tasks, with the y-axis representing datasets and the x-axis showing performance metrics. This figure specifically highlights the performance of the GAT model. We can observe that the performance of the GAT model varies significantly across different hyperparameters in the node classification task. However, compared to the previous three models, the GAT model appears less sensitive to hyperparameters on the DBLP dataset. In the link prediction task, the model performs well overall but still shows sensitivity to hyperparameters, with a performance range of 0.2 between the best and worst configurations.

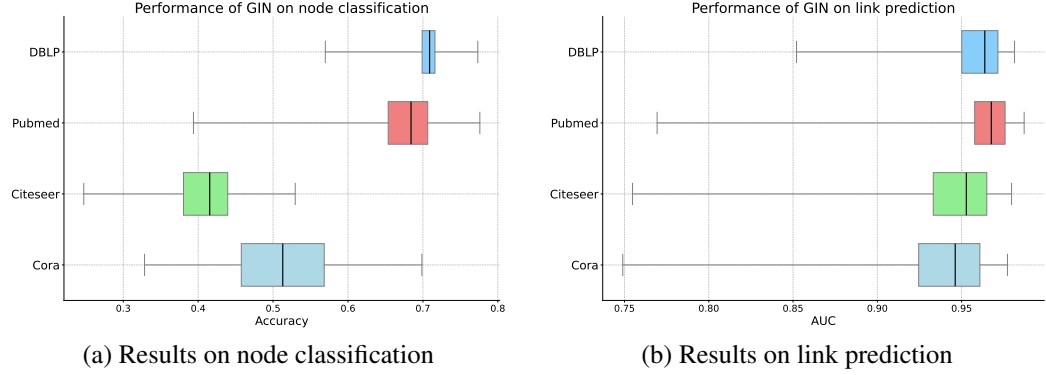

(a) Results on node classification   (b) Results on link prediction

Figure 9: These figures show the performance differences of seven GNN models (VGAE, GAE, ARGA, etc.) across four datasets (Cora, Citeseer, Pubmed, and DBLP) under different hyperparameters (as listed in Table 3). Each model is evaluated on both node classification (Accuracy) and link prediction (AUC) tasks, with the y-axis representing datasets and the x-axis showing performance metrics. This figure specifically highlights the performance of the GIN model. We can observe that the performance of the GIN model varies significantly across different hyperparameters in the node classification task, with particularly poor overall performance on the Citeseer dataset, only slightly above 0.5. In the link prediction task, most candidate hyperparameters perform well, which presents a challenge for HPO, as it requires selecting the best configuration from a set where the majority already show strong performance.

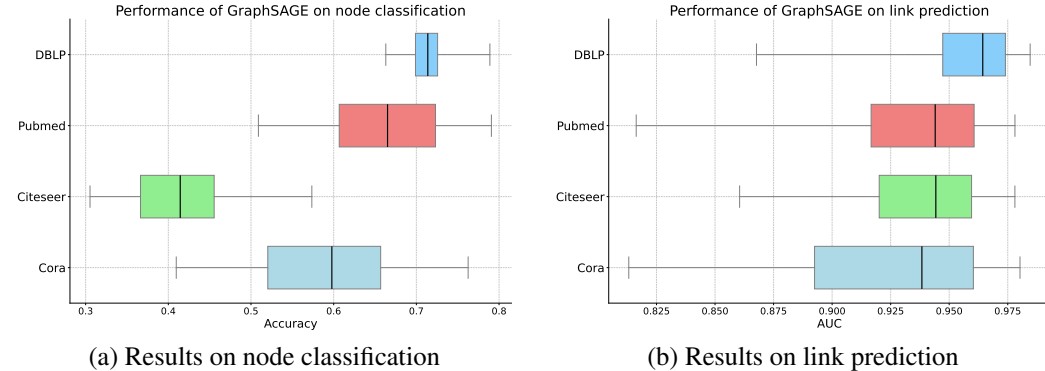

(a) Results on node classification                    (b) Results on link prediction

Figure 10: These figures show the performance differences of seven GNN models (VGAE, GAE, ARGA, etc.) across four datasets (Cora, Citeseer, Pubmed, and DBLP) under different hyperparameters (as listed in Table 3). Each model is evaluated on both node classification (Accuracy) and link prediction (AUC) tasks, with the y-axis representing datasets and the x-axis showing performance metrics. This figure specifically highlights the performance of the GraphSAGE model. We can observe that the performance of the GraphSAGE model varies significantly across different hyperparameters in the node classification task, with notably poor overall performance on the Citeseer dataset, where it falls below 0.6. Meanwhile, the differences are less pronounced on the DBLP dataset. In the link prediction task, GraphSAGE performs well overall, but remains highly sensitive to hyperparameters. Effective HPO can significantly enhance the model's performance in link prediction.

# C    REVISITING EXISTING INTERNAL STRATEGIES UNDER A UNIFIED FRAMEWORK

## C.1    INCOHERENCE

Incoherence (Tsitsulin et al., 2023a) shifted its approach by not attempting to hypothesize an **ideal distribution.** Instead, it considers the initial distribution as the **worst distribution,** a concept similar to our understanding in CSOR. However, the difference lies in its choice of the **initial distribution** as the standard basis, which are the basis vectors for each dimension. For example, when the dimensionality of the embedding is 3, the basis vectors are (1,0,0), (0,1,0), and (0,0,1). The quality is evaluated by calculating the degree of alignment between these basis vectors and the singular vectors. Overall, the lower the similarity to the basis vectors, the higher the quality of the embedding is deemed to be.

$$\text{Incoherence\_score}_{(h)} = \begin{cases} \frac{1}{\sum |\mathbf{V}^T \mathbf{I}_d|} & \text{if } \sum \left| \mathbf{V}^T \mathbf{I}_d \right| > 0 \\ \infty & \text{otherwise} \end{cases}$$

**Build prior belief:** The prior belief is that there is an ideal distribution of embeddings, and the closer an embedding is to this ideal distribution, the higher its quality.

**Quantify prior belief:** Since the unsupervised setting lacks labels, we assume that an embedding composed of basis vectors is the worst. The quality of other embeddings is measured by comparing their similarity to the basis vector embedding. The higher the similarity, the worse the quality, and vice versa.

## C.2    SELF CLUSTER

Self Cluster (Tsitsulin et al., 2023a) assess the quality of embeddings based on their clustering characteristics in high-dimensional spaces. The prior belief of Self Cluster is that embeddings with better structural quality and information richness are indicated by more effective clustering along various dimensions.

### C.2.1 MEASURING CLUSTERING IN EMBEDDINGS

To evaluate the clustering tendency of embeddings, the pairwise dot products of the embedding vectors are considered. This approach provides insights into how closely related or clustered the vectors are in the embedding space. A higher aggregation of dot products indicates stronger clustering, suggesting that the embeddings effectively capture meaningful relationships and structures within the data.

### C.2.2 ISOTROPIC RANDOM VECTORS

The concept of isotropic random vectors serves as a theoretical benchmark for comparison. In an ideal scenario where vectors are isotropic and uniformly distributed over a high-dimensional sphere, the embeddings would exhibit minimal bias towards any specific direction, resulting in a uniform spread. This distribution acts as the "prior belief" against which actual embedding distributions are measured. The expected dot product of such high-dimensional isotropic random vectors is typically very low, approaching zero as the dimensionality increases, except when vectors are identical.

### C.2.3 COMPONENTS OF THE SELF CLUSTER FORMULA

The Self Cluster metric incorporates three key components in its computation:

1. **Dot Product Matrix Norm** $Q = \|WW^T\|_F$: This term measures the sum of squared pairwise dot products among all embedding vectors, quantifying the overall similarity and potential clustering within the dataset. A higher norm suggests more pronounced clustering.

2. **Expected Dot Product for Isotropic Vectors** $E[Q] = n + \frac{n(n-1)}{2d}$: This component calculates what the norm of the dot product matrix would be if the embeddings were isotropic random vectors, providing a baseline for comparison. It helps determine if the actual embeddings are more clustered than would be expected by chance.

3. **Normalization by** $n^2$: In the extreme case of dimension collapse, where all vectors become identical, the dot product matrix turns into a matrix of ones, and its Frobenius norm reaches its maximum possible value of $n$. To ensure the Self Cluster metric is bounded between 0 and 1, the norm of the dot product matrix is normalized by $n^2$, which is the square of the number of embeddings. This normalization makes the metric robust to the number of embeddings and their dimensionality, facilitating comparisons across different datasets or models.

The Self Cluster metric effectively evaluates the clustering of embeddings by comparing the observed clustering level to that of a theoretical model of isotropic randomness. By understanding the deviations from this model, we can infer the degree of structure and the quality of the embeddings. A higher Self Cluster value indicates that the embeddings are significantly more clustered than expected under the isotropic model, suggesting richer structure and potentially higher quality embeddings for downstream tasks. This metric provides a quantitative tool to assess the ability of embedding algorithms to capture and preserve meaningful information in a high-dimensional space.

**Build prior belief:** The prior belief is that high-quality embeddings are more clustered than isotropic random vectors, which represent the worst-case distribution.

**Quantify prior belief:** The method computes the dot product matrix of the embeddings and compares it to the expected dot product of isotropic vectors. A higher clustering level, as indicated by the Frobenius norm of the dot product matrix, signals higher quality.

### C.3 NESUM

The principle of NESum (He & Ozay, 2022) is simple and straightforward. It involves normalizing the eigenvalues obtained from SVD and then summing them up directly. This sum is used as the ranking score $s(h)$:

The first step is to normalize the eigenvalues by the largest eigenvalue:

$$\sigma_i' = \frac{\sigma_i}{\sigma_{\max}}$$

where $\sigma_i$ is the $i$-th singular value from $\Sigma$, and $\sigma_{\max}$ is the largest singular value.

The second step is to sum all the normalized eigenvalues to obtain the NESum score:

$$Q_{NESum}(\mathbf{Z}(h)) = \sum_{i=1}^{r} \sigma_i'$$

**Build prior belief:** The prior belief is that the quality of embeddings can be reflected by the spread of their singular values, with more evenly distributed singular values indicating higher quality.

**Quantify prior belief:** NESum normalizes the singular values of the embedding matrix and sums them. A higher sum indicates higher embedding quality, as it reflects a more uniform distribution of the embeddings.

### C.4 RANKME

The normalization method used in RankMe (Garrido et al., 2023) involves using the sum of all eigenvalues as the denominator. Then, the entropy of the normalized singular values is calculated. The steps are as follows:

1. Normalize the eigenvalues by the sum of all eigenvalues:

$$\sigma_i' = \frac{\sigma_i}{\sum_j \sigma_j}$$

2. Calculate the entropy of the normalized singular values and take it as the ranking score:

$$Q_{RankMe}(\mathbf{Z}(h)) = -\sum_i \sigma_i' \log(\sigma_i')$$

The evaluation criterion of RankMe applies information entropy to measure uncertainty, which indicates the amount of information that can be carried. This means that the more space the embedding occupies in a multi-dimensional space and the more evenly it is distributed across dimensions, the greater the amount of information it carries, thus being considered of better quality.

**Build prior belief:** The prior belief is that high-quality embeddings exhibit more uniform distribution across dimensions, which reduces uncertainty and increases information content.

**Quantify prior belief:** RankMe calculates the entropy of the normalized singular values of the embedding matrix. Lower entropy suggests that the embeddings are more uniformly spread, indicating higher quality.

### C.5 STABLE RANK

Stable rank (Tsitsulin et al., 2023a) is a measure used to evaluate the quality of embeddings by considering the distribution of singular values. The stable rank is defined as the squared Frobenius norm of the matrix divided by the squared largest singular value. This metric provides insight into the effective dimensionality of the embedding space.

Given an embedding matrix $M$, we can calculate the stable rank as follows:

1. Compute the singular values $\sigma_i$ of the embedding matrix $M$.

2. Calculate the squared Frobenius norm of the matrix, which is the sum of the squares of all singular values:

$$\|M\|_F^2 = \sum_i \sigma_i^2$$

3. Identify the largest singular value $\sigma_{\max}$ and calculate its square:

$$\sigma_{\max}^2$$

4. Compute the stable rank:

$$\text{StableRank} = \frac{\|M\|_F^2}{\sigma_{\max}^2}$$

Stable Rank is essentially the same as RankMe in that both measure the extent and uniformity of the graph embedding distribution across different dimensions. However, Stable Rank does not use entropy; instead, it directly uses the largest singular value $\sigma_{\max}$ as the denominator. This means that for a constant sum of singular values $\|M\|_F^2$, a smaller $\sigma_{\max}$ is considered to indicate better quality embedding because a smaller $\sigma_{\max}$ represents a more uniform distribution.

**Build prior belief:** The prior belief is that a higher stable rank reflects a more uniform distribution of embeddings across dimensions, which is indicative of higher quality.

**Quantify prior belief:** StableRank computes the ratio of the squared Frobenius norm of the embedding matrix to the square of its largest singular value. A higher stable rank indicates a more evenly distributed embedding, suggesting higher quality.

## C.6 $\alpha$-REQ

The $\alpha$-ReQ (Assran et al., 2022) algorithm essentially considers the power-law distribution as the "ideal distribution." Therefore, evaluating the graph embedding quality is transformed into assessing the similarity between the distribution of graph embeddings in the multi-dimensional space and the power-law distribution. The key lies in projecting both distributions into a comparable space. The specific steps are as follows:

1. **Power-law Distribution Characteristics**: - The power-law distribution has the property that it becomes linear when subjected to a logarithmic transformation. Mathematically, a power-law distribution can be expressed as:

$$\lambda_i \propto i^{-\alpha}$$

Taking the logarithm of both sides, we get:

$$\log(\lambda_i) = -\alpha \log(i) + \log(C)$$

where $\lambda_i$ is the $i$-th eigenvalue, $\alpha$ is the power-law exponent, and $C$ is a constant.

2. **Log Transformation of Eigenvalues**: - Given the eigenvalues $\lambda_i$ obtained from the graph embedding's covariance matrix, we apply the logarithmic transformation to these eigenvalues:

$$\text{log\_eigenvalues} = \log(\lambda_i)$$

Additionally, we take the logarithm of their indices:

$$\text{log\_indices} = \log(i)$$

This transformation allows the power-law relationship to be represented as a linear relationship in the log-log space.

3. **Linear Regression to Estimate Alpha**: - By performing linear regression on the transformed singular valuess and their indices, we can estimate the decay coefficient $\alpha$. The linear regression model can be expressed as:

$$\log(\lambda_i) = \beta_1 \log(i) + \beta_0$$

where $\beta_1$ is the regression slope and $\beta_0$ is the intercept. The power-law exponent $\alpha$ is the negative of the slope:

$$\alpha = -\beta_1$$

Thus, the similarity between the graph embedding distribution and the power-law distribution is quantified by the estimated $\alpha$.

In summary, the Alpha-ReQ algorithm projects both the graph embedding distribution and the power-law distribution into a log-log space where they can be directly compared. By estimating the slope of the transformed singular values, the algorithm quantifies how closely the graph embedding follows the ideal power-law distribution.

**Build prior belief:** The prior belief is that the ideal distribution of embeddings follows a power-law distribution, with a specific decay pattern.

**Quantify prior belief:** $\alpha$-ReQ estimates the similarity between the embedding distribution and the power-law distribution by performing linear regression on the log-transformed singular values. The closer the decay pattern matches the power-law, the higher the quality of the embeddings.

### C.7    CONDITION NUMBER

The **ideal distribution** in the prior belief of Condition Number Ben-Israel (1966); Tsitsulin et al. (2023a) is that the high-quality embedding is evenly distributed across multiple dimensions. The metric it uses to evaluate the degree of distribution is the condition number, denoted as $k_2$, which is the ratio of the largest singular value to the smallest singular value.

Given an embedding matrix $M$, we can calculate the condition number $k_2$ as follows:

1. Identify the largest singular value $\sigma_{\max}$ and the smallest singular value $\sigma_{\min}$ from the diagonal elements of $\Sigma$.

2. Compute the condition number:

$$k_2 = \frac{\sigma_{\max}}{\sigma_{\min}}$$

A smaller condition number $k_2$ indicates a more uniformly distributed embedding, which is considered to be of higher quality. In contrast, a larger condition number suggests that the embedding is unevenly distributed across dimensions, indicating lower quality.

**Build prior belief:** The prior belief is that a more uniformly distributed embedding across dimensions, indicated by a smaller condition number, represents higher quality.

**Quantify prior belief:** ConditionNumber is calculated as the ratio between the largest and smallest singular values. A smaller condition number indicates that the embeddings are more evenly distributed, suggesting higher quality.

## D    MORE EXPERIMENT RESULTS FOR HPO

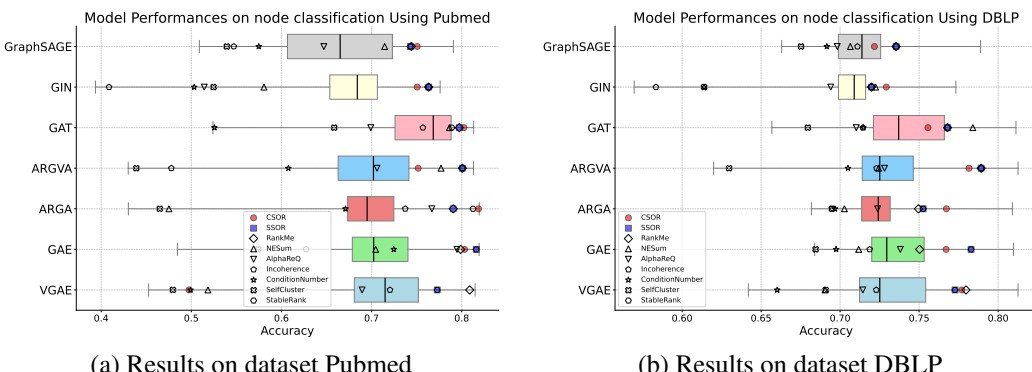

(a) Results on dataset Pubmed                    (b) Results on dataset DBLP

Figure 11: Node classification performance results on 7 GNN node embedding models on Pubmed and DBLP

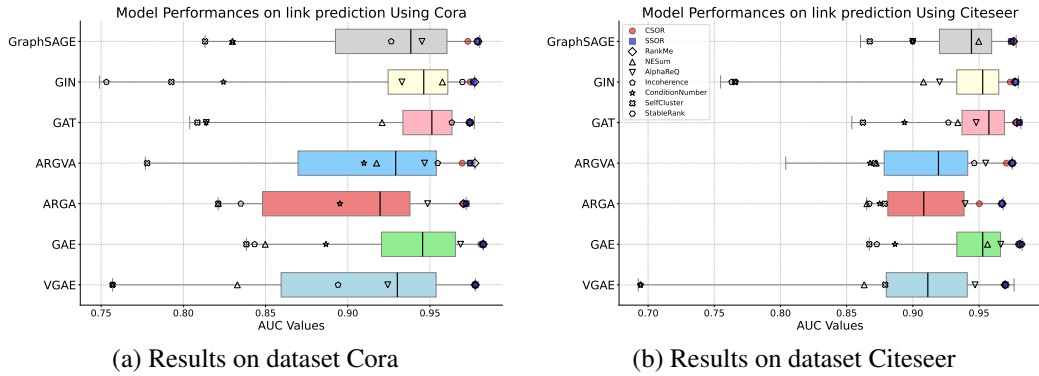

(a) Results on dataset Cora

(b) Results on dataset Citeseer

Figure 12: Link prediction performance results on 7 GNN node embedding models on Cora and Citeseer

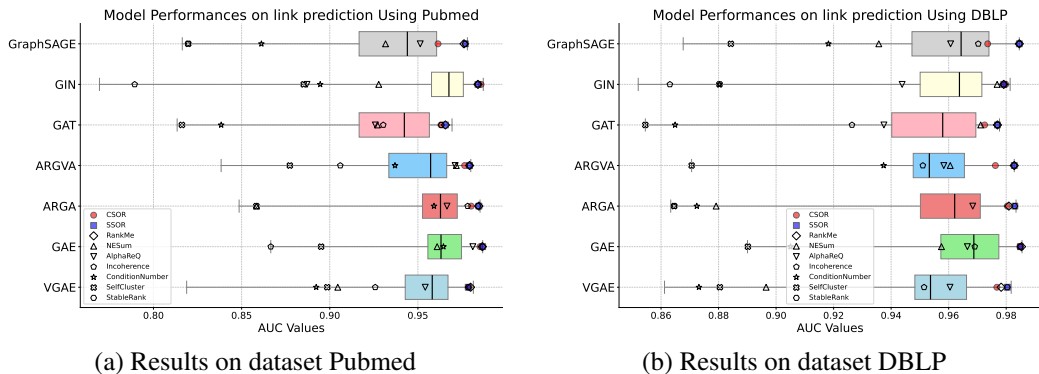

(a) Results on dataset Pubmed

(b) Results on dataset DBLP

Figure 13: Link prediction performance results on 7 GNN node embedding models on Pubmed and DBLP

# E PRELIMINARIES AND RELATED WORK

This paper, at a high conceptual level, attempts to apply Automated Machine Learning (AutoML) techniques to unsupervised learning. More specifically, it focuses on Neural Architecture Search (NAS), a subfield of AutoML, as applied to node embedding models within unsupervised Graph Representation Learning (GRL). However, in the unsupervised learning scenario, the absence of labels makes direct evaluation of the model difficult, and this is a necessary step in AutoML. To understand the context of this problem, the following sections will introduce some background knowledge related to AutoML and GRL. In the latter part of this section, we introduce some related work, including the general categorization of Internal Strategies (IS) into consensus-based and stand-alone approaches.

## E.1 AUTOMATED MACHINE LEARNING(AUTOML)

**Automated Machine Learning (AutoML).** AutoML (He et al., 2021) aims to reduce the need for manual effort to optimize model performance by automatically setting HP values (Melis et al., 2018; Snoek et al., 2012). This is particularly crucial for deep learning models, where the architecture of the neural network significantly impacts performance compared to traditional machine learning models. Simultaneously, with the boom in Graph Machine Learning, AutoML on Graphs (Zhang et al., 2021) has also garnered considerable attention.

Automatic hyperparameter optimization (HPO) encounters a main challenge: the computation is expensive. This challenge manifests itself in two ways: individual model evaluations can be very

costly when the training process requires substantial computing resources, and the candidate hyper-parameter space is vast, necessitating numerous trials to find the optimal combination. This issue is present in both supervised and unsupervised learning. However, if we want to extend AutoML techniques to unsupervised learning scenarios, we encounter an additional challenge: *evaluating the quality of the model output without labels*. **Extending AutoML techniques to unsupervised learning scenarios** is the main motivation and focus of this paper.

**Neural Architecture Search(NAS).** Neural networks have achieved breakthroughs in many fields, with recent focus primarily on computer vision (CV) (Krizhevsky et al., 2012; He et al., 2016; Dosovitskiy et al., 2021) and natural language processing (NLP) (Hochreiter & Schmidhuber, 1997; Vaswani et al., 2017). Many of these great works are due to the design of new neural architectures, but this largely relies on experts' understanding of specific domains. Neural Architecture Search (NAS) aims to automatically search for well-performing neural architectures to address varying application scenarios and different datasets.

Neural architecture is crucial to the performance of a deep learning model, and Neural Architecture Search (NAS), as a subfield of AutoML, has received much attention in recent years (Ren et al., 2021). In the context of NAS, there are three main components: search space, search strategy, and performance evaluation. The *search space* defines the candidate neural architectures, for example, by specifying the number of network layers and the number of neurons in each layer. *Search strategy* concerns how to explore the search space, which is about how to construct candidate neural architectures. The most classic approach is grid search, which looks for all possible neural architectures within the defined search space. *Performance evaluation* assesses how well these candidate neural architectures perform. In supervised representation learning, this usually involves evaluating the model-generated representations on specific downstream tasks (e.g., classification tasks) with labels. For unsupervised embedding, however, the challenge is to assess the quality of the embeddings without relying on labeled data, requiring alternative evaluation metrics.

### E.2 Graph Embedding

As one of the primary contexts for the problem studied in this paper, it is essential to understand graph embedding. *Graph embedding is a technique that converts graph data into low-dimensional real-numbered vectors*. Below, we introduce some key terms related to graph embedding and provide relevant background knowledge.

**Representation Learning.** Most machine learning tasks heavily rely on the quality of features built by experts, a process known as *feature engineering* (Guyon & Elisseeff, 2003). Consequently, the performance of models is highly dependent on the experts' domain knowledge of the target datasets. Representation learning (Bengio et al., 2013) can be viewed as an automated approach to feature engineering. It involves learning representations (i.e., features or embeddings) from datasets that can be utilized for specific machine learning tasks, such as classification or prediction.

**Graph Representation Learning (GRL) and Graph Embedding.** Please note that in current research, the terms *graph embedding* and *graph representation learning* are often used interchangeably. Therefore, they will not be distinguished in this paper and we will refer to both as *graph embedding*.

Graph Representation Learning (GRL) (Hamilton, 2020) is a specialized subset of representation learning where the input data is structured as graphs. It focuses on learning representations from graph data, capturing the relationships and structures inherent in graphs. GRL can be approached in three main ways: 1) traditional statistical methods based on graph theory; 2) node embedding methods based on random walk mechanisms; and 3) Graph Neural Networks (GNNs). Recently, GNNs have demonstrated dominant performance in the GRL field, making them the focus of this paper.

Graph Embedding (Cai et al., 2018), a technique within GRL, maps graph data into low-dimensional vectors of real numbers. This process primarily focuses on leveraging the structural information of the graph, and in the case of GNN-based methods, it also incorporates node feature information. Through graph embedding, the learned representations (embeddings) are optimized for various

downstream tasks such as node classification, link prediction, and clustering (Zhou et al., 2020; Wu et al., 2021; Zhang et al., 2020).

**Graph Embeddings at Different Levels.** Depending on specific application requirements (Cai et al., 2018), graph embeddings can be obtained at a node-level or graph-level. Node-level embedding involves generating representations for individual nodes, utilizing the structural information, node features, and/or edge weights of the graph. In contrast, graph-level embeddings are generated for the entire graph, summarizing its overall structure and properties. In this paper, all mentioned graph embeddings are at the node level. Therefore, the terms "node embedding" and "graph embedding" are used interchangeably and both refer to node embeddings.

**Node information aggregation.** Node information aggregation (Kipf & Welling, 2016a; Hamilton et al., 2017a; Veličković et al., 2018) is a key step in Graph Neural Networks (GNNs) that involves the message passing mechanism. In this process, each node collects information from its neighbors, aggregates this information using functions such as mean, sum, or max, and updates its own representation based on the aggregated information. This step is crucial for capturing the local graph structure and node features, and it is essential for understanding the relationships between spatial- and spectral-based GNNs discussed in section E.3.

**Different downstream tasks.** Node embeddings can be used for various downstream tasks, including node classification (Wang et al., 2017), node clustering (Nie et al., 2017), link prediction (Zhang & Chen, 2018), anomaly detection (Ma et al., 2021), etc. These downstream tasks represent practical applications of node representation learning and provide a means to assess the quality of learned node embeddings. While node classification is commonly used to evaluate embeddings, this paper employs link prediction as the downstream task because it provides a more direct measure of the embeddings' ability to capture the underlying graph structure. Link prediction is particularly useful for evaluating the quality of node embeddings in unsupervised settings, as it does not rely on node labels and focuses on the structural properties of the graph, aligning better with our focus on embedding quality.

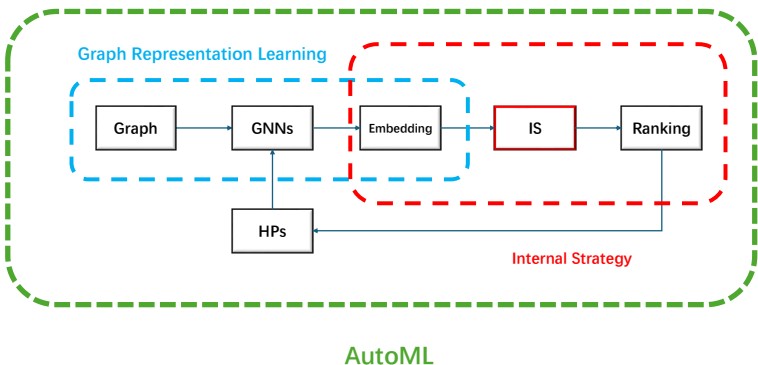

Figure 14: The figure illustrates the entire process of performing HPO tasks. AutoML is a broader concept than HPO, with its fundamental task being the HPO task. GRL is the process of converting a graph into graph embeddings using GRL models (in this paper, GNNs). IS is a process that takes embeddings as input, ranks the embeddings generated by all HP configurations, and outputs a ranking score. The intersection of IS and GRL is the embedding. IS is the focus of this paper, and this figure shows the relationships among AutoML, GRL, and IS.

### E.3 FROM SPATIAL-BASED GNNs TO SPECTRAL-BASED GNNs

Graph Neural Networks (GNNs) have developed along two primary routes: spatial-based and spectral-based approaches. These two methods fundamentally differ in how they aggregate node information during the embedding process.

Spatial-based (Hamilton et al., 2017a; Atwood & Towsley, 2016; Veličković et al., 2018) methods aggregate information directly from the neighboring nodes in the graph. They operate on the graph's structure by iteratively combining the features of a node with those of its neighbors. This process is intuitive and straightforward, as it directly reflects the graph's topology.

Spectral-based (Monti et al., 2017; Hamilton et al., 2017b; Zhang et al., 2018; Zhou et al., 2018; Wu et al., 2019) methods take a different approach by utilizing the graph's spectral properties. These methods rely on the eigenvalues and eigenvectors of graph-related matrices (such as the Laplacian matrix) to perform convolution operations in the frequency domain. Spectral methods transform the graph into a spectral space, apply filters to the eigenvalues of the graph Laplacian matrix, and then transform it back, effectively aggregating information across the entire graph in a way that is analogous to applying a global filter. This approach is mathematically elegant and leverages the powerful tools of spectral graph theory.

Chen et al. (2023) provide a comprehensive analysis of spatial-based and spectral-based approaches in GNNs, proposing a unified framework that links them. The authors demonstrate that both spatial and spectral methods aim to achieve similar goals—effective information aggregation and node representation—through different mechanisms. Spatial methods can be interpreted as a form of spectral filtering in the node domain by setting the step size of message passing, while spectral methods approximate the aggregation process by filtering in the frequency domain. This unified perspective reveals that the distinction between the two methods lies more in their implementation techniques rather than their fundamental objectives (more details can be found in Appendix I).

### E.4 INTERNAL STRATEGY (IS)

Internal Strategies refer to the specific type of evaluation algorithms that can assess the quality of embeddings without relying on any external evaluation methods. In this paper, **IS** specifically represents the type of algorithm we are introducing. The location of IS in the context of AutoML and GRL can be seen in Figure 14.

**Definition of Internal Strategy.** An Internal Strategy (IS) is an unsupervised model evaluation method that assesses model performance without using labels. More specifically, in this paper, IS takes graph representations as input and outputs a corresponding ranking score, representing the quality of the graph representations.

To avoid ambiguity, we clarify the target of IS: In this paper, IS directly works on graph embeddings. However, since each graph embedding is generated using specific HP values, it can also be said that IS evaluates the performance of these HP values. Furthermore, since hyperparameters are an essential part of the model, when we refer to the evaluation of GNN models, it pertains to the same concept.

The term **Internal** signifies that IS evaluates by leveraging information from within the models and data, rather than relying on external information (e.g. labels) or human intelligence. This domain remains largely unexplored due to the challenges posed by the absence of labels. Currently, there is no existing work specifically focused on evaluating the quality of node embeddings, except for a meta-learning approach (more details in Appendix J) and some internal strategies (IS) from the computer vision (CV) community (Appendix C).

**Stand-alone and Consensus-based IS.** However, we can draw some ideas from other fields such as Computer Vision (Garrido et al., 2023; Duan et al., 2020) and Anomaly Detection (Ma et al., 2023). Existing approaches in these fields fall into two categories: stand-alone and consensus-based. *Stand-alone* approaches evaluate a hyperparameter (HP) setting independently, whereas *consensus-based* approaches require information from multiple HP settings. Stand-alone methods only need **a single HP setting**, while consensus-based methods depend on **a pool of candidate HP settings** .

**Stand-alone Internal Strategy.** In Ma et al. (2023), seven Internal Strategy (IS) methods were evaluated for the unsupervised outlier model selection challenge, including four stand-alone and three consensus-based approaches. The findings indicated that none of the IS methods outperformed the leading iForest detector (Liu et al., 2008), but consensus-based methods showed more promise than stand-alone approaches.

**Consensus-based Internal Strategy.** In Duan et al. (2020), the Unsupervised Disentanglement Ranking (UDR) was introduced as a consensus-based method aimed at hyperparameter tuning for unsupervised disentangled representation learning models. UDR's objective is to identify HP values of models that offer the highest degree of disentanglement. Disentanglement (Siddharth et al., 2017) refers to the ability of a model to separate distinct, interpretable factors of variation in the data, such that each factor corresponds to a different dimension in the latent space. This means that changes in one latent variable should correspond to changes in only one aspect of the data, allowing for more interpretable and manipulable representations. This approach was evaluated on six leading Variational Autoencoder (VAE)-based models (Kingma & Welling, 2014; Rezende et al., 2014) for unsupervised disentangled representation learning. The findings demonstrated a correlation between UDR and four supervised disentanglement metrics, indicating its potential for identifying models with highly disentangled representations without the need for labeled data.

## F EXPERIMENTAL SETTINGS AND ADDITIONAL RESULTS OF THE VISUALIZATION EXPERIMENTS

This section provides a detailed description of all visualization experiments and hypothesis validation experiments. The HPs used in the visualization of the node embedding distribution experiments are shown in Table 3, and those used in the hypothesis validation experiments are shown in Table 4. Section F.1 and F.3 contains additional results of node embedding distribution visualizations. Section F.2 and F.4 present further experimental results demonstrating the quantification of distinctness from spatial and spectral perspective.

Table 4: HP configurations for validating hypothesis

| Hyperparameter | Values |
|---|---|
| Num of neurons (Hidden Layer 1) | {8, 16, 32, 48, 64} |
| Num of neurons (Hidden Layer 2) | {8, 16, 32, 48, 64} |
| Num of epochs | {100, 200, 300} |

### F.1 SPATIAL DISTRIBUTION FOR CSOR

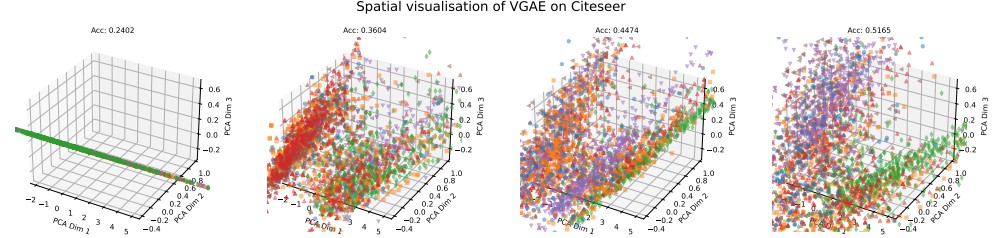

Figure 15: These four images are intended to demonstrate the relationship between the distribution characteristics of node embeddings and their performance on the downstream node classification task. The VGAE model is run on the Citeseer dataset, with candidate HP settings provided in Appendix 3. "Acc" at the top of the images represents Accuracy, where higher values indicate better performance. We can observe that as Acc increases, the node embeddings become more dispersed.

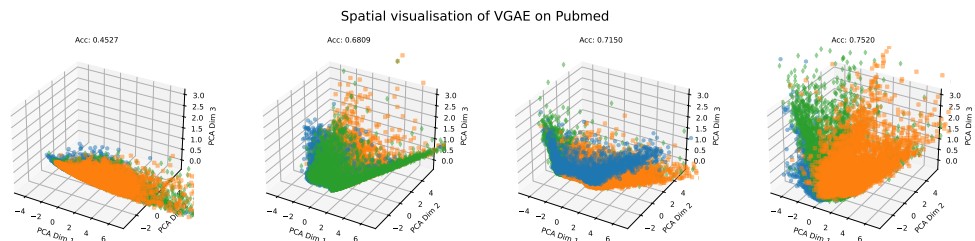

Figure 16: These four images are intended to demonstrate the relationship between the distribution characteristics of node embeddings and their performance on the downstream node classification task. The VGAE model is run on the Pubmed dataset, with candidate hyperparameter settings provided in Appendix 3. "Acc" at the top of the images represents Accuracy, where higher values indicate better performance. We can observe that as Acc increases, the node embeddings appear more dispersed. However, due to the large number of nodes in this dataset, the visualization becomes cluttered. Despite this, based on the increasing Acc values and the patterns at the edges, we can infer that the node embeddings are likely becoming more dispersed, although they overlap on the 2D plane.

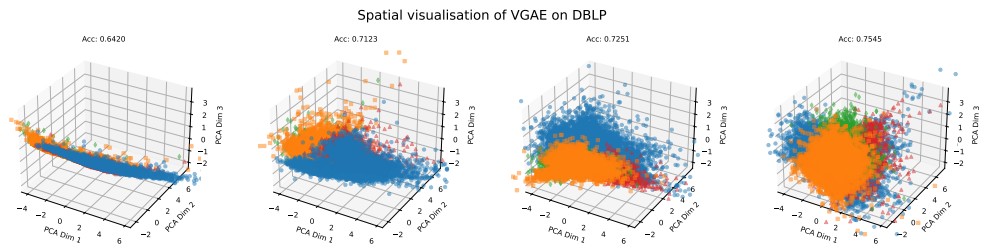

Figure 17: These four images are intended to demonstrate the relationship between the distribution characteristics of node embeddings and their performance on the downstream node classification task. The VGAE model is run on the DBLP dataset, with candidate hyperparameter settings provided in Appendix 3. "Acc" at the top of the images represents Accuracy, where higher values indicate better performance. We can observe that as Acc increases, the node embeddings appear more dispersed. However, due to the large number of nodes in this dataset, the visualization becomes cluttered. Despite this, based on the increasing Acc values and the patterns at the edges, we can infer that the node embeddings are likely becoming more dispersed, although they overlap on the 2D plane.

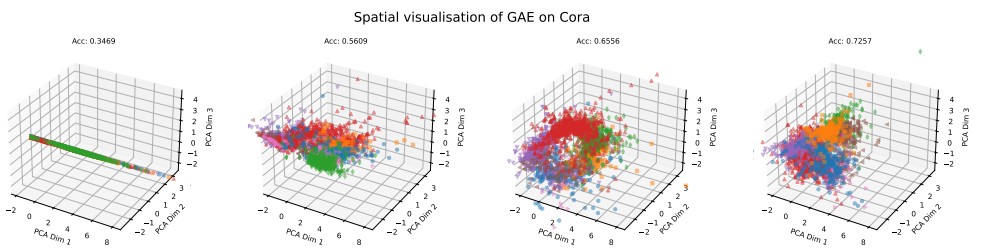

Figure 18: These four images are intended to demonstrate the relationship between the distribution characteristics of node embeddings and their performance on the downstream node classification task. The GAE model is run on the Cora dataset, with candidate hyperparameter settings provided in Appendix 3. "Acc" at the top of the images represents Accuracy, where higher values indicate better performance. We can observe that as Acc increases, the node embeddings become more dispersed.

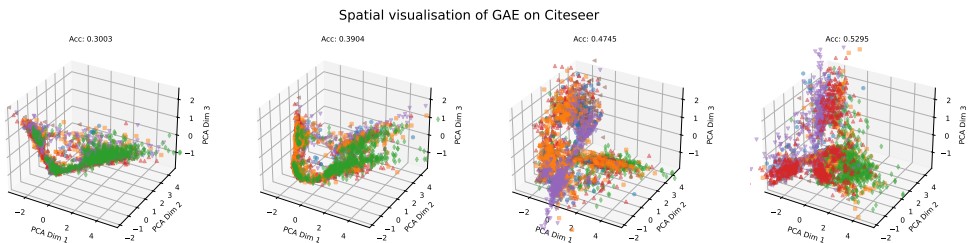

Figure 19: These four images are intended to demonstrate the relationship between the distribution characteristics of node embeddings and their performance on the downstream node classification task. The GAE model is run on the Citeseer dataset, with candidate hyperparameter settings provided in Appendix 3. "Acc" at the top of the images represents Accuracy, where higher values indicate better performance. We can observe that as Acc increases, the node embeddings become more dispersed.

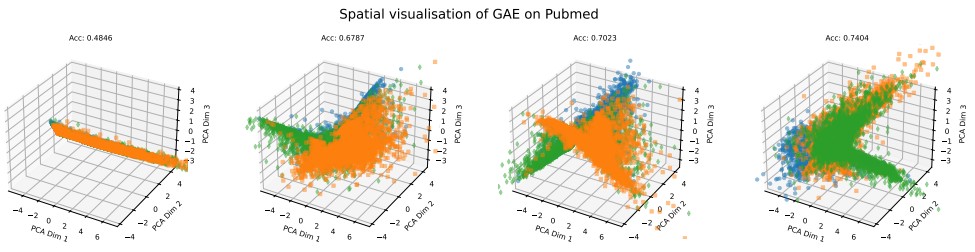

Figure 20: These four images are intended to demonstrate the relationship between the distribution characteristics of node embeddings and their performance on the downstream node classification task. The GAE model is run on the Pubmed dataset, with candidate hyperparameter settings provided in Appendix 3. "Acc" at the top of the images represents Accuracy, where higher values indicate better performance. We can observe that as Acc increases, the node embeddings appear more dispersed. However, due to the large number of nodes in this dataset, the visualization becomes cluttered. Despite this, based on the increasing Acc values and the patterns at the edges, we can infer that the node embeddings are likely becoming more dispersed, although they overlap on the 2D plane.

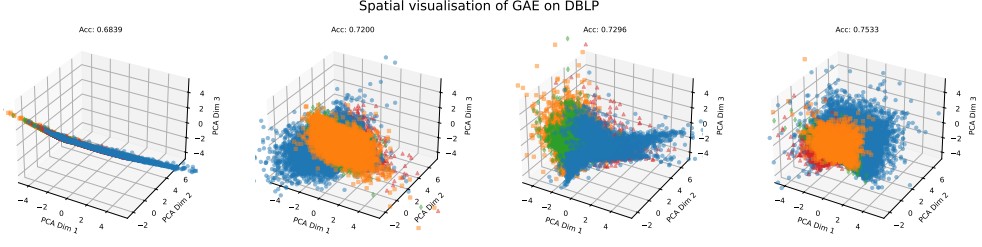

Figure 21: These four images are intended to demonstrate the relationship between the distribution characteristics of node embeddings and their performance on the downstream node classification task. The GAE model is run on the DBLP dataset, with candidate hyperparameter settings provided in Appendix 3. "Acc" at the top of the images represents Accuracy, where higher values indicate better performance. We can observe that as Acc increases, the node embeddings appear more dispersed. However, due to the large number of nodes in this dataset, the visualization becomes cluttered. Despite this, based on the increasing Acc values and the patterns at the edges, we can infer that the node embeddings are likely becoming more dispersed, although they overlap on the 2D plane.

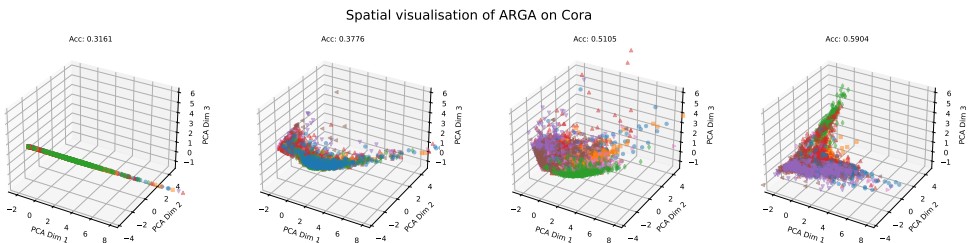

Figure 22: These four images are intended to demonstrate the relationship between the distribution characteristics of node embeddings and their performance on the downstream node classification task. The ARGA model is run on the Cora dataset, with candidate hyperparameter settings provided in Appendix 3. "Acc" at the top of the images represents Accuracy, where higher values indicate better performance. We can observe that as Acc increases, the node embeddings become more dispersed.

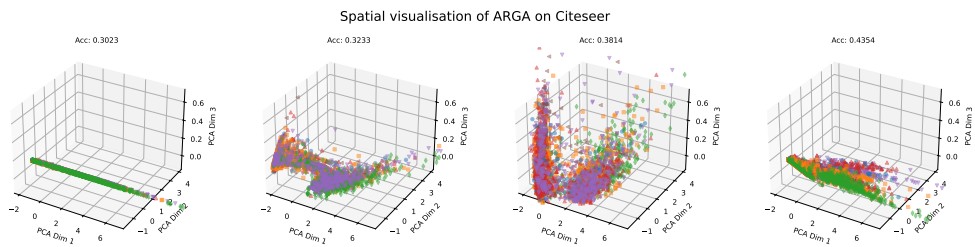

Figure 23: These four images are intended to demonstrate the relationship between the distribution characteristics of node embeddings and their performance on the downstream node classification task. The ARGA model is run on the Citeseer dataset, with candidate hyperparameter settings provided in Appendix 3. "Acc" at the top of the images represents Accuracy, where higher values indicate better performance. We can observe that as Acc increases, the node embeddings become more dispersed.

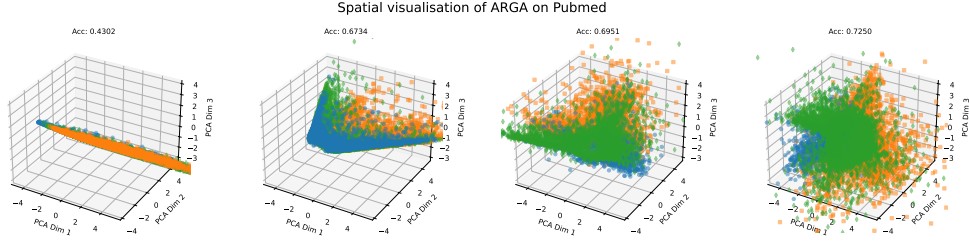

Figure 24: These four images are intended to demonstrate the relationship between the distribution characteristics of node embeddings and their performance on the downstream node classification task. The ARGA model is run on the Pubmed dataset, with candidate hyperparameter settings provided in Appendix 3. "Acc" at the top of the images represents Accuracy, where higher values indicate better performance. We can observe that as Acc increases, the node embeddings appear more dispersed. However, due to the large number of nodes in this dataset, the visualization becomes cluttered. Despite this, based on the increasing Acc values and the patterns at the edges, we can infer that the node embeddings are likely becoming more dispersed, although they overlap on the 2D plane.

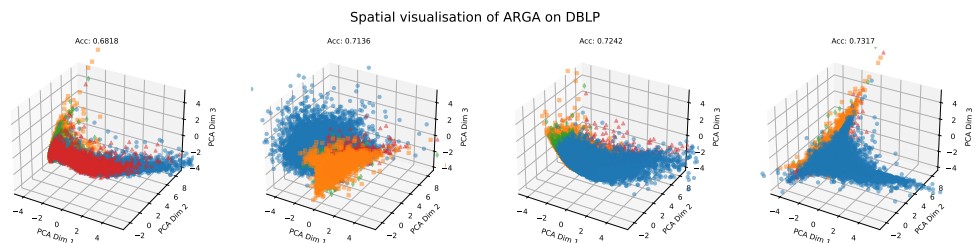

Figure 25: These four images are intended to demonstrate the relationship between the distribution characteristics of node embeddings and their performance on the downstream node classification task. The ARGA model is run on the DBLP dataset, with candidate hyperparameter settings provided in Appendix 3. "Acc" at the top of the images represents Accuracy, where higher values indicate better performance. We can observe that as Acc increases, the node embeddings appear more dispersed. However, due to the large number of nodes in this dataset, the visualization becomes cluttered. Despite this, based on the increasing Acc values and the patterns at the edges, we can infer that the node embeddings are likely becoming more dispersed, although they overlap on the 2D plane.

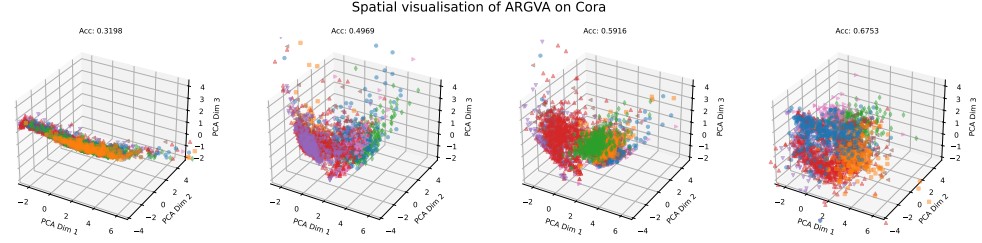

Figure 26: These four images are intended to demonstrate the relationship between the distribution characteristics of node embeddings and their performance on the downstream node classification task. The ARGVA model is run on the Cora dataset, with candidate hyperparameter settings provided in Appendix 3. "Acc" at the top of the images represents Accuracy, where higher values indicate better performance. We can observe that as Acc increases, the node embeddings become more dispersed.

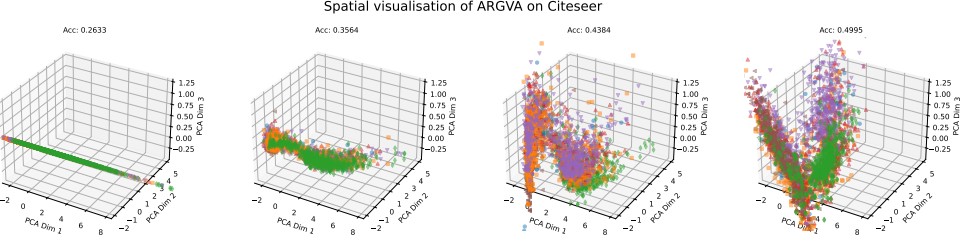

Figure 27: These four images are intended to demonstrate the relationship between the distribution characteristics of node embeddings and their performance on the downstream node classification task. The ARGVA model is run on the Citeseer dataset, with candidate hyperparameter settings provided in Appendix 3. "Acc" at the top of the images represents Accuracy, where higher values indicate better performance. We can observe that as Acc increases, the node embeddings become more dispersed.

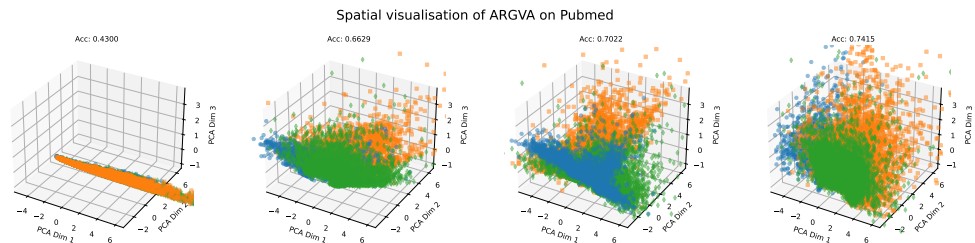

Figure 28: These four images are intended to demonstrate the relationship between the distribution characteristics of node embeddings and their performance on the downstream node classification task. The ARGVA model is run on the Pubmed dataset, with candidate hyperparameter settings provided in Appendix 3. "Acc" at the top of the images represents Accuracy, where higher values indicate better performance. We can observe that as Acc increases, the node embeddings appear more dispersed. However, due to the large number of nodes in this dataset, the visualization becomes cluttered. Despite this, based on the increasing Acc values and the patterns at the edges, we can infer that the node embeddings are likely becoming more dispersed, although they overlap on the 2D plane.

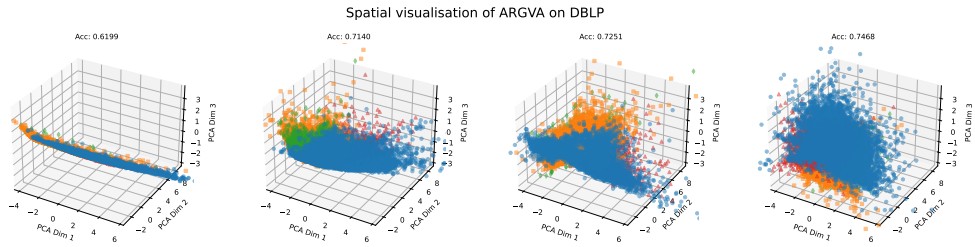

Figure 29: These four images are intended to demonstrate the relationship between the distribution characteristics of node embeddings and their performance on the downstream node classification task. The ARGVA model is run on the DBLP dataset, with candidate hyperparameter settings provided in Appendix 3. "Acc" at the top of the images represents Accuracy, where higher values indicate better performance. We can observe that as Acc increases, the node embeddings appear more dispersed. However, due to the large number of nodes in this dataset, the visualization becomes cluttered. Despite this, based on the increasing Acc values and the patterns at the edges, we can infer that the node embeddings are likely becoming more dispersed, although they overlap on the 2D plane.

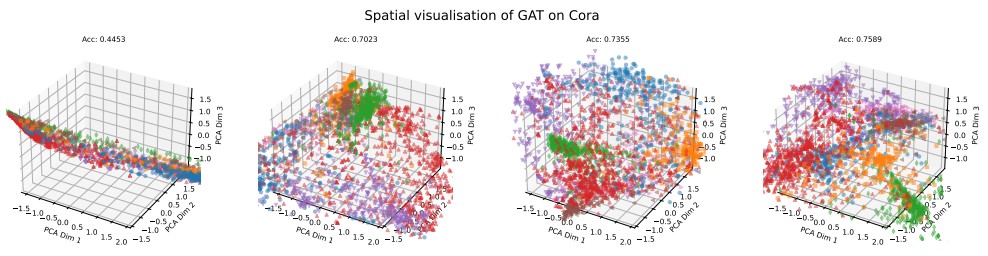

Figure 30: These four images are intended to demonstrate the relationship between the distribution characteristics of node embeddings and their performance on the downstream node classification task. The GAT model is run on the Cora dataset, with candidate hyperparameter settings provided in Appendix 3. "Acc" at the top of the images represents Accuracy, where higher values indicate better performance. We can observe that as Acc increases, the node embeddings become more dispersed.

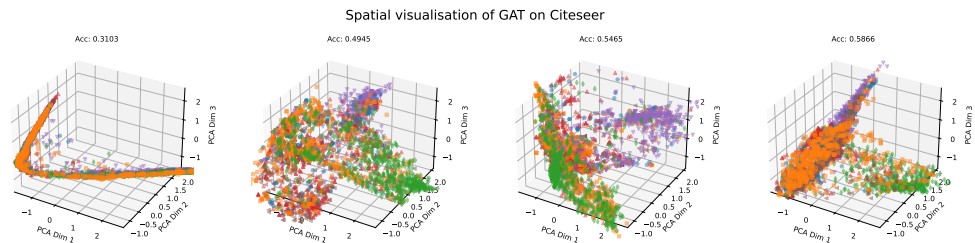

Figure 31: These four images are intended to demonstrate the relationship between the distribution characteristics of node embeddings and their performance on the downstream node classification task. The GAT model is run on the Citeseer dataset, with candidate hyperparameter settings provided in Appendix 3. "Acc" at the top of the images represents Accuracy, where higher values indicate better performance. We can observe that as Acc increases, the node embeddings become more dispersed.

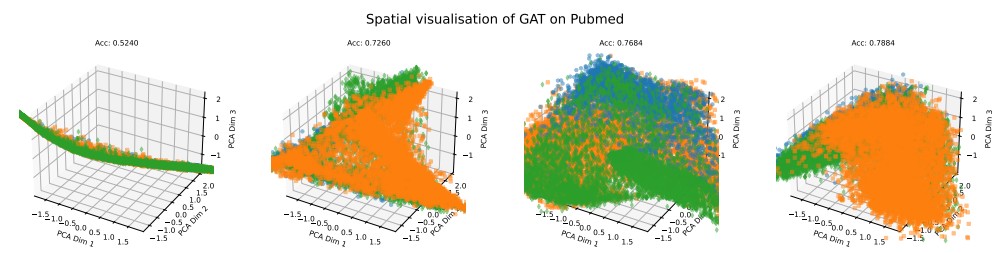

Figure 32: These four images are intended to demonstrate the relationship between the distribution characteristics of node embeddings and their performance on the downstream node classification task. The GAT model is run on the Pubmed dataset, with candidate hyperparameter settings provided in Appendix 3. "Acc" at the top of the images represents Accuracy, where higher values indicate better performance. We can observe that as Acc increases, the node embeddings appear more dispersed. However, due to the large number of nodes in this dataset, the visualization becomes cluttered. Despite this, based on the increasing Acc values and the patterns at the edges, we can infer that the node embeddings are likely becoming more dispersed, although they overlap on the 2D plane.

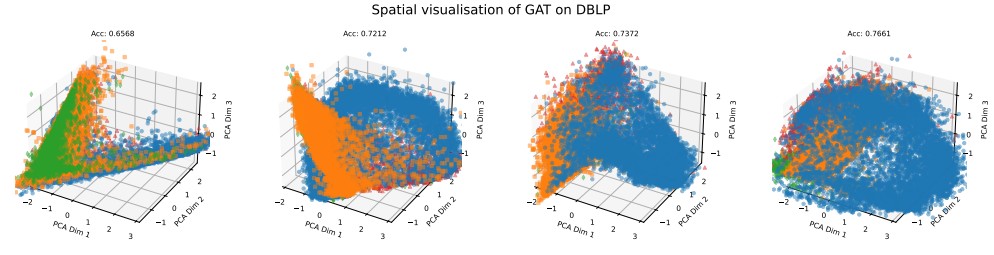

Figure 33: These four images are intended to demonstrate the relationship between the distribution characteristics of node embeddings and their performance on the downstream node classification task. The GAT model is run on the DBLP dataset, with candidate hyperparameter settings provided in Appendix 3. "Acc" at the top of the images represents Accuracy, where higher values indicate better performance. We can observe that as Acc increases, the node embeddings appear more dispersed. However, due to the large number of nodes in this dataset, the visualization becomes cluttered. Despite this, based on the increasing Acc values and the patterns at the edges, we can infer that the node embeddings are likely becoming more dispersed, although they overlap on the 2D plane.

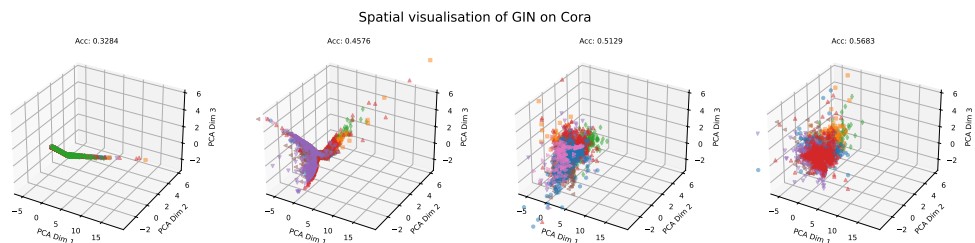

Figure 34: These four images are intended to demonstrate the relationship between the distribution characteristics of node embeddings and their performance on the downstream node classification task. The GIN model is run on the Cora dataset, with candidate hyperparameter settings provided in Appendix 3. "Acc" at the top of the images represents Accuracy, where higher values indicate better performance. We can observe that as Acc increases, the node embeddings become more dispersed.

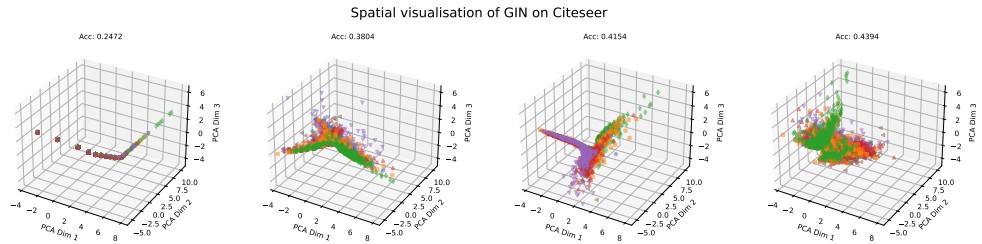

Figure 35: These four images are intended to demonstrate the relationship between the distribution characteristics of node embeddings and their performance on the downstream node classification task. The GIN model is run on the Citeseer dataset, with candidate hyperparameter settings provided in Appendix 3. "Acc" at the top of the images represents Accuracy, where higher values indicate better performance. We can observe that as Acc increases, the node embeddings become more dispersed.

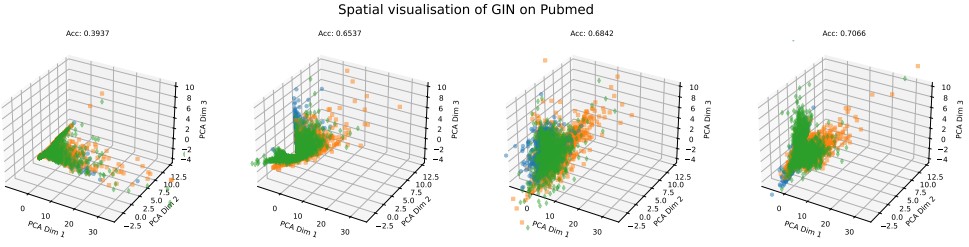

Figure 36: These four images are intended to demonstrate the relationship between the distribution characteristics of node embeddings and their performance on the downstream node classification task. The GIN model is run on the Pubmed dataset, with candidate hyperparameter settings provided in Appendix 3. "Acc" at the top of the images represents Accuracy, where higher values indicate better performance. We can observe that as Acc increases, the node embeddings appear more dispersed. However, due to the large number of nodes in this dataset, the visualization becomes cluttered. Despite this, based on the increasing Acc values and the patterns at the edges, we can infer that the node embeddings are likely becoming more dispersed, although they overlap on the 2D plane.

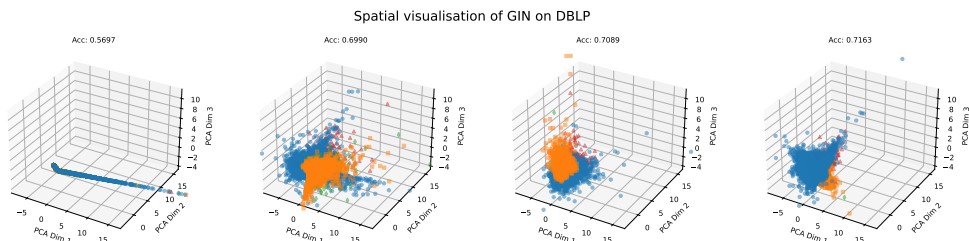

Figure 37: These four images are intended to demonstrate the relationship between the distribution characteristics of node embeddings and their performance on the downstream node classification task. The GIN model is run on the DBLP dataset, with candidate hyperparameter settings provided in Appendix 3. "Acc" at the top of the images represents Accuracy, where higher values indicate better performance. We can observe that as Acc increases, the node embeddings appear more dispersed. However, due to the large number of nodes in this dataset, the visualization becomes cluttered. Despite this, based on the increasing Acc values and the patterns at the edges, we can infer that the node embeddings are likely becoming more dispersed, although they overlap on the 2D plane..

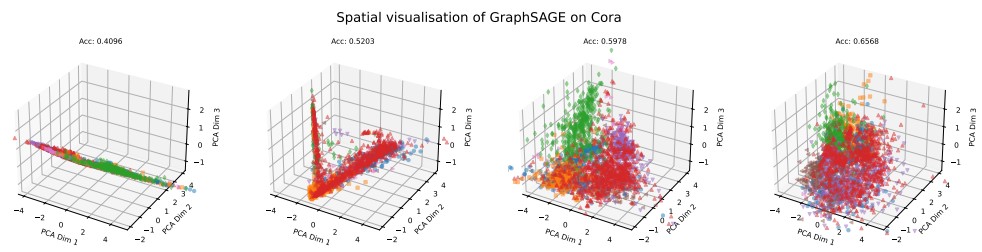

Figure 38: These four images are intended to demonstrate the relationship between the distribution characteristics of node embeddings and their performance on the downstream node classification task. The GraphSAGE model is run on the Cora dataset, with candidate hyperparameter settings provided in Appendix 3. "Acc" at the top of the images represents Accuracy, where higher values indicate better performance. We can observe that as Acc increases, the node embeddings become more dispersed.

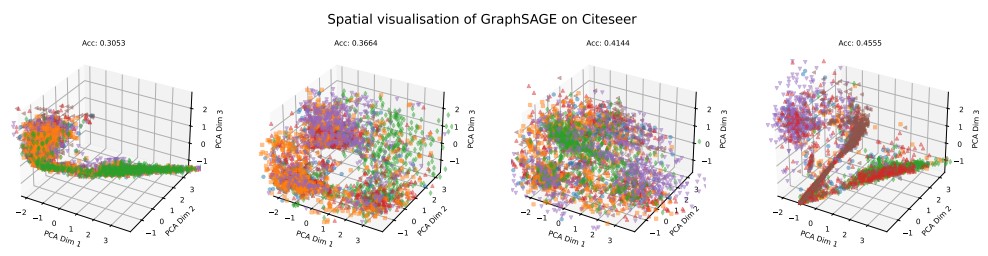

Figure 39: These four images are intended to demonstrate the relationship between the distribution characteristics of node embeddings and their performance on the downstream node classification task. The GraphSAGE model is run on the Citeseer dataset, with candidate hyperparameter settings provided in Appendix 3. "Acc" at the top of the images represents Accuracy, where higher values indicate better performance. We can observe that as Acc increases, the node embeddings become more dispersed.

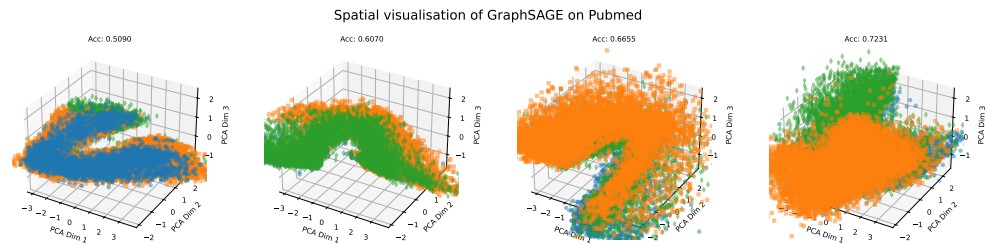

Figure 40: These four images are intended to demonstrate the relationship between the distribution characteristics of node embeddings and their performance on the downstream node classification task. The GraphSAGE model is run on the Pubmed dataset, with candidate hyperparameter settings provided in Appendix 3. "Acc" at the top of the images represents Accuracy, where higher values indicate better performance. We can observe that as Acc increases, the node embeddings appear more dispersed. However, due to the large number of nodes in this dataset, the visualization becomes cluttered. Despite this, based on the increasing Acc values and the patterns at the edges, we can infer that the node embeddings are likely becoming more dispersed, although they overlap on the 2D plane.

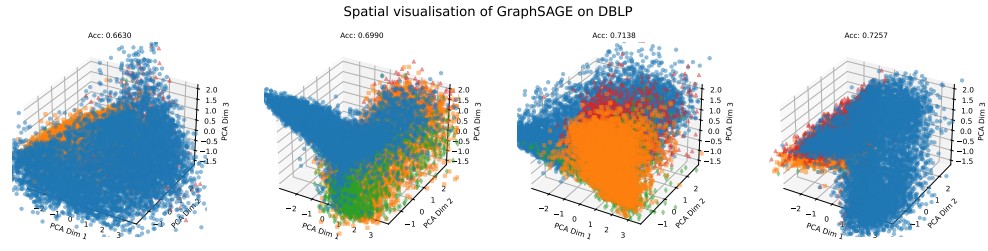

Figure 41: These four images are intended to demonstrate the relationship between the distribution characteristics of node embeddings and their performance on the downstream node classification task. The GraphSAGE model is run on the DBLP dataset, with candidate hyperparameter settings provided in Appendix 3. "Acc" at the top of the images represents Accuracy, where higher values indicate better performance. We can observe that as Acc increases, the node embeddings appear more dispersed. However, due to the large number of nodes in this dataset, the visualization becomes cluttered. Despite this, based on the increasing Acc values and the patterns at the edges, we can infer that the node embeddings are likely becoming more dispersed, although they overlap on the 2D plane.

## F.2 VALIDATE HYPOTHESIS FROM SPATIAL PERSPECTIVE FOR CSOR

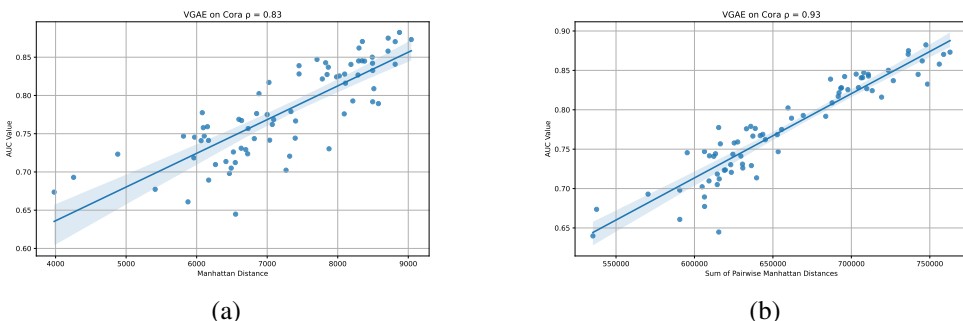

(a)                                     (b)

Figure 42: This figure demonstrates the hypothesis that "higher quality node embeddings tend to be farther away from lower quality node embeddings across each dimension. In other words, as the quality improves, the spatial distance from other node embeddings increases." We use the Manhattan distance to calculate the spatial distance between two node embeddings, using the worst-performing node embedding (the one with the lowest AUC value in downstream tasks) as the baseline. In plot (a), the distances of all other node embeddings from this baseline are calculated. In plot (b), the distances are derived from pairwise comparisons of all node embeddings and then summed. The x-axis represents the Manhattan distance, and the y-axis represents the AUC value corresponding to each node embedding. The $\rho$ character represents the Spearman correlation coefficient between the AUC values and the Manhattan distances. These plots represent the results of running the VGAE model on the Cora dataset. Each dataset is evaluated using 75 sets of HP values.

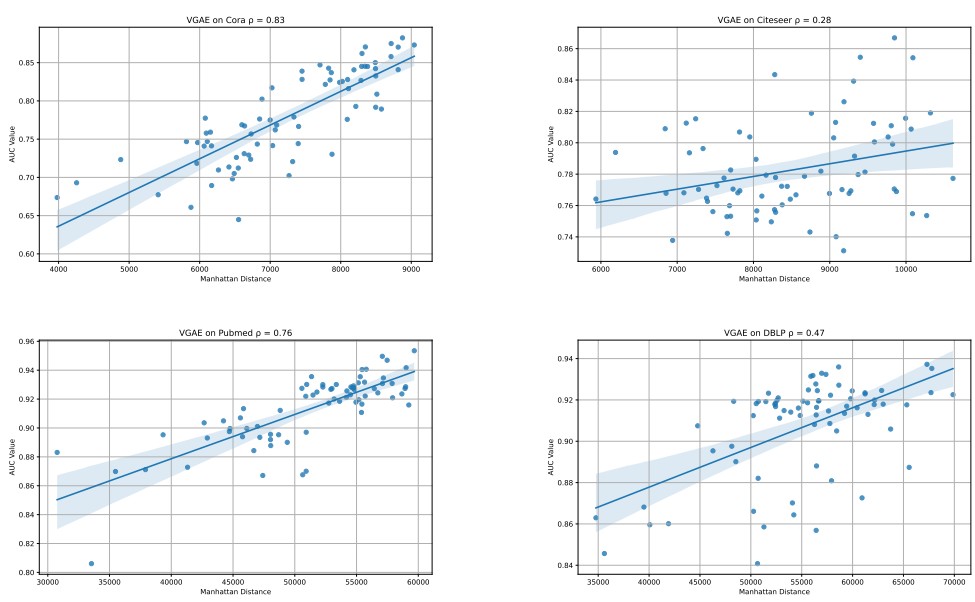

Figure 43: Complete results for VGAE. The interpretations are similar to those given in Figure 42.

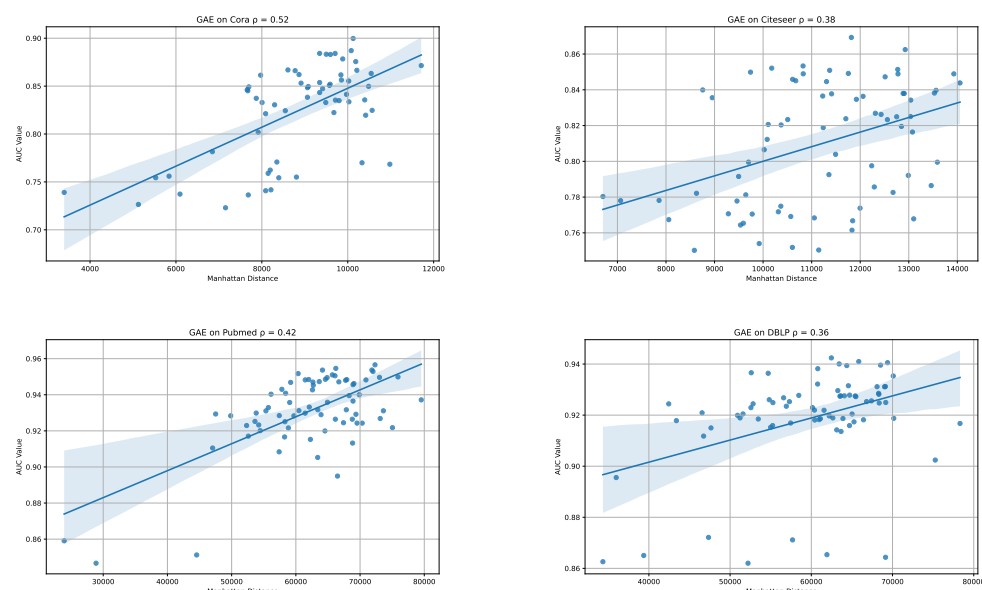

Figure 44: Complete results for GAE. The interpretations are similar to those given in Figure 42.

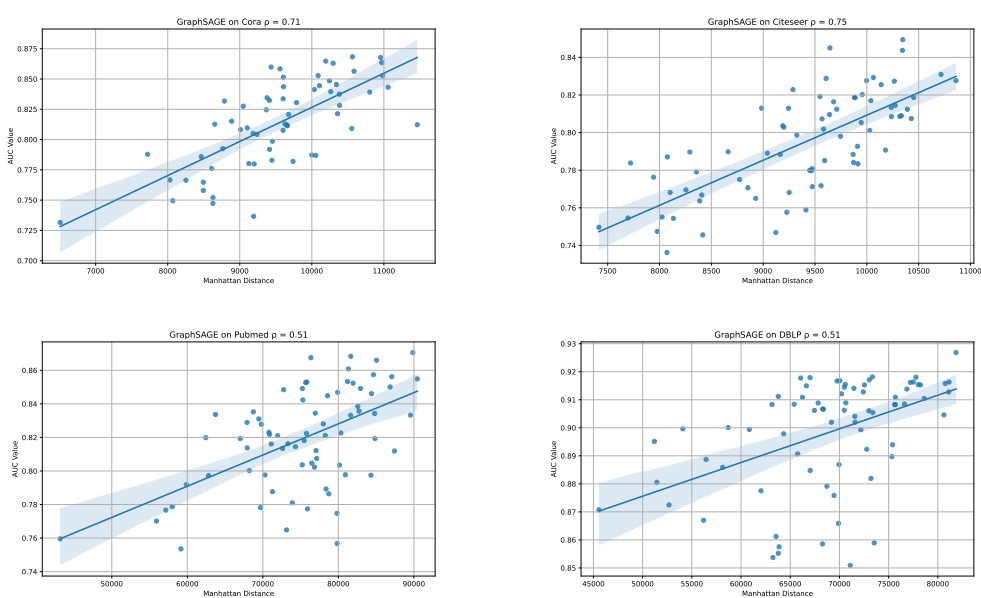

Figure 45: Complete results for GraphSAGE. The interpretations are similar to those given in Figure 42.

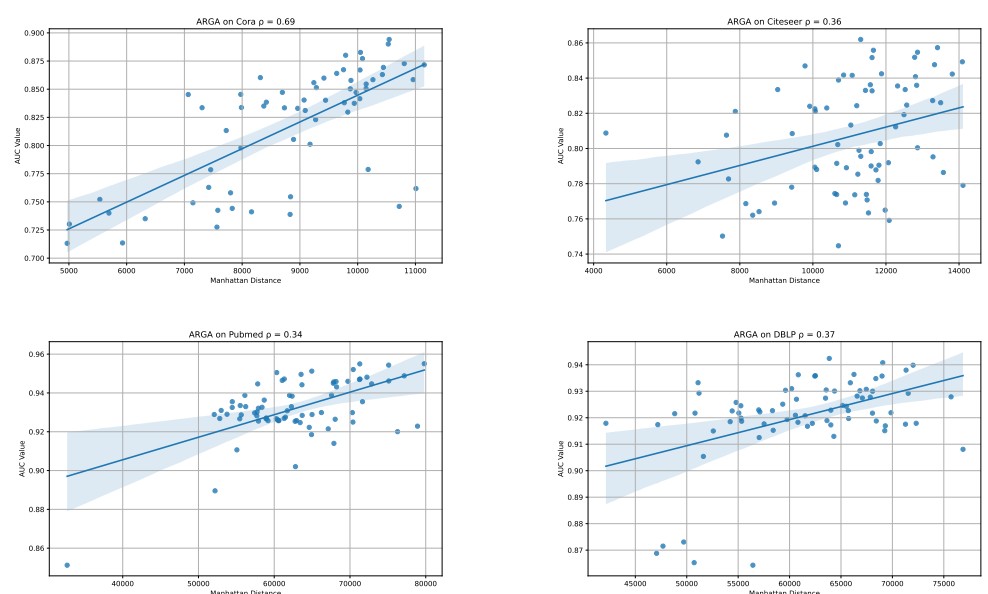

Figure 46: Complete results for ARGA. The interpretations are similar to those given in Figure 42.

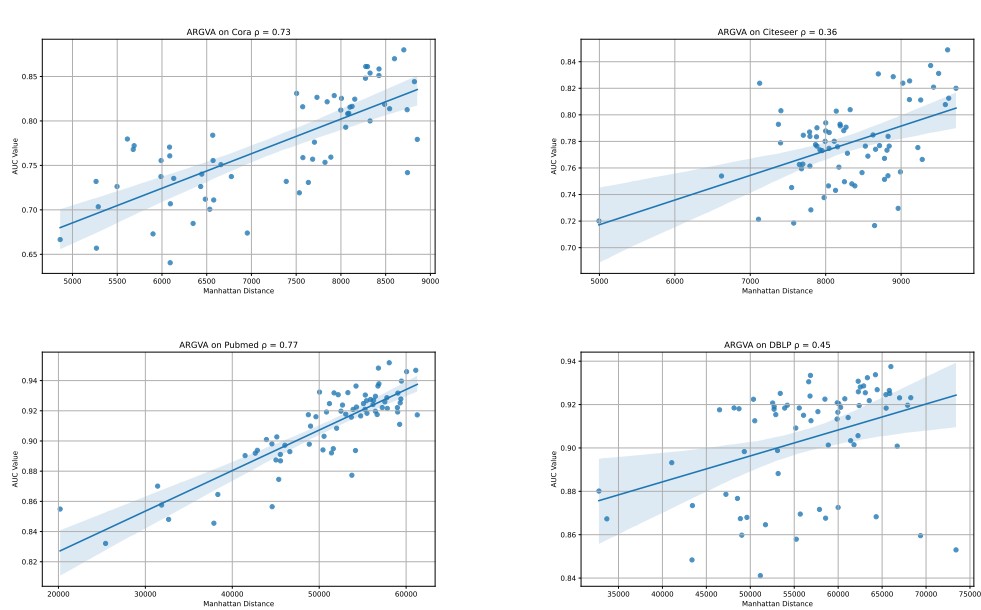

Figure 47: Complete results for ARGVA. The interpretations are similar to those given in Figure 42.

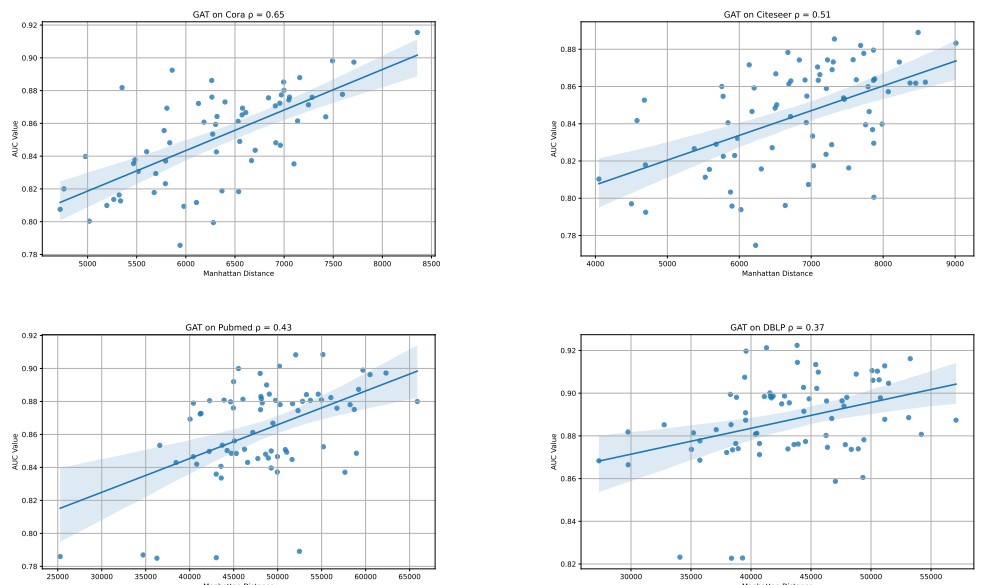

Figure 48: Complete results for GAT. The interpretations are similar to those given in Figure 42.

### F.3 SPECTRAL DISTRIBUTION FOR SSOR

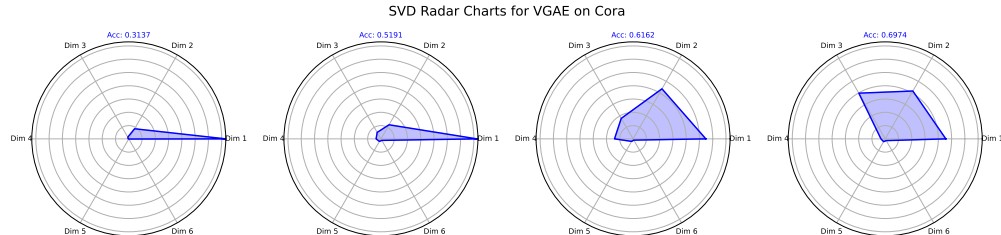

Figure 49: This figure illustrates that node embedding performance is correlated with the spatial occupancy of the embeddings when observed from a spectral-based perspective. We ran VGAE on the Cora dataset with 1280 different sets of hyperparameter values to obtain 1280 node embeddings. From these, we uniformly selected 4 embeddings from the worst to the best performance (AUC values) to observe the performance variation. For each of these 4 node embeddings, SVD was performed to obtain the singular values. Each singular value was then placed on its respective axis, evenly distributed over 360 degrees. This approach aligns with our observation objective in CSOR, suggesting that the area of radar can represent the space occupancy rate of the node embeddings.

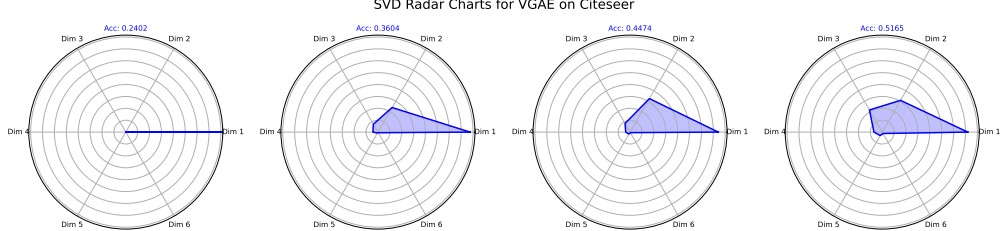

Figure 50: VGAE on Citeseer. The interpretations are similar to those given in Figure 49.

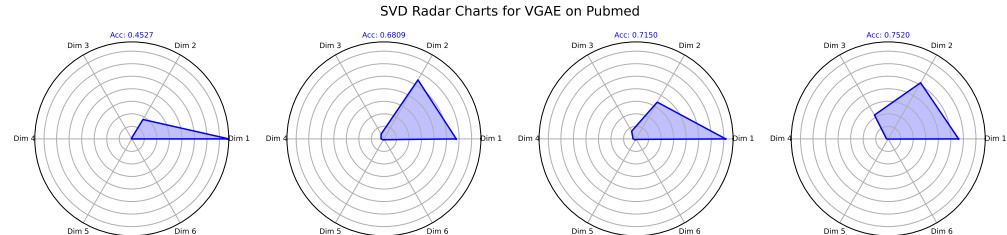

Figure 51: VGAE on Pubmed. The interpretations are similar to those given in Figure 49.

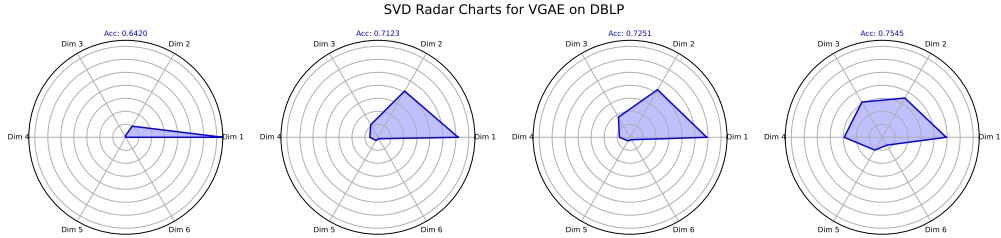

Figure 52: VGAE on DBLP. The interpretations are similar to those given in Figure 49.

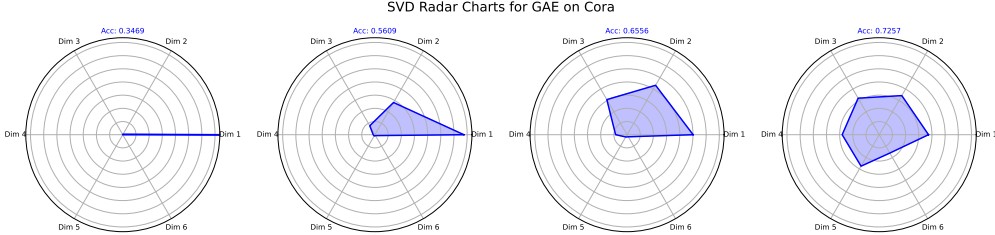

Figure 53: GAE on Cora. The interpretations are similar to those given in Figure 49.

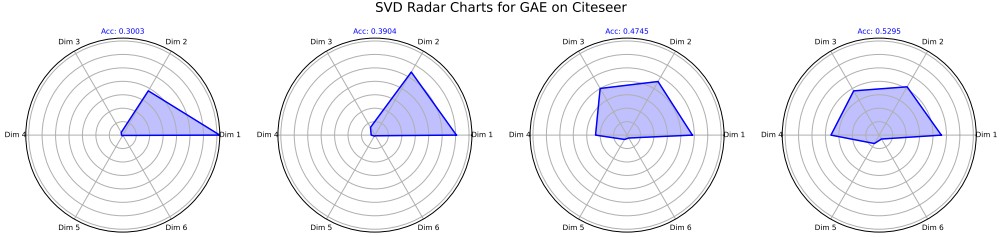

Figure 54: GAE on Citeseer. The interpretations are similar to those given in Figure 49.

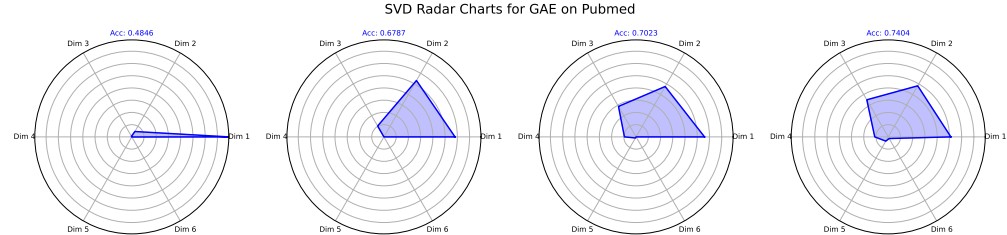

Figure 55: GAE on Pubmed. The interpretations are similar to those given in Figure 49.

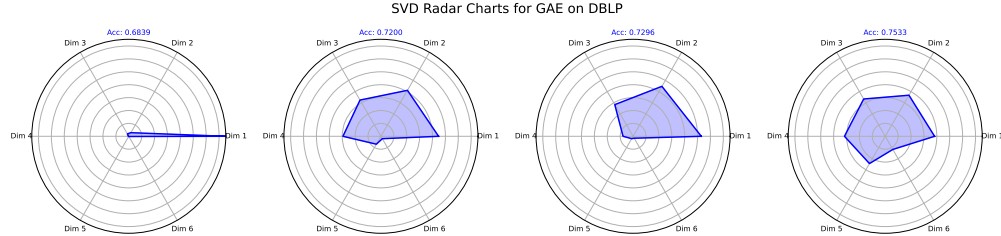

Figure 56: GAE on DBLP. The interpretations are similar to those given in Figure 49.

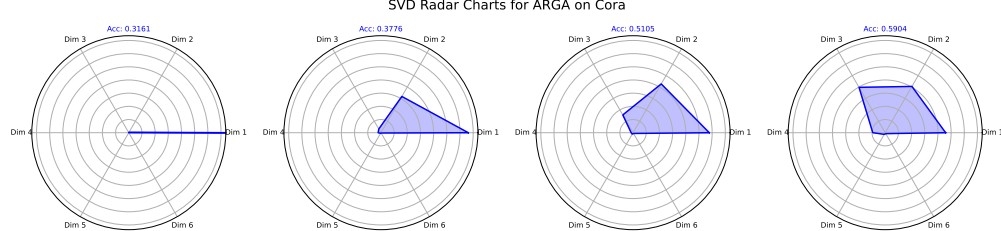

Figure 57: ARGA on Cora. The interpretations are similar to those given in Figure 49.

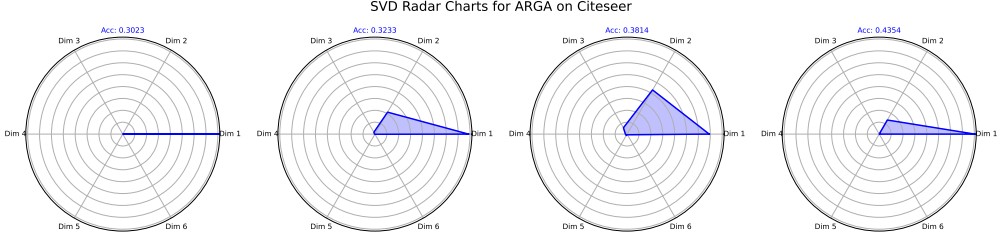

Figure 58: ARGA on Citeseer. The interpretations are similar to those given in Figure 49.

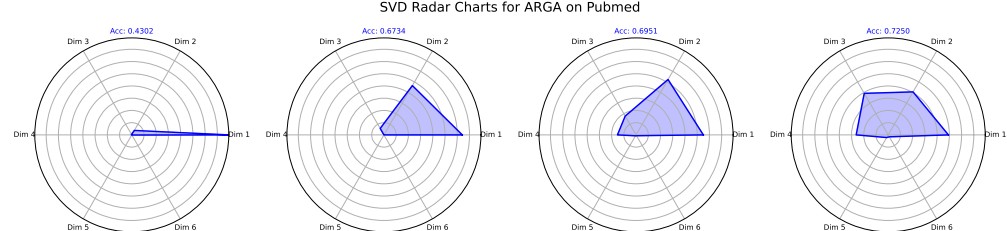

Figure 59: ARGA on Pubmed. The interpretations are similar to those given in Figure 49.

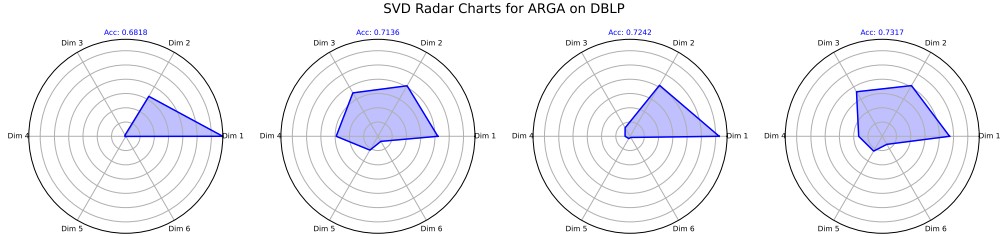

Figure 60: ARGA on DBLP. The interpretations are similar to those given in Figure 49.

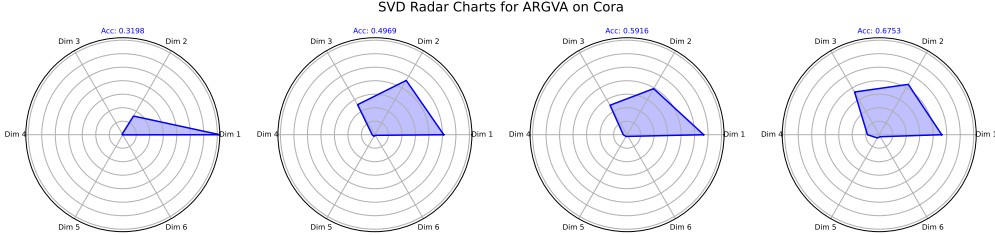

Figure 61: ARGVA on Cora. The interpretations are similar to those given in Figure 49.

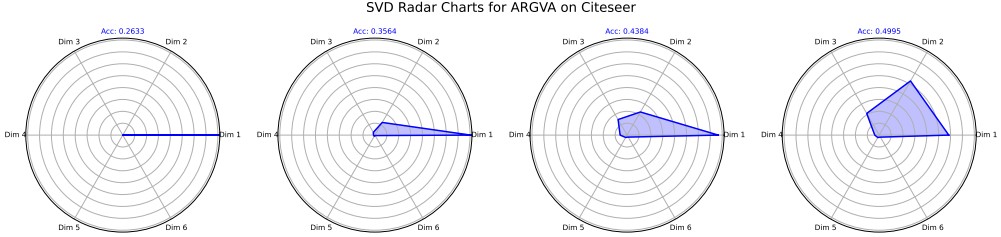

Figure 62: ARGVA on Citeseer. The interpretations are similar to those given in Figure 49.

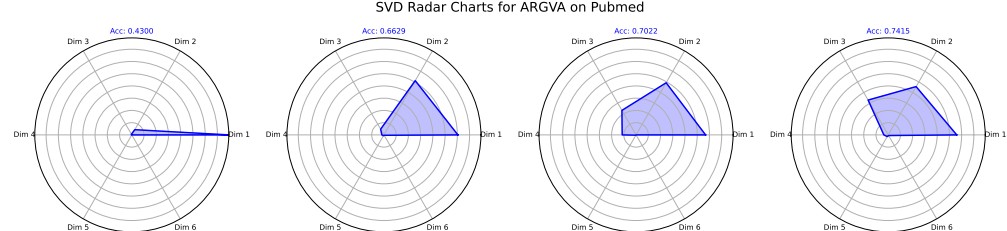

Figure 63: ARGVA on Pubmed. The interpretations are similar to those given in Figure 49.

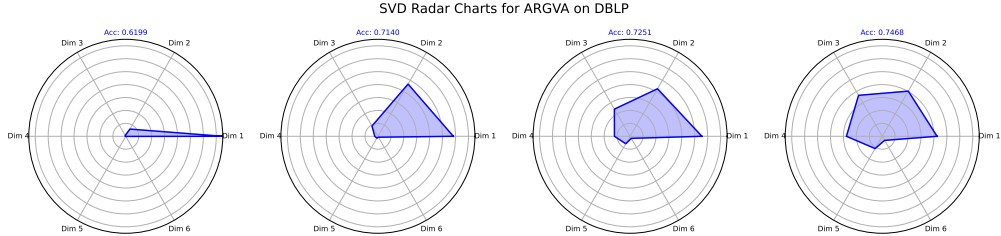

Figure 64: ARGVA on DBLP. The interpretations are similar to those given in Figure 49.

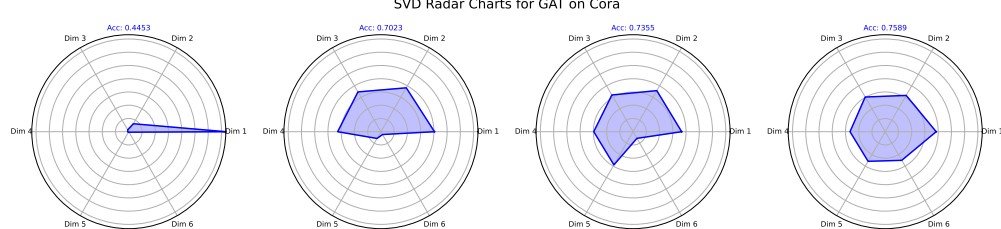

Figure 65: GAT on Cora. The interpretations are similar to those given in Figure 49.

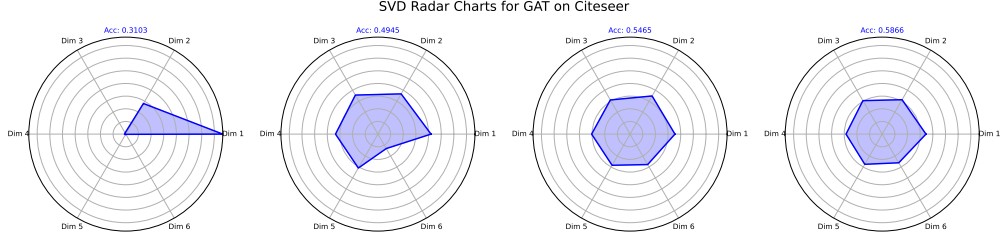

Figure 66: GAT on Citeseer. The interpretations are similar to those given in Figure 49.

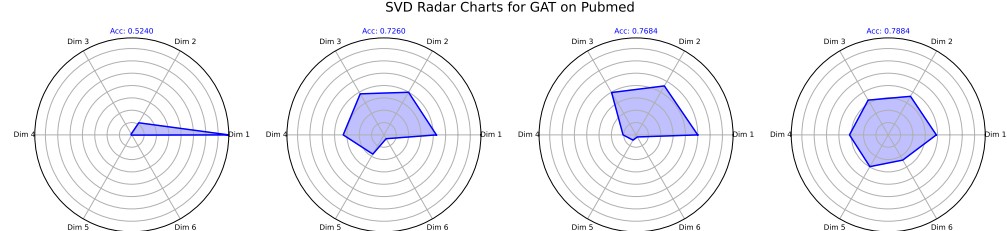

Figure 67: GAT on Pubmed. The interpretations are similar to those given in Figure 49.

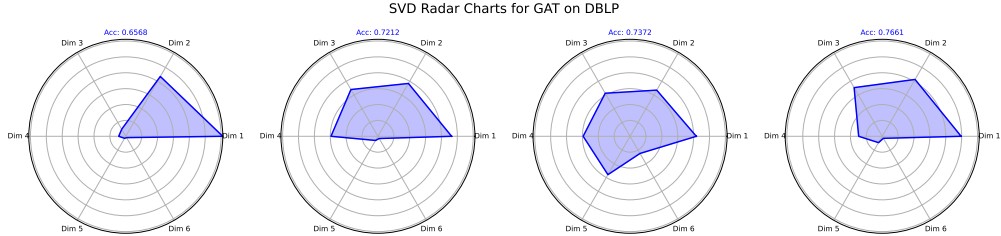

Figure 68: GAT on DBLP. The interpretations are similar to those given in Figure 49.

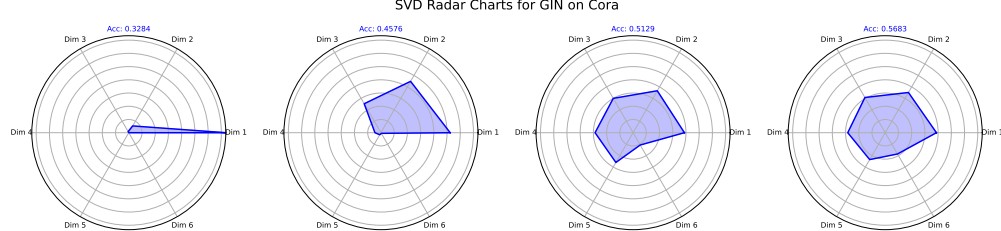

Figure 69: GIN on Cora. The interpretations are similar to those given in Figure 49.

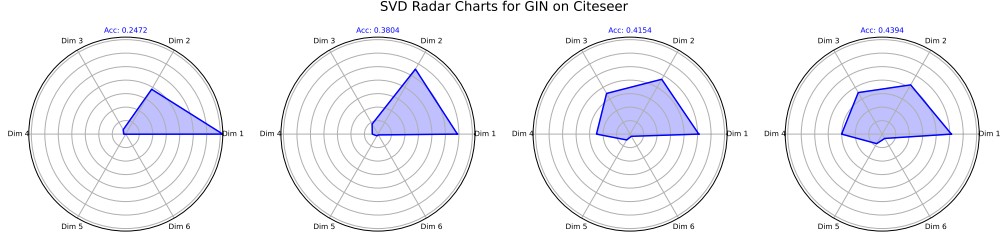

Figure 70: GIN on Citeseer. The interpretations are similar to those given in Figure 49.

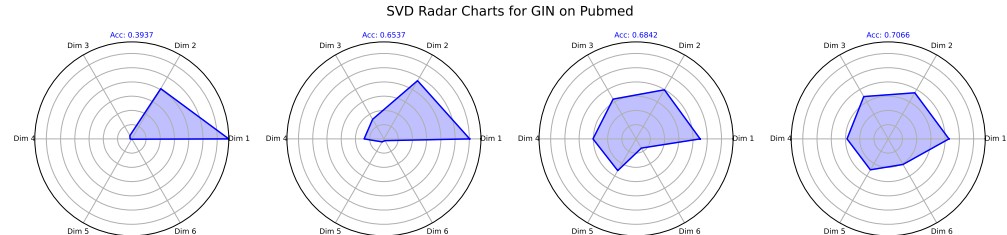

Figure 71: GIN on Pubmed. The interpretations are similar to those given in Figure 49.

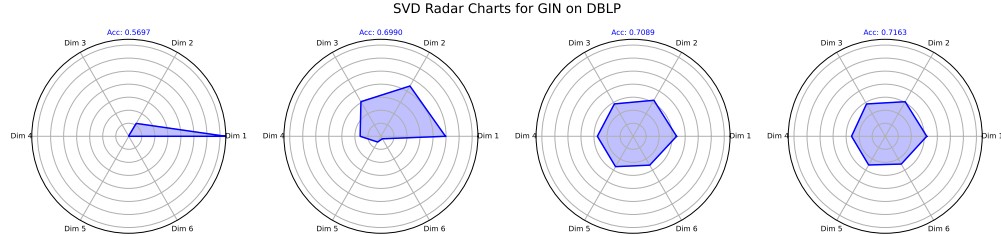

Figure 72: GIN on DBLP. The interpretations are similar to those given in Figure 49.

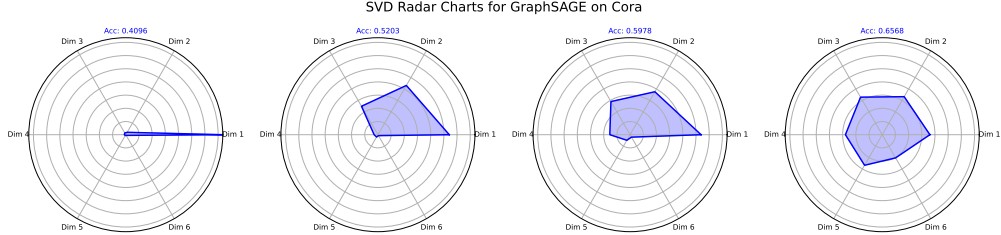

Figure 73: GraphSAGE on Cora. The interpretations are similar to those given in Figure 49.

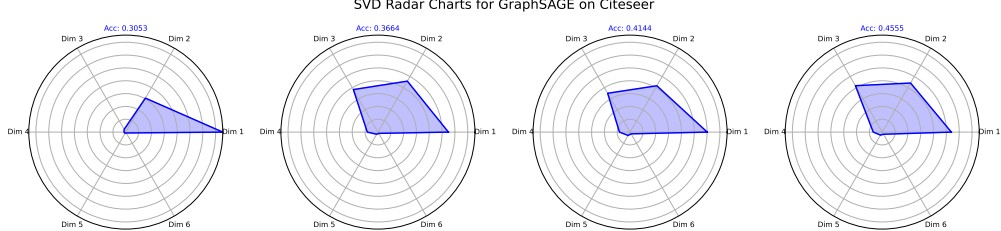

Figure 74: GraphSAGE on Citeseer. The interpretations are similar to those given in Figure 49.

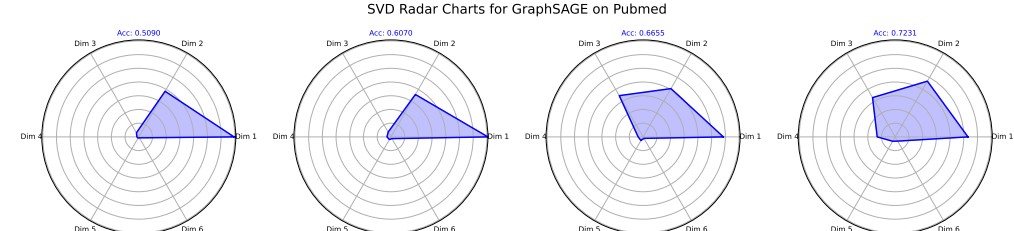

Figure 75: GraphSAGE on Pubmed. The interpretations are similar to those given in Figure 49.

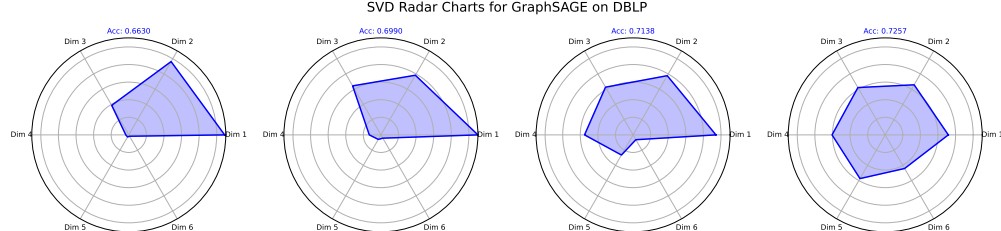

Figure 76: GraphSAGE on DBLP. The interpretations are similar to those given in Figure 49.

## F.4 VALIDATE HYPOTHESIS FROM SPECTRAL PERSPECTIVE FOR SSOR

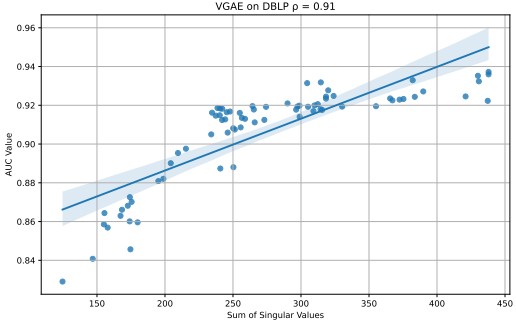

Figure 77: This figure is designed to prove our hypothesis that "quantifying the distinctness of node embeddings using the sum of singular values can achieve similar effects as CSOR, which utilizes accumulating pairwise distances." Using VGAE with 75 different hyperparameter values on the DBLP dataset, corresponding node embeddings were generated. The x-axis represents the sum of singular values of the node embeddings, while the y-axis represents the performance (AUC values) of the node embeddings in the downstream task of link prediction. The results show a high correlation between the sum of singular values and AUC values, supporting our hypothesis.

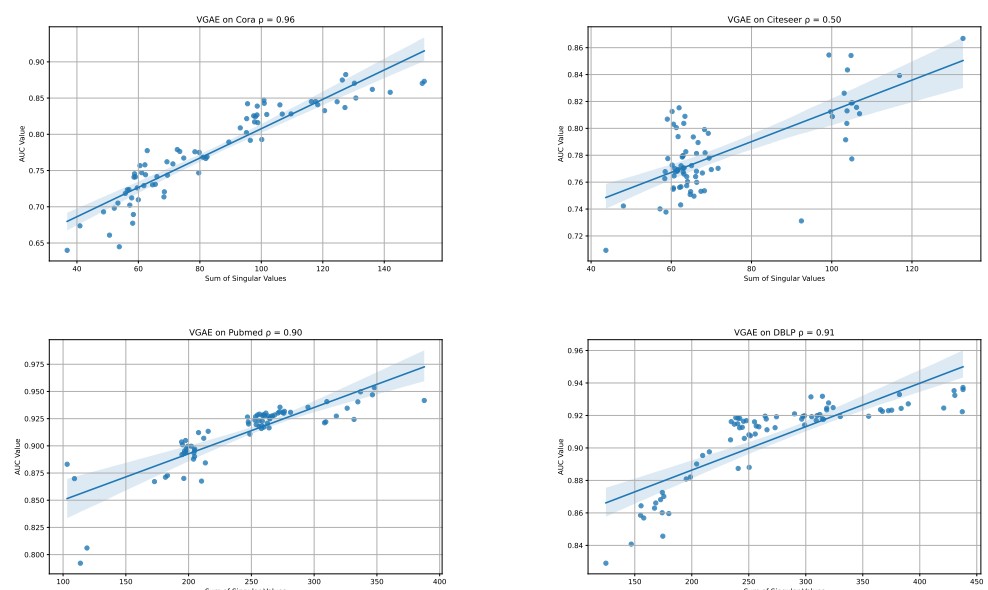

Figure 78: Complete results for VGAE. The interpretations are similar to those given in Figure 77.

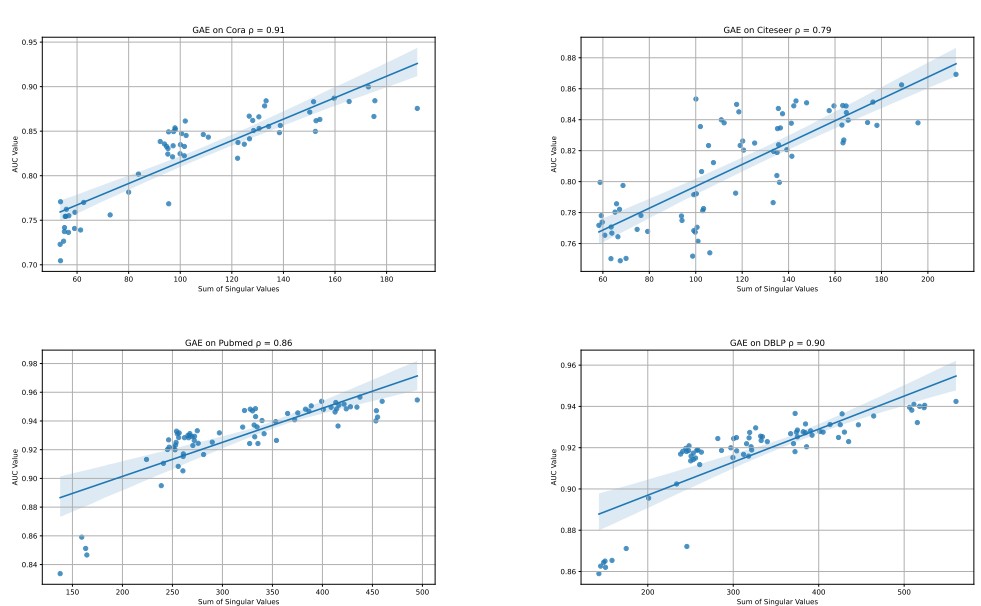

Figure 79: Complete results for GAE. The interpretations are similar to those given in Figure 77.

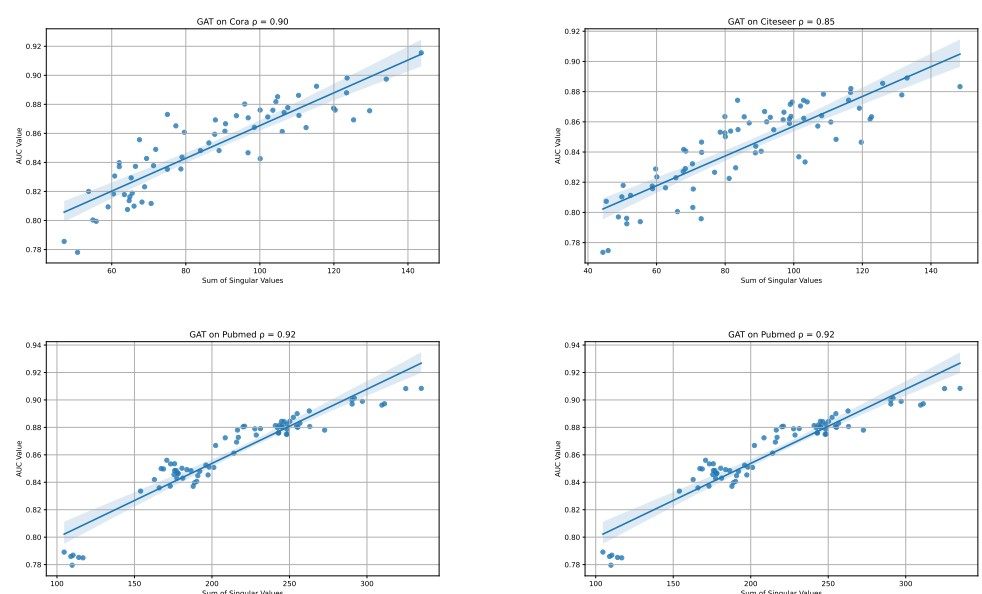

Figure 80: Complete results for GAT. The interpretations are similar to those given in Figure 77.

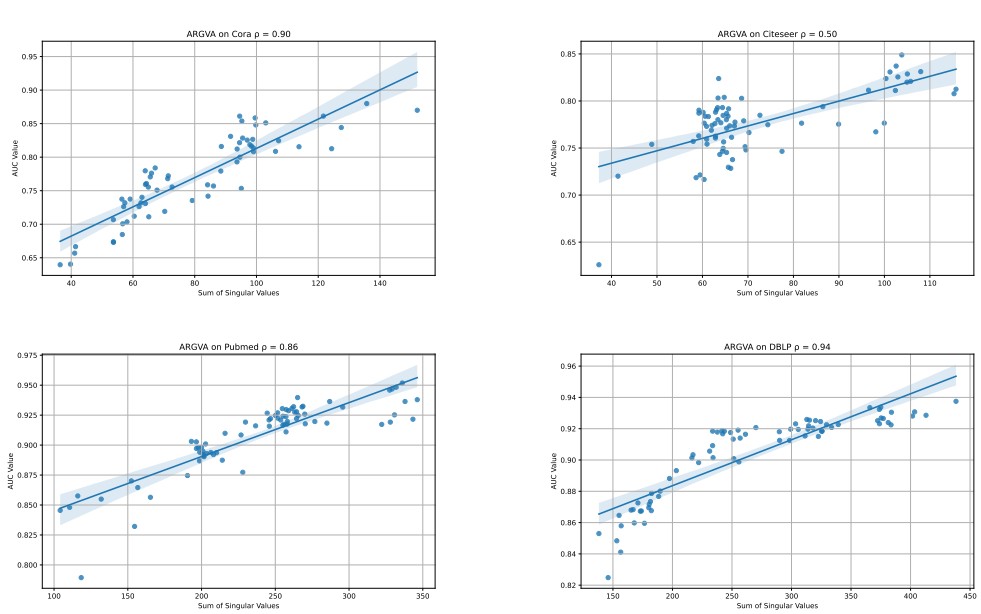

Figure 81: Complete results for ARGVA. The interpretations are similar to those given in Figure 77.

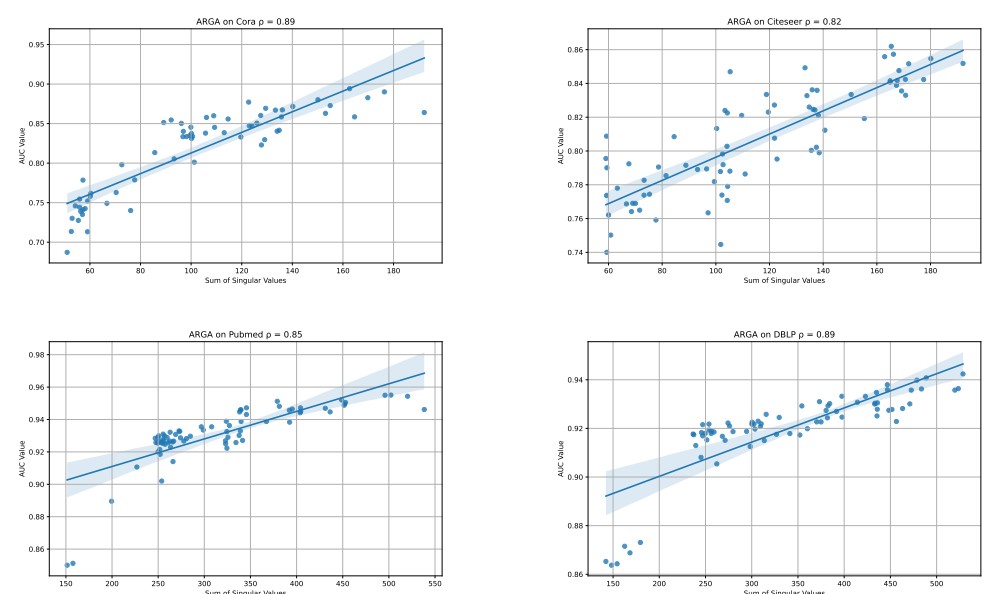

Figure 82: Complete results for ARGA. The interpretations are similar to those given in Figure 77.

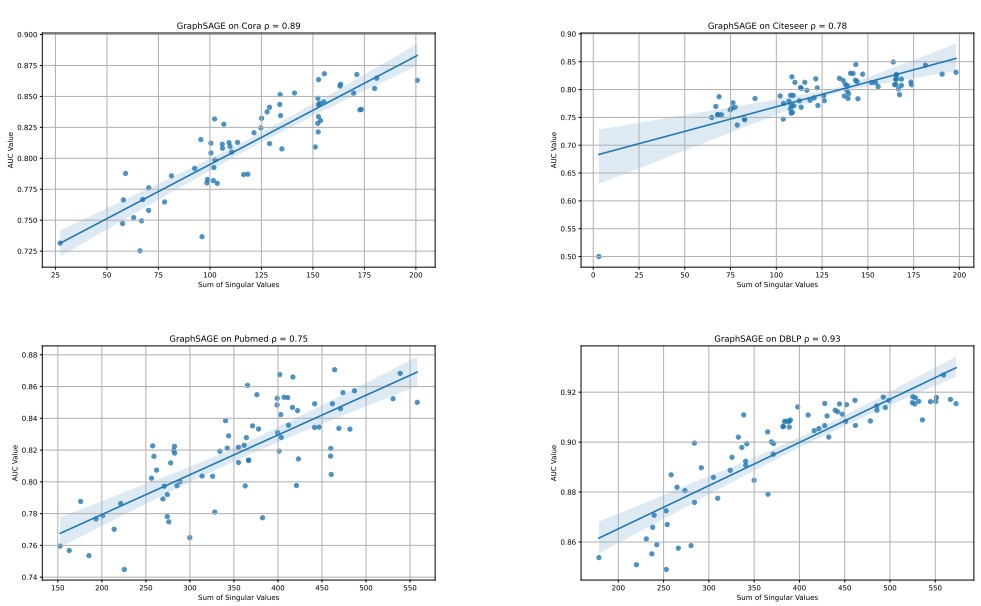

Figure 83: Complete results for GraphSAGE. The interpretations are similar to those given in Figure 77.

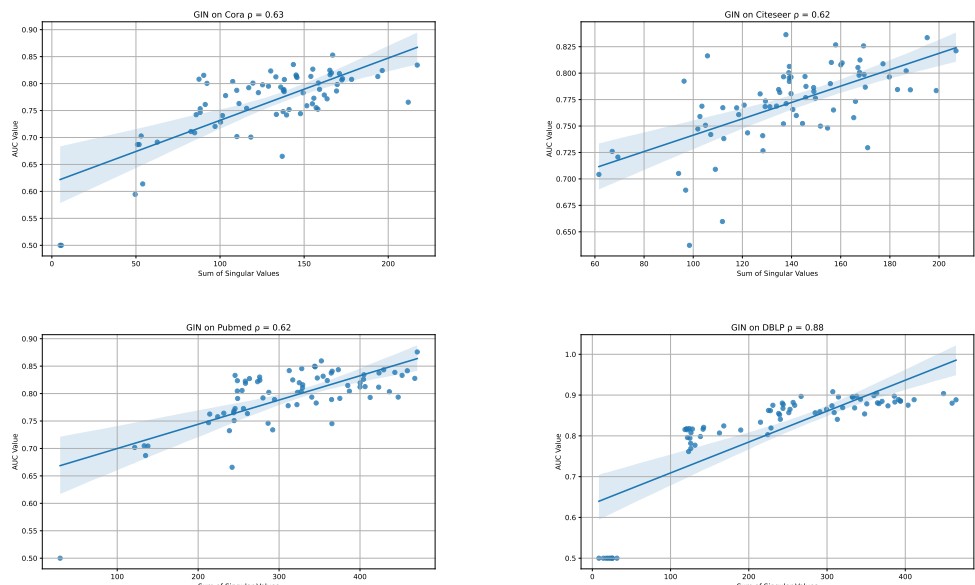

Figure 84: Complete results for GIN. The interpretations are similar to those given in Figure 77.

## G COMPLEXITY ANALYSIS ON CSOR AND SSOR

We perform the time and space complexity on CSOR and SSOR. The following symbols are used in our analysis: $k$ represents the number of hyperparameter settings, $N$ is the number of samples (nodes) in each embedding, and $D$ is the number of dimensions in each embedding.

---

**Algorithm 1** HPO for unsupervised node embedding using CSOR

---

**Input:** A graph $\mathcal{G}$ and a set of HP configurations $\mathcal{H}$ for UGRL function $f(\cdot)$.
**Output:** Optimal HP configuration $h^*$ and the corresponding graph embeddings $f_{h^*}(\mathcal{G})$.
**for** each $h \in \mathcal{H}$ **do**
    Generate graph embeddings $\mathbf{Z}(h) = f_h(\mathcal{G}) \in \mathbb{R}^{N \times D}$ using $f(\cdot)$ with configuration $h$.
**end for**
**for** $i = 1$ to $|\mathcal{H}|$ **do**
    **for** $j = 1$ to $|\mathcal{H}|$, $j \neq i$ **do**
        Calculate the difference (distance) $D_{i,j} = \text{diff}(\mathbf{Z}(h_i), \mathbf{Z}(h_j))$.
    **end for**
    Calculate the sum of differences $s_i = \sum_{j=1, j \neq i}^{|\mathcal{H}|} D_{i,j}$ for $h_i$.
**end for**
Select $h^*$ as $h^* = \arg\max_{h_i \in \mathcal{H}} \{s_i \mid i = 1, \ldots, |\mathcal{H}|\}$.
**return** $h^*$ and $f_{h^*}(\mathcal{G})$.

---

### G.1 TIME COMPLEXITY ANALYSIS

**CSOR** For CSOR, we quantify the distinctness of node embeddings by accumulating the distances of a given node embedding compared to all others. This pairwise comparison operation can capture more distributional information about all node embeddings. Given a graph $\mathcal{G}$ with $N$ nodes and an embedding dimensionality of $D$, and considering $k$ hyperparameter configurations, the total time complexity for CSOR is influenced by the number of hyperparameter configurations ($k$) and the dimensionality of the embeddings ($D$). Specifically, the time complexity is $O(k^2 \cdot N \cdot D)$, as each pairwise distance calculation involves $O(N \cdot D)$ operations, and there are $\binom{k}{2}$ pairs.

**SSOR**  For SSOR, the distinctness of node embeddings is measured by summing the singular values of the embedding matrix obtained through Singular Value Decomposition (SVD). The computation involves converting embeddings to a suitable format, calculating the covariance matrix, and performing SVD. Given $N$ nodes, $D$ dimensions, and $k$ hyperparameter configurations, the total time complexity for SSOR is determined by $O(k \cdot (D^2 \cdot N + D^3))$, where $D^2 \cdot N$ accounts for the covariance matrix calculation and $D^3$ accounts for the SVD.

## G.2 Space Complexity Analysis

**CSOR**  Each embedding requires space proportional to the number of nodes ($N$) and the dimensionality of the embeddings ($D$). For $k$ embeddings, the space complexity is $O(k \cdot N \cdot D)$. Temporary storage for distance calculations requires $O(N \cdot D)$, which does not significantly impact the overall space complexity. Thus, the total space complexity for CSOR is $O(k \cdot N \cdot D)$.

**SSOR**  Each embedding also requires space proportional to $N$ and $D$. For $k$ embeddings, the space complexity is $O(k \cdot N \cdot D)$. Storing the covariance matrix requires $O(D^2)$ space, and storing the singular values requires $O(D)$ space. Thus, the total space complexity for SSOR is $O(k \cdot N \cdot D + D^2)$.

## G.3 Comparison of Space and Time Complexity

**Time Complexity**:

- CSOR: $O(k^2 \cdot N \cdot D)$

- SSOR: $O(k \cdot (D^2 \cdot N + D^3))$

CSOR has a quadratic dependency on the number of hyperparameter settings ($k$), while SSOR has a cubic dependency on the number of dimensions ($D$) but is linear with respect to $k$.

**Space Complexity**:

- CSOR: $O(k \cdot N \cdot D)$

- SSOR: $O(k \cdot N \cdot D + D^2)$

Both methods primarily depend on $k$, $N$, and $D$, but spectral-based methods also include an additional $D^2$ term due to the covariance matrix.

The CSOR method, a spatial-based approach, is characterized by its quadratic time complexity with respect to the number of hyperparameter settings ($k$) and its linear space complexity with respect to the product of the number of samples ($N$) and the number of dimensions ($D$). CSOR has better scalability with respect to graph size, making it more suitable for handling larger graph embeddings. However, its performance significantly slows down when the number of candidate hyperparameter settings is very large.

In contrast, SSOR and any other IS relying on singular values, while linear in the number of hyperparameter settings ($k$), exhibit higher time complexity due to their cubic dependence on the number of dimensions ($D$). These methods also require additional space to store the covariance matrix, resulting in a space complexity of $O(k \cdot N \cdot D + D^2)$. Due to the presence of the SVD operation, spectral-based methods are not as scalable for very large graph embeddings but can handle a larger number of candidate hyperparameter settings more efficiently compared to CSOR.

## H  COMPLETE EXPERIMENTAL RESULTS

Table 5: Experimental results on node classification: accuracy values (relative rankings) of various internal strategies across 7 GNN models on datasets Pubmed and DBLP.

| Dataset | Method | VGAE | GAE | ARGA | ARGVA | GAT | GIN | GraphSAGE | Avg. Rank |
|---|---|---|---|---|---|---|---|---|---|
| | CSOR | 0.78(2) | 0.80(3) | 0.82(1) | 0.75(5) | 0.80(1) | 0.75(4) | 0.75(1) | 2.43 |
| | SSOR | 0.77(3) | 0.82(1) | 0.79(3) | 0.80(1) | 0.80(2) | 0.76(1) | 0.74(3) | **2.00** |
| | RankMe | 0.81(1) | 0.80(4) | 0.79(3) | 0.80(1) | 0.80(2) | 0.76(1) | 0.74(2) | **2.00** |
| | NESum | 0.52(7) | 0.70(7) | 0.48(8) | 0.78(4) | 0.79(5) | 0.58(5) | 0.71(5) | 5.86 |
| Pubmed | AlphaReQ | 0.69(6) | 0.79(5) | 0.77(5) | 0.71(6) | 0.70(7) | 0.51(7) | 0.65(6) | 6.00 |
| | Incoherence | 0.72(5) | 0.63(8) | 0.74(6) | 0.48(8) | 0.76(6) | 0.41(9) | 0.55(8) | 7.14 |
| | ConditionNumber | 0.50(8) | 0.72(6) | 0.67(7) | 0.61(7) | 0.53(9) | 0.50(8) | 0.57(7) | 7.43 |
| | SelfCluster | 0.48(9) | 0.57(9) | 0.47(9) | 0.44(9) | 0.66(8) | 0.52(6) | 0.54(9) | 8.43 |
| | StableRank | 0.77(3) | 0.82(1) | 0.81(2) | 0.80(1) | 0.79(4) | 0.76(1) | 0.74(3) | 2.14 |
| | CSOR | 0.78(2) | 0.77(3) | 0.77(1) | 0.78(4) | 0.76(5) | 0.73(1) | 0.72(4) | 2.86 |
| | SSOR | 0.77(3) | 0.78(1) | 0.75(2) | 0.79(1) | 0.77(2) | 0.72(3) | 0.74(1) | **1.86** |
| | RankMe | 0.78(1) | 0.75(4) | 0.75(4) | 0.79(1) | 0.77(2) | 0.72(3) | 0.74(1) | 2.29 |
| | NESum | 0.69(8) | 0.71(7) | 0.70(6) | 0.72(6) | 0.78(1) | 0.72(2) | 0.71(6) | 5.14 |
| DBLP | AlphaReQ | 0.71(6) | 0.74(5) | 0.72(5) | 0.73(5) | 0.71(8) | 0.69(6) | 0.70(7) | 6.00 |
| | Incoherence | 0.72(5) | 0.72(6) | 0.69(8) | 0.72(7) | 0.71(7) | 0.58(9) | 0.71(5) | 6.71 |
| | ConditionNumber | 0.66(9) | 0.70(8) | 0.70(7) | 0.70(8) | 0.71(6) | 0.61(7) | 0.69(8) | 7.57 |
| | SelfCluster | 0.69(7) | 0.68(9) | 0.69(8) | 0.63(9) | 0.68(9) | 0.61(7) | 0.68(9) | 8.29 |
| | StableRank | 0.77(3) | 0.78(1) | 0.75(2) | 0.79(1) | 0.77(2) | 0.72(3) | 0.74(1) | **1.86** |

Table 6: Experimental results on link prediction: AUC-ROC values and the relative rankings of various internal strategies on 7 GNN models across 4 benchmark datasets.

| Dataset | Method | VGAE | GAE | ARGA | ARGVA | GAT | GIN | GraphSAGE | Avg. Rank |
|---|---|---|---|---|---|---|---|---|---|
| | CSOR | 0.98(1) | 0.98(4) | 0.97(4) | 0.97(4) | 0.97(1) | 0.97(3) | 0.97(4) | 3.00 |
| | SSOR | 0.98(1) | 0.98(1) | 0.97(1) | 0.97(2) | 0.97(2) | 0.98(2) | 0.98(1) | **1.43** |
| | RankMe | 0.98(1) | 0.98(1) | 0.97(3) | 0.98(1) | 0.97(2) | 0.98(1) | 0.98(1) | **1.43** |
| | NESum | 0.83(7) | 0.85(7) | 0.82(8) | 0.92(7) | 0.92(6) | 0.96(5) | 0.83(7) | 6.71 |
| Cora | AlphaReQ | 0.92(5) | 0.97(5) | 0.95(5) | 0.95(6) | 0.81(7) | 0.93(6) | 0.95(5) | 5.57 |
| | Incoherence | 0.89(6) | 0.84(8) | 0.83(7) | 0.95(5) | 0.96(5) | 0.75(9) | 0.93(6) | 6.57 |
| | ConditionNumber | 0.76(8) | 0.89(6) | 0.90(6) | 0.91(8) | 0.81(7) | 0.82(7) | 0.83(7) | 7.00 |
| | SelfCluster | 0.76(8) | 0.84(9) | 0.82(8) | 0.78(9) | 0.81(9) | 0.79(8) | 0.81(9) | 8.57 |
| | StableRank | 0.98(1) | 0.98(1) | 0.97(1) | 0.97(2) | 0.97(2) | 0.97(4) | 0.98(3) | 2.00 |
| | CSOR | 0.97(4) | 0.98(4) | 0.95(4) | 0.97(4) | 0.98(4) | 0.97(4) | 0.98(1) | 3.57 |
| | SSOR | 0.97(1) | 0.98(2) | 0.97(1) | 0.98(1) | 0.98(1) | 0.98(1) | 0.97(3) | 1.43 |
| | RankMe | 0.97(3) | 0.98(3) | 0.97(1) | 0.98(1) | 0.98(3) | 0.98(3) | 0.98(1) | 2.14 |
| | NESum | 0.86(7) | 0.96(6) | 0.87(9) | 0.87(7) | 0.93(6) | 0.91(6) | 0.95(5) | 6.57 |
| Citeseer | AlphaReQ | 0.95(5) | 0.97(5) | 0.94(5) | 0.96(5) | 0.95(5) | 0.92(5) | 0.90(6) | 5.14 |
| | Incoherence | 0.69(8) | 0.87(8) | 0.87(8) | 0.95(6) | 0.93(7) | 0.76(9) | 0.90(6) | 7.43 |
| | ConditionNumber | 0.69(8) | 0.89(7) | 0.87(7) | 0.87(9) | 0.89(8) | 0.77(7) | 0.90(6) | 7.43 |
| | SelfCluster | 0.88(6) | 0.87(9) | 0.88(6) | 0.87(8) | 0.86(9) | 0.77(8) | 0.87(9) | 7.86 |
| | StableRank | 0.97(1) | 0.98(1) | 0.97(1) | 0.98(1) | 0.98(1) | 0.98(1) | 0.97(3) | **1.29** |
| | CSOR | 0.98(2) | 0.99(4) | 0.98(4) | 0.98(4) | 0.96(4) | 0.99(1) | 0.96(4) | 3.29 |
| | SSOR | 0.98(3) | 0.99(1) | 0.98(1) | 0.98(1) | 0.97(1) | 0.98(2) | 0.98(1) | **1.43** |
| | RankMe | 0.98(1) | 0.99(3) | 0.98(1) | 0.98(1) | 0.97(1) | 0.98(2) | 0.98(3) | 1.71 |
| | NESum | 0.90(7) | 0.96(7) | 0.86(8) | 0.97(5) | 0.93(6) | 0.93(5) | 0.93(6) | 6.29 |
| Pubmed | AlphaReQ | 0.95(5) | 0.98(5) | 0.97(6) | 0.97(6) | 0.93(7) | 0.89(7) | 0.95(5) | 5.86 |
| | Incoherence | 0.93(6) | 0.87(9) | 0.98(5) | 0.91(8) | 0.93(5) | 0.79(9) | 0.82(9) | 7.29 |
| | ConditionNumber | 0.89(9) | 0.96(6) | 0.96(7) | 0.94(7) | 0.84(8) | 0.89(6) | 0.86(7) | 7.14 |
| | SelfCluster | 0.90(8) | 0.90(8) | 0.86(9) | 0.88(9) | 0.82(9) | 0.89(8) | 0.82(8) | 8.43 |
| | StableRank | 0.98(3) | 0.99(1) | 0.98(3) | 0.98(1) | 0.96(3) | 0.98(2) | 0.98(1) | 2.00 |
| | CSOR | 0.98(4) | 0.98(4) | 0.98(4) | 0.98(4) | 0.97(4) | 0.98(1) | 0.97(4) | 3.57 |
| | SSOR | 0.98(1) | 0.98(2) | 0.98(1) | 0.98(1) | 0.98(1) | 0.98(2) | 0.98(1) | **1.29** |
| | RankMe | 0.98(3) | 0.99(1) | 0.98(3) | 0.98(1) | 0.98(1) | 0.98(2) | 0.98(1) | 1.71 |
| | NESum | 0.90(7) | 0.96(7) | 0.88(6) | 0.96(5) | 0.97(5) | 0.98(5) | 0.94(7) | 6.00 |
| DBLP | AlphaReQ | 0.96(5) | 0.97(6) | 0.97(5) | 0.96(6) | 0.94(6) | 0.94(6) | 0.96(6) | 5.71 |
| | Incoherence | 0.95(6) | 0.97(5) | 0.86(8) | 0.95(7) | 0.93(7) | 0.86(9) | 0.97(5) | 6.71 |
| | ConditionNumber | 0.87(9) | 0.91(8) | 0.87(7) | 0.94(8) | 0.86(8) | 0.88(7) | 0.92(8) | 7.86 |
| | SelfCluster | 0.88(8) | 0.89(9) | 0.86(8) | 0.87(9) | 0.85(9) | 0.88(7) | 0.88(9) | 8.43 |
| | StableRank | 0.98(1) | 0.98(2) | 0.98(1) | 0.98(1) | 0.98(1) | 0.98(2) | 0.98(1) | **1.29** |

Table 7: Experimental results on link prediction: Spearman coefficients between ranking scores given by internal strategies and AUC ROC values.

| Dataset | Method | VGAE | GAE | ARGA | ARGVA | GAT | GIN | GraphSAGE | Average |
|---------|--------|------|-----|------|-------|-----|-----|-----------|---------|
| Cora | CSOR | 0.90 | 0.93 | 0.92 | 0.95 | 0.96 | 0.94 | 0.92 | 0.932 |
| | SSOR | 0.97 | 0.98 | 0.97 | 0.98 | 0.98 | 0.96 | 0.98 | 0.972 |
| | RankMe | 0.95 | 0.97 | 0.96 | 0.96 | 0.97 | 0.97 | 0.97 | 0.964 |
| | NESum | -0.01 | -0.01 | -0.02 | 0.03 | -0.08 | 0.04 | 0.01 | -0.004 |
| | AlphaReQ | -0.30 | -0.46 | -0.27 | -0.51 | -0.88 | -0.88 | -0.15 | -0.493 |
| | Incoherence | 0.03 | 0.03 | -0.11 | -0.00 | -0.03 | -0.16 | 0.00 | -0.035 |
| | ConditionNumber | -0.81 | -0.83 | -0.76 | -0.80 | -0.89 | -0.89 | -0.86 | -0.837 |
| | SelfCluster | -0.97 | -0.98 | -0.96 | -0.97 | -0.98 | -0.98 | -0.98 | -0.977 |
| | StableRank | 0.97 | 0.99 | 0.97 | 0.98 | 0.98 | 0.95 | 0.98 | 0.974 |
| Citeseer | CSOR | 0.85 | 0.92 | 0.84 | 0.90 | 0.96 | 0.90 | 0.82 | 0.888 |
| | SSOR | 0.93 | 0.99 | 0.94 | 0.93 | 0.99 | 0.98 | 0.98 | 0.963 |
| | RankMe | 0.93 | 0.98 | 0.94 | 0.93 | 0.98 | 0.98 | 0.97 | 0.958 |
| | NESum | -0.01 | 0.03 | 0.04 | 0.02 | 0.04 | 0.00 | 0.00 | 0.017 |
| | AlphaReQ | 0.01 | -0.65 | -0.24 | -0.20 | -0.88 | -0.88 | -0.17 | -0.391 |
| | Incoherence | 0.04 | -0.06 | -0.35 | -0.02 | -0.04 | -0.33 | -0.03 | -0.114 |
| | ConditionNumber | -0.73 | -0.86 | -0.80 | -0.74 | -0.92 | -0.92 | -0.88 | -0.836 |
| | SelfCluster | -0.94 | -0.99 | -0.92 | -0.89 | -0.99 | -0.99 | -0.98 | -0.957 |
| | StableRank | 0.94 | 0.98 | 0.94 | 0.93 | 0.99 | 0.98 | 0.98 | 0.949 |
| Pubmed | CSOR | 0.87 | 0.86 | 0.91 | 0.94 | 0.95 | 0.92 | 0.71 | 0.882 |
| | SSOR | 0.95 | 0.95 | 0.96 | 0.97 | 0.98 | 0.97 | 0.93 | 0.961 |
| | RankMe | 0.94 | 0.94 | 0.96 | 0.96 | 0.98 | 0.94 | 0.93 | 0.936 |
| | NESum | -0.00 | 0.01 | 0.01 | 0.02 | 0.02 | -0.02 | 0.02 | 0.001 |
| | AlphaReQ | -0.40 | -0.59 | -0.48 | -0.47 | -0.89 | -0.85 | -0.09 | -0.538 |
| | Incoherence | -0.07 | -0.09 | -0.05 | -0.06 | -0.01 | -0.35 | 0.01 | -0.089 |
| | ConditionNumber | -0.75 | -0.76 | -0.79 | -0.79 | -0.90 | -0.87 | -0.84 | -0.810 |
| | SelfCluster | -0.94 | -0.95 | -0.97 | -0.96 | -0.99 | -0.97 | -0.94 | -0.960 |
| | StableRank | 0.96 | 0.95 | 0.96 | 0.97 | 0.98 | 0.97 | 0.92 | 0.960 |
| DBLP | CSOR | 0.85 | 0.92 | 0.95 | 0.92 | 0.94 | 0.92 | 0.85 | 0.921 |
| | SSOR | 0.97 | 0.99 | 0.99 | 0.98 | 0.99 | 0.98 | 0.98 | 0.981 |
| | RankMe | 0.95 | 0.99 | 0.97 | 0.96 | 0.98 | 0.96 | 0.97 | 0.971 |
| | NESum | -0.00 | -0.02 | 0.02 | 0.03 | 0.01 | 0.01 | -0.01 | 0.002 |
| | AlphaReQ | -0.54 | -0.86 | -0.76 | -0.67 | -0.91 | -0.93 | -0.76 | -0.781 |
| | Incoherence | 0.00 | 0.00 | -0.10 | -0.07 | 0.04 | -0.25 | -0.01 | -0.057 |
| | ConditionNumber | -0.81 | -0.90 | -0.86 | -0.83 | -0.91 | -0.93 | -0.92 | -0.882 |
| | SelfCluster | -0.97 | -0.99 | -0.98 | -0.97 | -0.99 | -0.98 | -0.98 | -0.981 |
| | StableRank | 0.97 | 0.99 | 0.99 | 0.98 | 0.99 | 0.98 | 0.98 | 0.981 |

Table 8: Experimental results on node classification: Spearman coefficients between ranking scores given by internal strategies and accuracy values.

| Dataset | Method | VGAE | GAE | ARGA | ARGVA | GAT | GIN | GraphSAGE | Average |
|---------|--------|------|-----|------|-------|-----|-----|-----------|---------|
| Cora | CSOR | 0.81 | 0.85 | 0.83 | 0.86 | 0.75 | 0.81 | 0.85 | 0.823 |
| | SSOR | 0.90 | 0.92 | 0.88 | 0.91 | 0.77 | 0.86 | 0.92 | 0.880 |
| | RankMe | 0.92 | 0.92 | 0.91 | 0.92 | 0.77 | 0.88 | 0.92 | 0.891 |
| | NESum | -0.02 | -0.03 | -0.02 | 0.03 | -0.07 | 0.04 | 0.02 | - 0.157 |
| | AlphaReQ | -0.36 | -0.47 | -0.3 | -0.53 | -0.73 | -0.8 | -0.13 | -0.332 |
| | Incoherence | 0.03 | 0.02 | -0.07 | -0.02 | -0.02 | -0.18 | 0.02 | -0.031 |
| | ConditionNumber | -0.81 | -0.82 | -0.74 | -0.78 | -0.74 | -0.82 | -0.84 | -0.793 |
| | SelfCluster | -0.90 | -0.92 | -0.89 | -0.91 | -0.76 | -0.88 | -0.92 | -0.883 |
| | StableRank | 0.89 | 0.92 | 0.88 | 0.91 | 0.77 | 0.85 | 0.92 | 0.877 |
| Citeseer | CSOR | 0.71 | 0.76 | 0.71 | 0.79 | 0.69 | 0.76 | 0.77 | 0.741 |
| | SSOR | 0.83 | 0.82 | 0.80 | 0.85 | 0.72 | 0.82 | 0.84 | 0.811 |
| | RankMe | 0.88 | 0.85 | 0.82 | 0.91 | 0.73 | 0.82 | 0.84 | 0.836 |
| | NESum | -0.02 | 0.02 | 0.01 | 0.04 | 0.04 | -0.02 | 0.03 | 0.014 |
| | AlphaReQ | -0.10 | -0.56 | -0.17 | -0.25 | -0.69 | -0.71 | -0.11 | -0.370 |
| | Incoherence | 0.06 | -0.05 | -0.17 | -0.03 | -0.04 | -0.32 | 0.02 | -0.076 |
| | ConditionNumber | -0.72 | -0.71 | -0.61 | -0.72 | -0.71 | -0.75 | -0.73 | -0.707 |
| | SelfCluster | -0.85 | -0.83 | -0.82 | -0.85 | -0.72 | -0.83 | -0.85 | -0.821 |
| | StableRank | 0.82 | 0.81 | 0.80 | 0.84 | 0.71 | 0.82 | 0.84 | 0.806 |
| Pubmed | CSOR | 0.63 | 0.83 | 0.79 | 0.75 | 0.87 | 0.74 | 0.79 | 0.771 |
| | SSOR | 0.74 | 0.88 | 0.86 | 0.80 | 0.87 | 0.82 | 0.91 | 0.840 |
| | RankMe | 0.81 | 0.90 | 0.89 | 0.86 | 0.86 | 0.79 | 0.90 | 0.859 |
| | NESum | -0.01 | 0.0 | 0.01 | 0.02 | 0.03 | 0.0 | -0.01 | 0.001 |
| | AlphaReQ | -0.42 | -0.62 | -0.48 | -0.46 | -0.78 | -0.73 | 0.01 | -0.497 |
| | Incoherence | -0.07 | -0.09 | -0.02 | -0.05 | 0.0 | -0.35 | 0.05 | -0.076 |
| | ConditionNumber | -0.67 | -0.75 | -0.74 | -0.70 | -0.78 | -0.74 | -0.74 | -0.731 |
| | SelfCluster | -0.74 | -0.89 | -0.85 | -0.78 | -0.87 | -0.81 | -0.91 | -0.836 |
| | StableRank | 0.71 | 0.87 | 0.85 | 0.78 | 0.87 | 0.82 | 0.91 | 0.830 |
| DBLP | CSOR | 0.66 | 0.74 | 0.68 | 0.71 | 0.73 | 0.71 | 0.74 | 0.710 |
| | SSOR | 0.74 | 0.77 | 0.71 | 0.78 | 0.77 | 0.77 | 0.83 | 0.767 |
| | RankMe | 0.79 | 0.80 | 0.74 | 0.81 | 0.76 | 0.76 | 0.83 | 0.784 |
| | NESum | 0.00 | -0.03 | 0.05 | 0.01 | -0.01 | 0.00 | -0.04 | -0.003 |
| | AlphaReQ | -0.54 | -0.70 | -0.63 | -0.67 | -0.73 | -0.72 | -0.63 | -0.660 |
| | Incoherence | -0.01 | -0.02 | -0.10 | -0.06 | -0.01 | -0.24 | -0.01 | -0.064 |
| | ConditionNumber | -0.73 | -0.73 | -0.68 | -0.76 | -0.74 | -0.71 | -0.79 | -0.736 |
| | SelfCluster | -0.75 | -0.78 | -0.71 | -0.78 | -0.76 | -0.75 | -0.83 | -0.765 |
| | StableRank | 0.73 | 0.76 | 0.70 | 0.77 | 0.77 | 0.77 | 0.82 | 0.761 |

# I COMPARISON OF SPATIAL-BASED AND SPECTRAL-BASED METHODS IN NODE AGGREGATION

Both spatial-based and spectral-based methods in Graph Neural Networks (GNNs) share a common objective: to effectively aggregate node information to produce meaningful node embeddings. However, they achieve this goal through different mechanisms. This appendix elucidates the relationship between these two approaches in their realization of node aggregation.

### SPATIAL-BASED METHODS

Spatial-based methods operate directly in the node domain. They use the adjacency matrix $A$ to represent the graph structure and perform node aggregation through message passing among neighboring nodes. Each node updates its embedding by aggregating information from its immediate neighbors using predefined aggregation functions such as mean, sum, or max.

**Key Characteristics**:

- **Graph Representation**: Uses the adjacency matrix $A$.
- **Node Domain Operations**: Aggregation is performed directly on nodes and their neighbors.
- **Message Passing**: Nodes receive and aggregate information from their neighboring nodes.

- **Examples**: Graph Convolutional Networks (GCN), GraphSAGE, Graph Attention Networks (GAT).

In GCN, for instance, the aggregation operation can be expressed as:

$$H^{(l+1)} = \sigma(\tilde{D}^{-1/2}\tilde{A}\tilde{D}^{-1/2}H^{(l)}W^{(l)})$$

where $\tilde{A} = A + I$ is the adjacency matrix with added self-loops, $\tilde{D}$ is the degree matrix, $H^{(l)}$ is the node embedding at layer $l$, $W^{(l)}$ is the learnable weight matrix, and $\sigma$ is an activation function.

SPECTRAL-BASED METHODS

Spectral-based methods, on the other hand, operate in the frequency domain. They leverage the graph Laplacian matrix $L = D - A$ and perform node aggregation through spectral filtering. The graph Laplacian is decomposed into its eigenvalues and eigenvectors, transforming the node features into the spectral domain. Aggregation is performed by filtering these spectral components, and the result is then transformed back to the node domain.

**Key Characteristics**:

- **Graph Representation**: Uses the Laplacian matrix $L$.
- **Frequency Domain Operations**: Aggregation is performed by filtering in the spectral domain.
- **Eigen Decomposition**: The Laplacian matrix is decomposed into eigenvalues and eigenvectors.
- **Examples**: Spectral CNN, ChebNet.

A typical spectral-based aggregation can be described as:

$$H = Ug(\Lambda)U^T X$$

where $U$ and $\Lambda$ are the eigenvectors and eigenvalues of the Laplacian matrix $L$, respectively, $X$ is the node feature matrix, and $g(\Lambda)$ is a spectral filter applied to the eigenvalues.

CONNECTING THE TWO APPROACHES

Despite their different mechanisms, both spatial-based and spectral-based methods aim to aggregate node information to produce effective node embeddings. The connection between these two approaches can be understood through their respective domains of operation:

- **Spatial-based methods** perform aggregation directly in the node domain by iteratively combining information from neighboring nodes.
- **Spectral-based methods** perform aggregation in the frequency domain by applying filters to the eigenvalues of the Laplacian matrix, capturing global graph properties.

Spatial-based methods can be viewed as a localized approximation of spectral methods. The direct message passing and aggregation in the node domain approximate the spectral filtering operations performed in the frequency domain. Both methods can be considered complementary, offering different perspectives and advantages for graph representation learning.

In summary, while spatial-based and spectral-based methods differ in their implementation, they share the fundamental goal of node information aggregation. Understanding the relationship between these methods provides a unified perspective on the diverse approaches used in GNNs for node embedding generation.

## J  GNN EVALUATOR

Existing work on evaluating graph embeddings without labels is limited. The GNN Evaluator (Zheng et al., 2023), which comes closest to this task, assesses the generalization ability of trained

GNNs on new datasets. In contrast, we aim to evaluate the quality of graph embeddings. While both methods share similar goals, they are not identical.

Let's examine the GNN Evaluator and discuss why it is not suitable for our needs.

Given a training graph $G_0$ and a well-trained GNN model $f$, we obtain the corresponding graph embedding $Z_0 = f(G_0)$.

The GNN Evaluator follows two steps:

## J.1 CONSTRUCT A DISCGRAPH SET

1. Simulate unseen meta-graphs using data augmentation:

$$G_{\text{meta}} = f_{\text{da}}(G_0)$$

where $G_{\text{meta}}$ represents generated meta-graphs by the data augmentation function $f_{\text{da}}$.

2. Input $G_{\text{meta}}$ to the trained GNN model to obtain graph embeddings and evaluate them to get AUC values:

$$Z_{\text{meta}} = f(G_{\text{meta}})$$
$$\text{auc}_{\text{meta}} = f_{\text{dt}}(Z_{\text{meta}})$$

where $Z_{\text{meta}}$ are the graph embeddings of $G_{\text{meta}}$, and $\text{auc}_{\text{meta}}$ represents their performance on downstream tasks.

3. Calculate DiscGraph, which is the spatial distance between $Z_{\text{meta}}$ and $Z_0$:

$$Z_{\text{disc}} = D(Z_{\text{meta}}, Z_0)$$

4. Form the training data for the GNN Evaluator as:

$$\{(Z_{\text{disc}}^1, \text{auc}_{\text{meta}}^1), \ldots, (Z_{\text{disc}}^i, \text{auc}_{\text{meta}}^i)\}$$

## J.2 TRAIN THE GNN EVALUATOR

The GNN Evaluator is a deep learning model that takes DiscGraph as input and outputs the corresponding performance (e.g., AUC value). It predicts a GNN model's performance on an unseen dataset by measuring the difference between the unseen dataset and the training dataset and mapping that difference to the AUC value.

However, the GNN Evaluator is not suitable for Hyperparameter Optimization (HPO) because it still requires labels for training data. We want to directly evaluate the performance of a given model on the training data without needing labels. The GNN Evaluator is designed for inductive learning, while our focus is on transductive learning.

## J.3 SUMMARY

- **GNN Evaluator**: Assesses GNN generalization on new datasets; requires labels for initial training.
- **Our Goal**: Evaluate graph embedding quality without labels; for unsupervised learning which is no labels during the whole process.

