# OpenReview forum: "Labels Are Not All You Need: Evaluating Node Embedding Quality without Relying on Labels"
_ICLR.cc/2025/Conference — Submitted to ICLR 2025_

### Official Review · Reviewer_5hnh · 2024-11-03

**Soundness:** 3
**Presentation:** 3
**Contribution:** 2
**Rating:** 3
**Confidence:** 3

**Summary:**

The paper proposed a new hyperparameter optimization method for unsupervised graph learning scheme. The new method leveraged a observation-driven scheme, which is to find prior belief (heuristic on the correlation between certain properties of the embeddings over hyperparameter configurations and the quality of the embedding) given observations and visualization over existing graph data and then rank the hyperparameter settings based on the observed heuristic scores. Specifically for graph, the paper proposed two beliefs: spectral based and spatial based to quantify the quality of the learned embeddings, corresponding to the separability of the learned embeddings. The experiments is conducted over four homophilic graph datasets and compared against existing hyperparameter optimization approaches, showing some improvements overall.

**Strengths:**

1. The presentation of the paper in the main text is clear and easy to follow. The motivation and the problem is clear and interesting. As a graph domain researcher, I personally find the sensitivity of the GNN models' performance over hyperparameter annoying and it took time to obtain optimal results especially when a deeper GNN is needed to alleviate oversmoothing.
2. The observation-driven approach is simple and the intuition is logical.
3. The visualization and analysis over the experimental data is comprehensive.

**Weaknesses:**

1. While the observation-driven approach is simple, I am not convinced that such CSOR or SSOR based heuristic is suitable for all graph datasets. As the experiments are carried only on limited four homophilic datasets, it is hard to be convinced that such heuristic can be generalized to other graph datasets.
2. The time complexity over both approaches are very high, especially for the Spectral based approach that involve SVD operation, which is known to be computationally expensive.
3. The improvements of the proposed approach is very limited compared to existing approaches such as RankMe.
4. There lack theoretical analysis on the embedding singular value over the quality of the embeddings, such as separability or the distance between embeddings over different HP configuration related to it separability.

**Questions:**

See weakness. Additionally, I am curious how many datasets are used to establish the prior belief in the experiment or is there random generated graph used?

---

> ### Author Response · Authors · 2024-11-28
>
> We sincerely thank the reviewer for their detailed feedback. Below, we address each concern.
>
> ---
>
> ## 1. Suitability and Generalization of CSOR and SSOR Approaches
>
> We understand and respect the concern regarding the generalization of CSOR and SSOR heuristics across diverse graph datasets. It is important to clarify that the primary goal of this paper is not to establish an evaluation metric that is universally applicable across all graph datasets. Instead, our work centers on demonstrating a systematic, observation-driven approach for proposing evaluation metrics by leveraging a two-step framework.
>
> We believe this approach offers valuable insight into how metrics can be devised based on phenomena observed within specific datasets and downstream tasks. We acknowledge that these phenomena may change for heterophilic datasets or different downstream tasks, and such shifts would necessitate returning to the first step of the framework to propose new metrics tailored to these contexts.
>
> ---
>
> ## 2. Time Complexity of CSOR and SSOR
>
> Regarding the complexity concerns, we note that CSOR and SSOR exhibit distinct scalability characteristics. Specifically, CSOR remains scalable for large graphs but faces limitations when dealing with a large search space, whereas SSOR’s reliance on SVD operations results in higher computational costs. We recognize that this is a common challenge across SVD-based methods.
>
> However, it is important to emphasize that the primary purpose of these two methods is to illustrate different means of depicting the distribution of embeddings rather than optimizing computational aspects. We acknowledge the reviewer’s point and agree that future work can focus on improving the efficiency of these methods.
>
> ---
>
> ## 3. Improvements Compared to Existing Approaches
>
> When considered through the lens of our proposed two-step framework, the relative performance of our approach serves to validate the utility and practicality of the framework. Our framework provides a systematic pathway for proposing corresponding evaluation metrics, and achieving results comparable to SOTA methods, such as RankMe, further underscores its effectiveness.
>
> ---
>
> ## 4. Theoretical Analysis of Embedding Singular Value and Quality
>
> We acknowledge the lack of theoretical analysis regarding embedding singular value properties, such as separability. However, we believe that the contribution of this paper lies in providing a direction for theoretical analysis. Without the instantiation of metrics such as CSOR and SSOR through our framework, conducting a theoretical analysis would lack direction.
>
> As highlighted in our discussion of UDR, its origin as a metric stemmed from the discovery of local orthogonality in VAE embeddings—a phenomenon identified through empirical observation. Our two-step framework aims to systematize this process, offering guidance for further theoretical exploration and metric development.

---

> > ### Comment · Reviewer_5hnh · 2024-12-03
> >
> > Thanks for the detailed response. I understand that this is an attempt to establish a two-step procedure for optimization. However, the paper provides limited evidence of how to achieve first step on other model or datasets: "metric stemmed from the discovery of local orthogonality in VAE embeddings" is only valid for VAE structure and the dataset used to observe such metric is not clear ( I asked this in the main comment question part but the reply didn't resolve my problem). So it is hard to be convinced that how one can reproduce a similar case on other datasets or models. I think a more clear and systematical path of how to obtaining step one is more important for the whole structure to be widely applicable and accepted.

---

### Official Review · Reviewer_9Wfy · 2024-11-03

**Soundness:** 2
**Presentation:** 3
**Contribution:** 2
**Rating:** 3
**Confidence:** 4

**Summary:**

The paper first empirically demonstrates that the effectiveness of learned node embeddings for downstream tasks is sensitive to the GNN method's hyperparameter settings. Then it introduces a framework to evaluate the quality of node embeddings without using labels. The proposed framework is comprised of two steps: building prior beliefs and quantifying the extent. Specifically, it instantiates the proposed frameworks with two methods: a Consensus-based Space Occupancy Rate (CSOR) method and a Spectral Space Occupancy Rate (SSOR). Extensive experiments on seven GNN methods demonstrate the effectiveness of the proposed methods.

**Strengths:**

1. This paper studies an interesting problem of how to evaluate the embedding quality without using labels.
2. Extensive empirical
3. This paper is easy to follow and understand.

**Weaknesses:**

1. The problems of the motivating experiments presented in Fig. 1
- Experiments are solely conducted for a single model (VGAE). More popular GNNs such as GCN, GAT or recent Transformer based methods should be used.
- Experiments are conducted on a single dataset (Cora), and a single task (node classification). Will other datasets or tasks also exhibit a similar trend?
2. It is not clear how the authors compute the distance between embedding matrices that have different embedding sizes, e.g., $Z_1\in\mathbb{R}^{N\times d_1}$ and  $Z_2\in\mathbb{R}^{N\times d_2}$.
3. Event with the same hyper-parameters, for the same node $x$, models $f_1, f_2$ trained with different seeds could learn embeddings $z_1$ and $z_2$ at different positions in the space. I'm not fully convinced that we can directly use the embedding distance as an evaluation unless more evidence is provided.
4. According to the formulation in 270-303, essentially, CSOR measures the distance between embeddings obtained by one HP $Z$ with the averaged embeddings obtained by all other HPs. Therefore, CSOR could give those bad-performing configurations a high score, which can also be observed in Fig 42. Do you have any ideas or insights for solving this problem?
5. There is a major issue for SSOR: if the learned embedding matrix is sort of random, then it seems that it will have a very high SSOR score.
6. It is not quite clear what's the relation between UDR in sec.2 and CSOR and SSOR introduced in the following sections.

**Questions:**

1. Please refer to weaknesses.
2. Have you tried other distance metrics other than Manhattan distance?

---

> ### Author Response · Authors · 2024-11-28
>
> We sincerely thank the reviewer for the detailed comments and valuable insights. Below, we provide responses to each point raised.
>
> ---
>
> ## 1. Issues with Fig. 1: Limited Models, Datasets, and Tasks
> First, we would like to clarify a potential misunderstanding: the goal of this paper is not to propose a "one-for-all" evaluation metric. Consequently, we did not aim for SSOR and CSOR to be universally applicable to recent Transformer-based methods.
>
> Regarding the models, datasets, and downstream tasks used in our experiments, we apologize if this was unclear. To clarify: our experiments are not solely focused on VGAE. Both the visualization analysis and performance validation experiments cover a total of seven GNN models: VGAE, GAE, GAT, GIN, ARGA, ARGVA, and GraphSAGE. Results for models other than VGAE have been provided in Appendices F and H.
> Cora and node classification represent only a subset of our experiments. The experiments encompass four datasets (Cora, Citeseer, Pubmed, and DBLP) and two downstream tasks: node classification and link prediction. More extensive experimental results and figures can be found in Appendices F and H.
>
> ---
>
> ## 2. Distance Computation with Different Embedding Sizes
> The reviewer's insight is highly valuable, and we acknowledge this as a limitation of CSOR. The inherent drawback of this consensus-based depiction of distribution is that it requires the candidate embeddings to have a consistent shape. In the future, we hope to explore other technical approaches that could better achieve a consensus-based method.
>
> ---
>
> ## 3. Seed Variance and Embedding Distance as an Evaluation Metric
> Let us clarify this point: the distances between embeddings themselves do not inherently possess the ability to evaluate embedding quality. The core idea is to capture the characteristics of the embedding distribution. Whether using embedding distances or singular values of embeddings, the aim is to depict the distributional traits of node embeddings. Specifically, the embedding distance serves to illustrate the overall embedding distribution across different hyperparameter settings by comparing different instances (i.e., embeddings generated under various hyperparameters).
> Therefore, trust does not stem from the embedding distance itself, but rather from our observations regarding how well-performing embeddings tend to exhibit specific distributional characteristics. The embedding distance is simply one method to depict this distribution.
>
> ---
>
> ## 4. CSOR Scoring Bad-Performing Configurations
> We completely understand the reviewer's concern regarding the robustness of CSOR. First, we would like to clarify that for hyperparameter optimization (HPO) problems, due to differences in downstream tasks and datasets, selecting the best-performing hyperparameter without label dependency is virtually impossible—this has been consistently demonstrated by existing works. Therefore, a more reasonable goal is to select one of the well-performing hyperparameters that is as close as possible to the optimal configuration.
>
> Regarding CSOR's robustness, we acknowledge similar concerns in the past. In extreme cases, where the majority of candidate hyperparameters perform very well while only a few—or even one—exhibit significantly poor performance, CSOR could indeed assign the highest score to the worst-performing configuration. However, CSOR's computation is not limited to a single node vector or a single dimension. For CSOR to fail under such extreme circumstances, the node embeddings produced by the bad-performing hyperparameter would need to consistently exhibit "very far" characteristics from well-performing embeddings across all node vectors and dimensions. For example, with 1,000 nodes and 8 dimensions, this would require 8,000 occurrences for CSOR to fail, making it much more robust than initially anticipated.

---

> ### Author Response · Authors · 2024-11-28
>
> ---
>
> ## 5. SSOR Scoring Random Embedding Matrices
> We had similar hypotheses in the early stages of our experiments. Such a situation can indeed occur, but it applies to random walk-based graph embedding models rather than message-passing GNN models. This arises due to differences in how these models, compared to message-passing mechanism-based GNNs, initialize embedding vectors. This is precisely where the value of the two-step framework proposed in this paper lies. For different types of models—whether random walk-based, message-passing mechanism-based, or other types of GNNs—there are notable differences in their initialization methods and how the distribution of embeddings evolves during training. The two-step framework systematically captures these differences and enables the formulation of distinct evaluation metrics tailored to a given model, dataset, and downstream task.
>
> ---
>
> ## 6. Relation Between UDR and CSOR/SSOR
> Let us clarify the structure of the paper. The first part revisits how existing work addresses the challenge of evaluating the quality of embeddings without labels and distills this process into the proposed two-step framework. UDR’s formulation plays a significant role in inspiring the two-step framework. It was designed as an evaluation metric based on the prior belief that VAEs exhibit local orthogonality—a belief that emerged from the observed phenomenon where VAE embeddings displayed disentanglement properties without explicit design. The logical flow here is to extract prior beliefs about embedding quality from observed phenomena and use these beliefs to design evaluation metrics.
>
> The relationship between UDR and CSOR/SSOR is as follows: by studying and summarizing the process through which UDR was formulated, we proposed the two-step framework, which was then instantiated to derive CSOR and SSOR.
>
> ---
>
> ## 7. Use of Distance Metrics Other Than Manhattan
> No, not yet. To clarify our choice of distance metric: the selection is based on the observations made during the first step of our framework. In this paper, we observed that as node embedding performance improves, the embeddings tend to become increasingly dispersed. Consequently, we chose to use the Manhattan distance as a natural measure to capture this behavior. We acknowledge that there are other distance metrics that could potentially capture more distributional information, and exploring these is part of our future work. Ideally, the most suitable distance metric would be custom-defined or selected based on the observed phenomenon to best quantify distributional characteristics.

---

### Official Review · Reviewer_9v5u · 2024-11-06

**Soundness:** 2
**Presentation:** 2
**Contribution:** 2
**Rating:** 5
**Confidence:** 4

**Summary:**

The authors propose a framework for evaluating node embedding quality in Graph Neural Networks (GNNs) without relying on label information. This framework introduces two novel metrics: the Consensus-based Spatial Occupancy Rate (CSOR) and the Spectral Space Occupancy Rate (SSOR).
CSOR and SSOR aim to assess embedding quality from spatial and spectral perspectives, respectively. The authors use unsupervised graph representation learning algorithms to build the framework for unsupervised training.
The authors also conduct experiments on different combinations of HP across multiple GNN models and citation network graph datasets, showing that these metrics can rank hyperparameter configurations effectively, identifying those that yield high-quality embeddings for downstream tasks.
The study provides unsupervised metrics for node embedding evaluation, potentially benefiting applications where labeled data is costly or unavailable.

**Strengths:**

New Metrics: The framework introduces two new metrics, CSOR and SSOR, which provide spatial and spectral assessments of embedding quality without labels. This is an innovative approach that aligns with the growing interest in unsupervised learning methods within GNN research.
Clarity: The paper is well-organized, and the explanations of CSOR and SSOR are detailed.

**Weaknesses:**

1. Method Comparison Clarity:
The proposed CSOR and SSOR methods may have limited significance, as their performance is generally inferior to that of RankMe. The authors do not provide a clear rationale for using CSOR or SSOR over RankMe, nor do they clarify the relationship between CSOR and SSOR. This lack of clarity makes it difficult to determine whether to prioritize HP chosen by CSOR or SSOR in future applications, thereby limiting the practical utility of these methods.

2. Use of Modified Models:
In the Node_Embedding_models.py file, the authors modify the original training methods of GAT and GIN, using them as encoders within the framework of the Graph Autoencoder (GAE) model for unsupervised training. However, this adaptation is not clearly stated in the paper, which could lead to confusion about the actual use of these models.

3. Lack of Comparative Experiment with Contrastive Learning:
The paper overlooks an important comparative experiment in unsupervised learning between generative learning and contrastive learning. Contrastive learning (eg. GraphCL), which focuses on distinguishing between similar and dissimilar instances at an abstract semantic level, offers simpler models with stronger capabilities compared to generative methods like GAE or VGAE.

4. Dataset Scope and Variety:
The paper primarily focuses on unsupervised learning for graphs, where the most common references are large-scale graph datasets due to the significant challenges posed by labeling costs. However, the four datasets used in this study are small-scale graphs, which do not fully reflect the real-world applicability of the proposed methods in large-scale settings. The authors should consider using larger graph datasets (eg. Reddit, OGBN-ARXIV) to strengthen the relevance of their work to large-scale graph learning scenarios.
Additionally, the choice of datasets is limited to citation networks, which may not represent the diverse range of graph structures encountered in real-world applications. To increase the generalizability of the findings, the authors should consider using datasets from a variety of domains, such as social networks, protein interaction networks, or recommendation systems, which would better highlight the robustness of CSOR and SSOR in different contexts.

5. Minor Comments:
Figure3: The caption is stated that RankMe is represented by "green circles" in the plots; however, the actual legend used is "white diamonds."
There are multiple instances where "hyperparameters" is misspelled as "hyperparamters"; please correct these throughout the paper.
In Section 5 "SSOR", the formula may contain an error where r should possibly be r−1.
Detailed code specifications (as the versions used) are missing.

**Questions:**

Please refer to the weaknesses above.

---

> ### Author Response · Authors · 2024-11-28
>
> First of all, we sincerely thank Reviewer 9v5u for the detailed feedback. Before addressing all comments one by one, we would like to clarify an important aspect of our work. This paper does not aim to propose evaluation metrics that are universally applicable to all types of datasets and GNN models. As evidenced by existing works, specific combinations of datasets, GNN models, and downstream tasks can give rise to distinct prior beliefs. Therefore, the core focus of this paper lies in the two-step framework, which provides a systematic and efficient approach to extracting prior beliefs from observed phenomena and designing evaluation metrics tailored to specific contexts.
>
> ## 1. Method Comparison Clarity
>
> We fully understand the reviewer’s concerns about the performance of CSOR and SSOR.
>
> - First, regarding SSOR, its performance is comparable to RankMe and, in some cases, slightly better.
> - Second, CSOR represents an attempt to depict the characteristics of embedding distribution from a consensus-based perspective (comparing instance by instance). While its current performance may not yet be outstanding, it has significant potential in terms of scalability for large graphs, which we believe makes it a valuable avenue for further development in future work.
>
> Most importantly, when viewed through the lens of the two-step framework, the instantiation of this framework into SSOR demonstrates performance comparable to the state-of-the-art RankMe. This serves as strong evidence of the framework’s effectiveness and its ability to guide the development of competitive evaluation metrics.
>
> ---
>
> ## 2. Use of Modified Models
>
> We sincerely thank the reviewer for pointing out this issue. This was an oversight on our part. To test the applicability of SSOR and CSOR while maintaining consistency in the training strategy, we used the autoencoder training framework for GAT, GIN, and GraphSAGE models. We will revise the paper to include a clear explanation of this adaptation to avoid any potential confusion about the actual use of these models.
>
> ---
>
> ## 3. Lack of Comparative Experiment with Contrastive Learning
>
> The reviewer’s observation is very insightful. As mentioned earlier, the purpose of the two-step framework is to enable the quick and straightforward development of evaluation metrics tailored to specific datasets, models, and downstream tasks. This approach is not limited to contrastive learning but could also be extended to random walk-based graph embedding models, which are part of our future plans. However, different types of models inherently bring different prior beliefs, necessitating distinct evaluation metrics.
>
> As the first attempt to systematically analyze and propose evaluation metrics in this domain, we aimed to present a lightweight study to facilitate discussion and exchange within the community. We appreciate the reviewer’s feedback and regard this as a valuable direction for future work.
>
> ---
>
> ## 4. Dataset Scope and Variety
>
> We acknowledge the limitations of the current study in using only small-scale citation datasets, and we agree that incorporating more diverse and larger-scale datasets would enhance the comprehensiveness of the evaluation. However, we would like to clarify that the inclusion of additional datasets is not aimed at directly testing the robustness of CSOR and SSOR. Instead, within our framework, the goal is to validate the effectiveness of the two-step framework across different contexts. Specifically, the key questions are whether the first step can observe characteristic distributional changes and whether the second step, using either spatial or spectral methods, can effectively quantify the prior belief.
>
> Given the constraints of the rebuttal period, it is challenging to incorporate such additional work at this time. However, we see this as an important direction for future research and appreciate the reviewer’s suggestions for expanding the dataset variety to include larger-scale graphs and domains such as social networks, protein interaction networks, and recommendation systems. These expansions will help further demonstrate the framework’s generalizability and practical applicability.
>
> ---
>
> ## 5. Minor Comments
>
> We sincerely appreciate the reviewer’s thorough reading of our paper and their constructive comments. We will carefully address these minor issues in the revised version of the manuscript.

---

> ### Comment · Reviewer_9v5u · 2024-11-30
>
> Dear authors,
>
> Thank you for your reply. I am afraid the responses have only partially solved my concerns. According to all the reviews and responses, I felt that this paper needs a major revision before being considered for acceptance. Thus, I will maintain my initial rating.
>
> Best,
> 9v5u

---

> > ### Author Response · Authors · 2024-11-30
> >
> > Dear Reviewer 9v5u,
> >
> > Thank you again for your detailed feedback and for taking the time to review our responses. We truly value your input and the opportunity to improve our manuscript.
> >
> > To better address your concerns in the next revision, we would greatly appreciate it if you could kindly clarify which of your initial concerns you feel have been resolved and which remain unaddressed. This would help us focus our efforts more effectively.
> >
> > Additionally, we would like to know your perspective on our clarification that the framework itself is the primary contribution of this work. If you find this focus acceptable or have further suggestions on how we can better demonstrate its value, your insights would be incredibly helpful in guiding our future revisions.
> >
> > Thank you again for your constructive comments. We look forward to your reply.
> >
> > Best regards, \
> > Authors

---

### Author Response · Authors · 2024-11-28
**Global Response**

First, we sincerely thank all reviewers for their valuable feedback. We would like to take this opportunity to clarify an important aspect of our work.

This paper does not aim to propose evaluation metrics that are universally applicable to all types of datasets and GNN models. As evidenced by existing works, specific combinations of datasets, GNN models, and downstream tasks can give rise to distinct prior beliefs. Therefore, the core focus of this paper lies in the two-step framework, which provides a systematic and efficient approach to extracting prior beliefs from observed phenomena and designing evaluation metrics tailored to specific contexts.

- **First**, regarding SSOR, its performance is comparable to RankMe and, in some cases, slightly better.
- **Second**, CSOR represents an attempt to depict the characteristics of embedding distribution from a consensus-based perspective (comparing instance by instance). While its current performance may not yet be outstanding, it has significant potential in terms of scalability for large graphs, which we believe makes it a valuable avenue for further development in future work.

**Most importantly**, when viewed through the lens of the two-step framework, the instantiation of this framework into SSOR demonstrates performance comparable to the state-of-the-art RankMe. This serves as strong evidence of the framework’s effectiveness and its ability to guide the development of competitive evaluation metrics.

---

### Meta-Review · Area_Chair_UF62 · 2024-12-11

**Metareview:**

This paper extends Unsupervised Disentanglement Ranking (UDR) (Duan et al., 2020) for hyperparameter optimization for unsupervised graph learning tasks. To this end, two instances, i.e., CSOR and SSOR, are presented and verified on four graphs. Although this paper presents an interesting topic, the following issues show it needs comprehensive improvements. Firstly, this paper only considers autoencoder-based unsupervised learning and ignores the contrastive ones. Secondly, the employed datasets are limited and small to justify it. Thirdly, the improvements over RankMe are limited. Finally, the complexity and the running time are not provided.

**Additional Comments On Reviewer Discussion:**

Although the authors provide feedback, all reviewers believe most of their concerns are not alleviated, and the paper needs major revision. Thus, they all keep the negative ratings.

---

### Decision · Program_Chairs · 2025-01-22

Reject